# LEARNING TO SOLVE OPTIMIZATION PROBLEMS CONSTRAINED WITH PARTIAL DIFFERENTIAL EQUATIONS

## ABSTRACT

Partial differential equation (PDE)-constrained optimization arises in many scientific and engineering domains, such as energy systems, fluid dynamics, and material design. In these problems, the decision variables (e.g., control inputs or design parameters) are tightly coupled with the PDE state variables, and the feasible set is implicitly defined by the governing PDE constraints. This coupling makes the problems computationally demanding, as it requires handling high-dimensional discretizations and dynamic constraints. To address these challenges, this paper introduces a learning-based framework that integrates a dynamic predictor with an optimization surrogate. The dynamic predictor, a novel time-discrete Neural Operator (Lu et al., 2021), efficiently approximates system trajectories governed by PDE dynamics, while the optimization surrogate leverages proxy optimizer techniques (Kotary et al., 2021) to approximate the associated optimal decisions. This dual-network design enables real-time approximation of optimal strategies while explicitly capturing the coupling between decisions and PDE dynamics. We validate the proposed approach on benchmark PDE-constrained optimization tasks including Burgers' equation, heat equation, and voltage regulation, and demonstrate that it achieves solution quality comparable to classical control-based algorithms such as the Direct Method and Model Predictive Control (MPC), while providing up to four orders of magnitude improvement in computational speed.

## 1 INTRODUCTION

Partial differential equations (PDEs) are central to modeling a wide range of physical and engineering systems, capturing phenomena that evolve jointly over space and time. Applications include fluid transport, which can be modeled with Burgers' equations which are key in fluid mechanics and gas dynamics, thermal processes captured by heat equations, often used in financial mathematics and image analysis, and voltage dynamics in power systems. Optimizing decisions in such domains requires solving *PDE-constrained optimization problems*, where control or design objectives must be optimized while ensuring consistency with the underlying governing PDE dynamics.

Despite their importance, PDE-constrained optimization remains computationally intractable. Traditional approaches typically rely on finite element and finite difference discretizations combined with adjoint-based optimization. These methods often suffer from high dimensionality, high computational cost, and convergence difficulties when applied to nonlinear dynamics. For example, when applied to the nonlinear Burgers' equations, our experiments show that the runtimes of the adjoint-based and Nonlinear MPC approaches run in approximately $120\ s$ and $878\ s$; such computational cost hinder their applicability in large-scale or real-time decision-making tasks.

In various classes of continuous optimization tasks not constrained by PDEs, proxy optimizers (Kotary et al., 2021) have recently emerged as a powerful tool to quickly find approximate solutions. In parallel, researchers have focused on developing a variety of surrogates (Raissi et al., 2017; Kovachki et al., 2024; Li et al., 2021; Chen et al., 2018) for capturing system dynamics. Our approach is a unique combination of elements from both these areas, and consists of a dual-network architecture to tackle PDE-constrained optimization problems where both the state dynamics and the control decisions evolve in space and time.

**Contributions.** Specifically, the paper makes the following contributions: **(1)** It introduces a novel learning-based method to efficiently approximate solutions of PDE-constrained optimization problems. Our approach, PDE-OP (Optimization Proxy), consists of a dual-network architecture in which a dynamic predictor captures the system dynamics, while a surrogate controller approximates the optimal control actions. These components are trained in a synergistic manner in a self-supervised fashion using a primal–dual method. **(2)** It shows that the proposed time-discrete Neural Operator accurately captures the dynamics governed by PDEs. **(3)** It demonstrates that the proposed framework achieves decision quality comparable to state-of-the-art numerical solvers, such as Model Predictive Control and the Adjoint-sensitivity method, while providing several orders of magnitude improvements in computational speed. The approach is validated on a set of widely adopted PDE-constrained optimization tasks, including Burgers', heat, and voltage equations. *We believe that these results could boost the adoption of learning-based methods for control problems that require high-fidelity dynamic modeling and near real-time optimal decision-making.*

## 2 RELATED WORK

Here we provide a brief overview of existing approaches for optimal control problems, learning-based methods for constrained optimization and system dynamics. Please refer to Appendix 9.1 for a comprehensive review of these methodologies.

A central paradigm for decision-making in dynamical systems is *Model Predictive Control* (Mayne et al., 2000), which repeatedly solves optimization problems on a finite horizon using predicted system states. Although effective, its reliance on repeated solver calls makes it computationally intractable, limiting real-time applicability. Similarly, exact PDE-constrained methods, such as the *adjoint-sensitivity* (Hinze et al., 2008; Tröltzsch, 2010), rely on time–space discretization of the governing equations, after which the resulting high-dimensional system is incorporated into an optimization problem. While theoretically well-founded, these approaches become quickly impractical as the problem size grows, especially in the presence of nonlinearities, as in the case of this work.

In an orthogonal community, machine learning and optimization researchers have developed *proxy optimizers* (Kotary et al., 2021) methods, which learn mappings from system parameters to optimal decisions, enabling real-time approximation of constrained optimization tasks. Both supervised (Fioretto et al., 2020) and self-supervised (Park & Van Hentenryck, 2023) approaches have been proposed, with feasibility representing a recurring challenge in this field. In parallel, *surrogate models for PDE dynamics* have been developed, including Physics-Informed Neural Networks (Raissi et al., 2019) and operator-learning approaches such as Fourier Neural Operators (Li et al., 2021) and DeepONets (Lu et al., 2021), which approximate PDE solution operators. More recently, researchers have introduced operator learning for PDE-constrained optimization, such Surrogate-Based Trajectory Optimization (SBTO) (Hwang et al., 2022), an open-loop control strategy that employs a neural surrogate of the dynamics, and DeepONet MPC (de Jong et al., 2025), a receding-horizon, closed-loop control method that uses the same surrogate model. These approaches, however, are either limited to capturing the system dynamics and cannot cope with PDE-constrained optimization tasks, or suffer of computational overhead to due repeated call to numerical optimization solvers. This work overcomes these limitations with a novel integration of operator-learning surrogates with proxy optimizers, coupling PDE dynamics with decision-making to enable real-time PDE-constrained optimization.

## 3 SETTINGS AND GOALS

This paper focuses on optimization problem constrained by a system of partial differential equations:

$$\underset{u:Q\to\mathbb{R}}{\text{Minimize}} \quad \overbrace{L\big(y(T,\cdot)\big) \;+\; \int_0^T\!\!\int_\Omega \Phi\big(u(t,x),y(t,x)\big)\,dx\,dt + \int_0^T\!\!\int_\Omega \|u(t,x)\|_2^2 dx\,dt}^{J(u(t,x),y(t,x))} \tag{1a}$$

$$\text{s.t.} \quad \partial_t y(t,x) \;=\; \mathcal{I}_t\big(y(t,x),u(t,x),t,x\big), \quad (t,x)\in Q, \tag{1b}$$

$$\partial_x y(t,x) \;=\; \mathcal{I}_x\big(y(t,x),u(t,x),t,x\big), \quad (t,x)\in Q, \tag{1c}$$

$$y(t_0,x) \;=\; y_{t_0}(x), \quad x\in\Omega, \tag{1d}$$

$$y(t,x) \;=\; y_\partial(t,x), \quad (t,x)\in[t_0,T]\times\partial\Omega, \tag{1e}$$

$$\boldsymbol{g}(u(t,x),y(t,x) \;\le\; \boldsymbol{0}, \quad \boldsymbol{h}(u(t,x),y(t,x)) \;=\; \boldsymbol{0}, \quad (t,x)\in Q. \tag{1f}$$

In this formulation, the control action is denoted by $u : Q \to \mathbb{R}$, where $Q = [t_0,T] \times \Omega$ is the space–time domain, with $t_0, T \in \mathbb{R}$ being the initial time instant and the time horizon, respectively, and $\Omega$ being the spatial domain, e.g., if $x \in \mathbb{R}$, then $\Omega \subseteq \mathbb{R}$. The system state is represented by $y : Q \to \mathbb{R}$. $t$ and $x$ denote the independent temporal and spatial variable, respectively. The optimization objective (1a) combines a terminal cost $L\big(y(T,\cdot)\big)$, which depends on the state at the final time $T$, with a running cost $\Phi$, integrated over time and space, and an effort cost, which is the energy used to control the system.

The system dynamics are governed by two PDE constraints. Equation (1b) models the temporal evolution of the state, while (1c) captures spatial dependencies, for example through diffusion–reaction relations. Initial conditions (1d) prescribe the state distribution at $t = t_0$, while boundary conditions (1e) specify the state behavior on the spatial boundary $\partial\Omega$. Finally, (1f) represents additional inequality and equality constraints on $(u,y)$, which may encode feasibility requirements or application-specific restrictions.

**Temperature control example.** In the context of thermal regulation problems, the decision variable $u(t,x)$ represents localized actuators applied along the medium, while the state variable $y(t,x)$ describes the temperature distribution over space and time. These two components are tightly coupled: the control inputs directly enter the governing PDE as forcing terms, shaping the spatio-temporal evolution of the state, while the optimization objective is evaluated on the resulting state trajectory. System dynamics (1b–1c) follow a controlled reaction–diffusion equation, which models heat

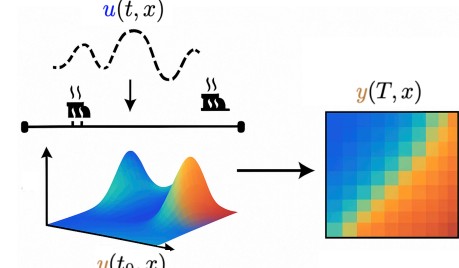

Figure 1: Temperature regulation process: control actions $u(t,x)$ steer the system from the initial state $y(t_0,x)$ towards a target state $y(T,x)$.

diffusion, dissipation, and externally applied control. The initial condition (1d) specifies the temperature distribution at $t = t_0$, and the boundary condition (1e) enforces zero-flux (Neumann) boundaries, preventing heat transfer across the domain boundary. The objective 1a combines a terminal cost to match the target temperature profile, a running cost for deviations along the horizon, and a control effort penalty. A simplified illustration is provided in Fig.1, while a full mathematical description is given in Appendix 9.8.

**Challenges.** While fundamental for many applications, Problem (1) presents three key challenges:

1. Finding optimal solutions to Problem (1) is computationally intractable. Even without the differential equation constraints, the decision version of the problem alone is NP-hard in general.
2. Achieving high-quality approximations of the system dynamics (1b)-(1c) in near real-time, poses the second significant challenge. The non-linearity of these dynamics further complicate the task.
3. Finally, the integration of the system dynamics in the optimization framework renders traditional numerical methods impractical for real-time applications.

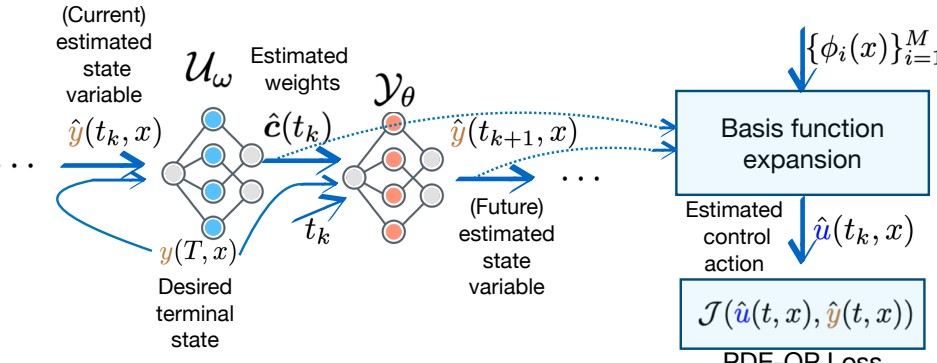

Figure 2: At each time instant $t_k$, PDE-OP uses a dual network architecture consisting of a proxy optimization model $\mathcal{U}_\omega$ to estimate the basis functions weights $c(t_k)$ and construct the control action $\hat{u}(t_k, x)$, and a neural-DE model $\mathcal{Y}_\theta$ to estimate the state-variables $\hat{y}(t_{k+1}, x)$, with the objective function $\mathcal{J}(\hat{u}(t, x), \hat{y}(t, x))$ capturing the overall loss.

## 4 PDE-OPTIMIZATION PROXY

To tackle the challenges discussed above, we introduce *PDE-Optimization Proxy* (PDE-OP): a fully differentiable surrogate for PDE-constrained optimization. PDE-OP adopts a dual-network architecture: the first network, $\mathcal{U}_\omega$, approximates the optimal decision variable $u^*(t, x)$, while the second, $\mathcal{Y}_\theta$, predicts the corresponding state variables $y(t, x)$ using a discrete-time neural operator architecture. Here, $\omega$ and $\theta$ denote the trainable parameters of $\mathcal{U}$ and $\mathcal{Y}$, respectively. These two components are jointly trained in an end-to-end differentiable manner, and a schematic illustration of their interaction is provided in Figure 2. In what follows, we first describe each network individually and then detail their integration within the proposed learning framework.

**Learning Goal.** PDE-OP trains on a distribution $\Pi$ of problem instances generated by different states $y(t, x)$ for various $t$ and $x$ values. For instance, in the thermal regulation example, this distribution corresponds to the variety of temperature profiles that arise from different control inputs. With reference to (1a), the learning objective is:

$$\underset{\omega, \theta}{\text{Minimize}} \ \mathbb{E}_{y(t,x) \sim \Pi} \left[ \mathcal{J} \left( \mathcal{U}_\omega(t, x, \hat{y}), \mathcal{Y}_\theta(t, x, \hat{u}) \right) \right] \tag{2a}$$

$$\texttt{s.t. } (1b)\text{--}(1f), \tag{2b}$$

where, to simplify the notation, we denote the estimated control variable as $\hat{u}(t, x) = \mathcal{U}_\omega(t, x, \hat{y})$, while the state variable estimate is denoted as $\hat{y}(t, x) = \mathcal{Y}_\theta(t, x, \hat{u})$. Given the complexity of solving Problem (2), the goal is to develop a fast and accurate neural PDE optimization surrogate. This approach uses the concept of *proxy optimizers* (Kotary et al., 2021), which is detailed next.

### 4.1 SURROGATE CONTROLLER

The first objective is to establish a neural network-based mapping that transforms the system state $y(t, x)$ of a PDE-constrained optimization problem 1 to a control action $u(t, x)$. Practically, this mapping is implemented as a neural network $\mathcal{U}_\omega$, which takes the current system state $\hat{y}(t, x)$ and desired terminal state $y(T, x)$ as input, and outputs an array of coefficients $\hat{c}(t)$ that are then used to estimate the corresponding optimal decision variables. These coefficients are the weights used to combine a set of basis functions to represent the control action, and are described next.

**Basis function representation of the control action.** To approximate the optimal control action using a neural network, we represent the control trajectory $u(t, x)$ as a linear combination of *basis functions*. This representation has also been proposed in Feng et al. (2025); Li et al. (2024). This design choice is motivated by the fact that, although the output of a neural network lies in a finite-dimensional space, combining it with continuous basis functions $\phi_i(x)$ allows us to construct a continuous approximation of the control trajectory $u(t, x)$. In this way, the controller model $\mathcal{U}_\omega$ operates in a low-dimensional space, while the basis functions map these finite number of outputs

into a continuous-time signal. As shown in the additional experiments provided in Appendix 9.10.2, direct modeling of continuous-time control actions, either with a neural network or classical method, leads to qualitatively similar performance to our basis expansion approach, but while the computational cost of our method remains extremely low, the running time of the classical methods grows significantly. These results show that our method is suitable for real-time tasks.

Consider the control action $u(t_k, x)$, where $t_k$ denotes a collocation point; at each time instant $t_k$, the control $u(t_k, x)$ is expressed as a weighted sum of $M$ continuous basis functions $\{\phi_i(x)\}_{i=1}^M$:

$$u(t_k, x) = \sum_{i=1}^M c_i(t_k)\,\phi_i(x), \qquad t_k = 0, \ldots, T, \tag{3}$$

where $\boldsymbol{c}(t_k) = [c_1(t_k) \ \ldots \ c_M(t_k)]^\top$ denotes the coefficients of the basis functions at time instant $t_k$. For example, in one of our experiments, we adopt sinusoidal basis functions, e.g. $\phi_1(x) = \sin(\pi x/X)$, $\phi_2(x) = \cos(\pi x/X)$, ..., $\phi_{2M-1}(x) = \sin(M\pi x/X)$, $\phi_{2M}(x) = \cos(M\pi x/X)$, which provide a Fourier-like expansion of the control signal. Here $M$ denotes the number of sinusoidal modes included in the expansion, with higher values of $M$ allowing for better approximations of the control variable.

**Surrogate controller model training.** The role of the controller surrogate $\mathcal{U}_\omega$ is to predict the array of coefficients $\boldsymbol{c}(t_k) = [c_1(t_k) \ \ldots \ c_M(t_k)]^T$. Formally, the network $\mathcal{U}_\omega$ approximates the function

$$U : \big(y(t_k, x), y(T, x)\big) \mapsto \boldsymbol{c}(t_k), \tag{4}$$

which maps the state variable $y(t_k, x)$ and desired terminal state $y(T, x)$ to the corresponding controller weights $\boldsymbol{c}(t_k)$, subsequently used to construct the control action $u(t_k, x)$ according to (3). The surrogate controller model $\mathcal{U}_\omega$ is trained *on-the-fly*, using a dataset $\mathcal{D} = \{(\hat{\boldsymbol{y}}^i(t_k, x), y(T, x)\}_{i=1}^N$, where $\hat{y}(t_k, x)$ denotes the state variables estimated by the dynamic predictor $\mathcal{Y}_\theta$, $y(T, x)$ is the desired terminal state and $N$ is the number of samples. Here, bold symbols $\boldsymbol{y}(t_k, x)$ denote the space-discrete version of the corresponding vector $y(t_k, x)$, with each collocation point corresponding to a specific output of $\mathcal{Y}_\theta$, e.g. $\boldsymbol{y}(t_k, x) = \big[\hat{y}(t_k, x_1) \ \ldots \ \hat{y}(t_k, x_n)\big]^T$. Note that this setting is *unsupervised*: optimal weights $\boldsymbol{c}(t_k)$ are unknown, and candidate weights $\hat{\boldsymbol{c}}(t_k) = \mathcal{U}_\omega(\hat{\boldsymbol{y}}(t_k, x), y(T, x))$ are evaluated solely by the objective value $\mathcal{J}$ in (1a) they induce. Figure 2 illustrates the interaction between PDE-OP's proxy optimizer $\mathcal{U}_\omega$ and dynamic predictor $\mathcal{Y}_\theta$ at time instant $t = t_k$. Given the predicted state $\hat{y}(t_k, x)$ (for $k = 0$ we have $\hat{y}(t_0, x) = y_0(x)$), the surrogate controller outputs an estimate of $\boldsymbol{c}(t_k)$ which are then used to construct the control action $u(t_k, x)$; this is subsequently used by $\mathcal{Y}_\theta$ to predict the next state $\hat{y}(t_{k+1}, x)$. This process is repeated at each collocation point, until the terminal state $y(T, x)$ is estimated.

To build our surrogate controller, we build on the proxy optimizer method proposed in Park & Van Hentenryck (2023), which allows self-supervised training without access to precomputed optimal solutions, while directly encouraging feasibility through primal–dual updates. Once trained, the model $\mathcal{U}_\omega$ can be used to generate near-optimal solutions at low cost. The next section discusses how the state variables are estimated using a discrete-time Neural Operator architecture.

## 4.2 Discrete-time Neural Operator

The second component of PDE-OP is a discrete-time Neural Operator architecture, here denoted as $\mathcal{Y}_\theta$, which models the PDE solution operator. The proposed dynamic predictor employs two subnetworks: the first, which we denote *branch network*, encodes input functions (e.g., control signals), and the second, called *trunk network*, encodes the coordinates where the solution is evaluated (e.g., spatial locations). Their outputs are then combined to approximate the target operator. This architecture is inspired by (Lu et al., 2021), a neural operator framework designed to approximate mappings between infinite-dimensional function spaces.

Recall that the PDE-constrained problem couples system dynamics and decision variables. To take account for this coupling, the proposed architecture incorporates both the system state and the controller output. Given the current estimated state $\hat{\boldsymbol{y}}(t_k, x)$ and the control weights $\hat{\boldsymbol{c}}(t_k)$ estimated by $\mathcal{U}_\omega$, the network $\mathcal{Y}_\theta$ outputs an estimate of the state at the subsequent collocation point in time:

$$\hat{\boldsymbol{y}}(t_{k+1}, x) = \mathcal{Y}_\theta(t, x, \hat{\boldsymbol{y}}(t_k, x), \hat{\boldsymbol{c}}(t_k)). \tag{5}$$

Rolling out (5) across $k = 0, \ldots, T-1$ yields the full estimated state trajectory, $\{\hat{\boldsymbol{y}}(t_k, x)\}_{k=0}^T$.

**Branch network.** The branch net encodes the estimated system state sampled at $n$ fixed sensor locations, $\hat{\boldsymbol{y}}(t_k, x)$, together with estimated control weights $\hat{\boldsymbol{c}}(t_k)$, and maps this concatenated input to a $d$-dimensional vector of coefficients $\boldsymbol{b}(\hat{\boldsymbol{y}}(t_k, x), \hat{\boldsymbol{c}}(t_k)) \in \mathbb{R}^d$.

**Trunk network.** The trunk net encodes the spatial query coordinate $x$, producing a latent feature vector $\boldsymbol{\gamma}(x) \in \mathbb{R}^d$.

The estimated subsequent (in time) state $\hat{\boldsymbol{y}}(t_{k+1}, x)$ at location $x_i$ is obtained by combining the branch coefficients and trunk features through an inner product:

$$\hat{y}(t_{k+1}, x_i) \; = \; \sum_{j=1}^{d} b_j(\hat{y}(t_k, x_i), \hat{\boldsymbol{c}}(t_k)) \; \gamma_j(x_i), \;\; i = 1, \ldots, n. \tag{6}$$

The remainder of this section details the method by which the state variables are precisely estimated using the proposed discrete-time Neural Operator.

**Model initialization and training.** Given the proposed time-discrete neural operator model $\mathcal{Y}_\theta$ takes as input the weights of the basis functions estimated by the surrogate controller $\mathcal{U}_\theta$, it is practical to initialize the dynamic predictor model. To achieve this, we initialize $\mathcal{Y}_\theta$ in a supervised fashion; its training data are generated by sampling time-varying control weight sequences $\boldsymbol{c}(t_k)$ from a gaussian distribution, e.g., $c_i(t_k) \sim \mathcal{N}(0, \sigma_i)$, and by specifying the terminal state $y(T, x)$. The corresponding solution $y(t, x)$ satisfying (1b)–(1f) is computed using a numerical PDE solver. Each trajectory is discretized into one-step training samples of the form $(y(t_k, x), \boldsymbol{c}(t_k)) \mapsto y(t_{k+1}, x)$. Formally, the dataset of training pairs is given by $\mathcal{D} = \Big\{ \big( y^i(t_k, x), \boldsymbol{c}^i(t_k), y^i(t_{k+1}, x) \big) \; \Big| \; t_k \in \{0, \ldots, T-1\} \Big\}_{i=1}^{N}$. The network $\mathcal{Y}_\theta$ is trained to minimize the prediction error between its outputs and the ground-truth values. Formally, this model is trained by minimizing the loss:

$$\min_{\theta} \; \mathbb{E}_{(\boldsymbol{c}, y) \sim \mathcal{D}} \Big[ \big\| \hat{y}(t, x) - y(t, x) \big\|^2 \Big], \tag{7}$$

where $\mathcal{D}$ denotes the dataset of pairs $(\boldsymbol{c}, y)$. This training strategy ensures that $\mathcal{Y}_\theta$ learns to approximate the PDE solution operator given a distribution of weights $\boldsymbol{c}(t)$ approximating the distribution of estimated weights generated by the surrogate controller $\mathcal{U}_\omega$, so that it can provide accurate PDE-solutions given the predicted controller weights $\hat{\boldsymbol{c}}(t)$.

### 4.3 Handling Static and Dynamics Constraints Jointly

To integrate the proposed time-discrete neural operator within the decision process, this paper proposes a Primal Dual (LD) learning approach, which is inspired by the augmented Lagrangian relaxation technique (Hestenes, 1969) adopted in classic optimization. In Lagrangian relaxation, some or all the problem constraints are relaxed into the objective function using Lagrangian multipliers to capture the penalties induced by violating them. The proposed formulation leverages Lagrangian duality to integrate trainable regularization terms that encapsulates violations of the constraint functions. When all the constraints are relaxed, the violation-based Lagrangian of problem (1) is

$$\underset{u}{\text{Minimize}} \; \mathcal{J}(u(t, x), y(t, x)) + \boldsymbol{\lambda}_{h'}^\top |\boldsymbol{h}'(u(t, x), y(t, x))| + \boldsymbol{\lambda}_g^\top \max(0, \boldsymbol{g}(u(t, x), y(t, x))),$$

where $\mathcal{J}, \boldsymbol{g}$ are defined in (1a), and (1f), respectively, and $\boldsymbol{h}'$ is defined as follows,

$$\boldsymbol{h}'(u, y) \; = \; \begin{bmatrix} \partial_t y(t, x) - \mathcal{I}_t\big(y(t, x), u(t, x), t, x\big) \\ \partial_x y(t, x) - \mathcal{I}_x\big(y(t, x), u(t, x), t, x\big) \\ y(0, x) - y_0(x) \\ y(t, x) - y_\partial(t, x) \\ \boldsymbol{h}(u, y) \end{bmatrix} = \boldsymbol{0}. \tag{8}$$

It denotes the set of *all* equality constraints of problem (1), thus extending the constraints $\boldsymbol{h}$ in (1e), with the system dynamics (1b)-(1c) and the initial and boundary conditions equations (1d)-(1e) written in an implicit form as above. Therein, $\boldsymbol{\lambda}_{h'}$ and $\boldsymbol{\lambda}_g$ are the vectors of Lagrange multipliers associated with functions $\boldsymbol{h}'$ and $\boldsymbol{g}$, e.g. $\lambda_{h'}^i, \lambda_g^j$ are associated with the $i$-th equality $h_i'$ in $\boldsymbol{h}'$ and

$j$-th inequality $g_j$ in $\boldsymbol{g}$, respectively. The key advantage of expressing the system dynamics (1b)-(1c) and initial and boundary conditions (1d)-(1e) in the same implicit form as the equality constraints $\boldsymbol{h}$, (as shown in (8)), is treating the system dynamics in the same manner as the constraint functions $\boldsymbol{h}$. This enables us to satisfy the system dynamics and the static set of constraints ensuring that they are incorporated seamlessly into the optimization process.

The proposed primal-dual learning method uses an iterative approach to find good values of the *primal* $\omega, \theta$ and dual $\boldsymbol{\lambda}_{h'}, \boldsymbol{\lambda}_g$ variables; it uses an augmented modified Lagrangian as a loss function to train the prediction $\hat{u}, \hat{y}(t)$ as employed

$$\mathcal{L}^{\text{PDE-OP}}(\hat{u}, \hat{y}) = \mathcal{J}(\hat{u}(t,x), \hat{y}(t,x)) + \boldsymbol{\lambda}_{h'}^{\top}|\boldsymbol{h}'(\hat{u}(t,x), \hat{y}(t,x))| + \boldsymbol{\lambda}_g^{\top}\max(0, \boldsymbol{g}(\hat{u}(t,x), \hat{y}(t,x))), \quad (9)$$

where $\mathcal{J}(\hat{u}(t,x), \hat{y}(t,x))$ represents the total objective cost, while $\boldsymbol{\lambda}_{h'}^{\top}|\boldsymbol{h}'(\hat{u}(t,x), \hat{y}(t,x))|$ and $\boldsymbol{\lambda}_g^{\top}\max(0, \boldsymbol{g}(\hat{u}(t,x), \hat{y}(t,x)))$ measures the constraint violations incurred by prediction $\hat{u}(t,x)$ and $\hat{y}(t,x)$. The loss function in (9) combines an objective term to ensure minimization of the objective function (1a) with a weighted penalty on violations of the constraint functions $\boldsymbol{h}', \boldsymbol{g}$. This accounts for the contribution of both networks $\mathcal{U}, \mathcal{Y}$ during training, which is balanced via iterative updates of the multipliers in (9) based on the amount of violation of the associated constraint function.

At iteration $j + 1$, finding the optimal parameters $\omega, \theta$ requires solving

$$\omega^{j+1}, \theta^{j+1} = \arg\min_{\omega, \theta} \mathbb{E}_{y(t,x) \sim \mathcal{D}} \left[ \mathcal{L}^{\text{PDE-OP}} \left( \mathcal{U}_{\omega^j}^{\boldsymbol{\lambda}^j}(t,x,\hat{y}), \mathcal{Y}_{\theta^j}^{\boldsymbol{\lambda}^j}(t,x,\hat{u}) \right) \right], \quad (10)$$

where $\mathcal{U}_{\omega}^{\boldsymbol{\lambda}^j}$ and $\mathcal{Y}_{\theta}^{\boldsymbol{\lambda}^j}$ denote the DE-OP's optimization and predictor models $\mathcal{U}_\omega$ and $\mathcal{Y}_\theta$, at iteration $j$, with $\boldsymbol{\lambda}^j = [\boldsymbol{\lambda}_{h'}^j, \boldsymbol{\lambda}_g^j]^{\top}$. This step is approximated using a stochastic gradient descent method

$$\omega^{j+1} = \omega^j - \eta\nabla_\omega \mathcal{L}^{\text{PDE-OP}} \left( \mathcal{U}_{\omega^j}^{\boldsymbol{\lambda}^j}(t,x,\hat{y}), \mathcal{Y}_{\theta^j}^{\boldsymbol{\lambda}^j}(t,x,\hat{u}) \right)$$
$$\theta^{j+1} = \theta^j - \eta\nabla_\theta \mathcal{L}^{\text{PDE-OP}} \left( \mathcal{U}_{\omega^j}^{\boldsymbol{\lambda}^j}(t,x,\hat{y}), \mathcal{Y}_{\theta^j}^{\boldsymbol{\lambda}^j}(t,x,\hat{u}) \right) \quad (11)$$

where $\eta$ denotes the learning rate and $\nabla_\omega \mathcal{L}$ and $\nabla_\theta \mathcal{L}$ represent the gradients of the loss function $\mathcal{L}$ with respect to the parameters $\omega$ and $\theta$, respectively, at the current iteration $k$. Importantly, this step does not recompute the training parameters from scratch, but updates the weights $\omega, \theta$ based on their value at the previous iteration. Finally, the Lagrange multipliers are updated as

$$\boldsymbol{\lambda}_{h'}^{j+1} = \boldsymbol{\lambda}_{h'}^j + \rho|\boldsymbol{h}'(\hat{u}(t,x), \hat{y}(t,x))|$$
$$\boldsymbol{\lambda}_g^{j+1} = \boldsymbol{\lambda}_g^j + \rho\max\left(\boldsymbol{0}, \boldsymbol{g}(\hat{u}(t,x), \hat{y}(t,x))\right) \quad (12)$$

where $\rho$ denotes the Lagrange step size and the max operation is performed element-wise.

Next, we evaluate our method on a range of PDE-constrained optimization tasks with varying levels of complexity, including cases space-time varying control actions and nonlinear PDE dynamics.

## 5 Experimental Setting

In this section, we evaluate the proposed method on three optimal control tasks of increasing complexity. In these tasks the dynamics are described by the following PDE equations: Voltage Dynamics (linear PDE with time-invariant control input $u(x)$), 1D Heat equation (linear PDE with time-variant control input $u(t,x)$), and 1D viscous Burgers' equation (nonlinear PDE with time-variant control input $u(t,x)$). PDE-OP is compared against state-of-the-art approaches for optimal control problems: the Direct method (Betts, 2010), the Adjoint-Sensitivity method (Cao et al., 2003), (nonlinear) Model Predictive Control (NMPC) (Schwenzer et al., 2021), Surrogate-based Trajectory Optimization (SBTO) (Hwang et al., 2022), and DeepONet MPC (de Jong et al., 2025). These methods are widely adopted in the literature, while Direct method, Adjoint-Sensitivity method and NMPC rely on numerical PDE solvers whenever dynamics or gradients are evaluated, the neural baselines, SBTO and DeepONet MPC, use trained surrogate models as dynamic predictor. The Open-loop (direct/adjoint) Control method requires a full forward rollout per gradient step and a backward adjoint evaluation, while MPC solves a finite-horizon problem at every control step (i.e., a receding horizon). The runtime of these methods is determined by repeated calls to the underlying numerical

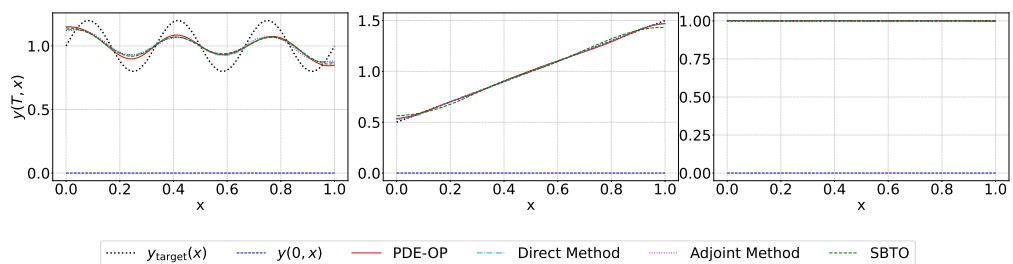

Figure 3: Voltage Equation. Comparison of PDE-OP and other methods for three target profiles: $y_{\text{target}}(x) = 1 + 0.2\sin(6\pi x)$ (left), $y_{\text{target}}(x) = x + 0.5$ (center), and $y_{\text{target}}(x) = 1$ (right).

PDE solver and grows with the spatial grid size ($n$), horizon length ($T$), and complexity (e.g., non-linearity) of the PDE dynamics. For each experiment and method, we report the accuracy, measured as the $L_2$ error between the terminal state and the predicted terminal state, $\|y(T,x) - \hat{y}(T,x)\|_2^2$, the inference time of PDE-OP, and runtime of the methods, both expressed in seconds. A comprehensive description of these methods is provided in Appendix 9.2–9.7, while we refer to Appendix 9.9 for additional experimental results on the predictive tasks of PDE-OP's $\mathcal{Y}_\theta$.

Additionally, Appendix 9.10 reports an ablation to evaluate the impact of our basis functions, introduced in Section 4.1). In summary, this experiment shows that using a full formulation is impractical for NMPC, pushing solve times well beyond real-time limits. Additionally, we show that Adjoint methods suffer from inaccurate gradients, which leads to a marked decrease in solution quality. Furthermore, Appendix 13 describes how the number of basis functions($M$) affects the PDE-OP model performance across different target profiles.

### 5.1 Voltage Control Optimization

Here we consider the problem of regulating voltage dynamics in a *leaky* electrical transmission line. The system is modeled by a controlled reaction–diffusion PDE with Neumann boundary conditions; the goal is to steer the voltage profile $y(t,x)$ toward a prescribed target distribution $y_{\text{target}}(x)$. We refer to Appendix 9.8 for a mathematical description of the problem, datasets, and related specifics. In this setting the control action is a sole function of the space: $u(t,x) = u(x)$.

**Results.** First, as the performance of PDE-OP's controller relies on accurate surrogate modeling of the PDE dynamics, in Appendix 9.9 we show that PDE-OP's $\mathcal{Y}_\theta$ faithfully captures the system's dynamics. Figure 3 presents a comparison between PDE-OP's predicted terminal state $\hat{y}(T,x)$ and that obtained using the methods, over 3 target profiles $y_{\text{target}}(x)$: sinusoidal, linear and constant. This figure shows that the voltage distribution produced by PDE-OP is well aligned with those produced by the Direct, Adjoint-sensitivity method, and SBTO. As shown in the first row of Table 1, the direct method is the slowest, requiring $26.097s$ and yielding an MSE of $7.46 \times 10^{-3}$. The Adjoint method runs in $0.755s$ with a similar accuracy, $7.13 \times 10^{-3}$. The SBTO improves upon the classical computation times, running in $0.520$ s, but yields a slightly higher MSE of $7.91 \times 10^{-3}$. Notably, PDE-OP is the fastest, by a significant margin: it runs in only $0.035$ s ($\sim 21.6\times$ faster than Adjoint; $\sim 746\times$ faster than Direct; $\sim 14.8\times$ faster than SBTO) and yield the lowest MSE, with $6.95 \times 10^{-3}$.

These results show the enormous potential of the proposed approach, which produces solutions that are qualitatively similar (or even slightly better) than the classical methods, while being between 20 and 750 times faster than them.

### 5.2 Optimal Control of the Heat Equation

Next, we consider the task of controlling the temperature distribution in a 1D domain, modeled by a reaction–diffusion equation. Starting from an initial temperature profile, the objective is to drive the system toward a prescribed target distribution while minimizing control effort. We refer to Appendix 9.8 for a mathematical description of this problem, datasets, and related specifics.

**Results.** As in the previous task, since the surrogate controller $\mathcal{U}_\omega$ depends on accurate predictions from $\mathcal{Y}_\theta$, in Appendix 9.9 we show that $\mathcal{Y}_\theta$ accurately captures the PDE equations. Figure 4 il-

Table 1: Comparison across PDE tasks and classical methods. All results use sinusoidal target profiles. Further details and results for linear and constant profiles can be found in Appendix 9.9.

| PDE System | Control Type | Methods | Runtime | Accuracy |
|---|---|---|---|---|
| **Voltage Control** (Reaction-Diffusion) | Open-Loop | Direct (Finite Diff.) | 26.097s | 0.00746 |
| | | Adjoint-Method | 0.755s | 0.00713 |
| | | SBTO | 0.520s | 0.00791 |
| | | **PDE-OP (ours)** | **0.035s** | **0.00695** |
| **1D Heat Equation** (Linear PDE) | Closed-Loop | LMPC | 0.384s | **0.00035** |
| | | Adjoint-Method | 0.662s | **0.00035** |
| | | SBTO | 15.706s | 0.00040 |
| | | DeepONet MPC | 144.304s | **0.00035** |
| | | **PDE-OP (ours)** | **0.124s** | **0.00035** |
| **1D Burgers' Equation** (Nonlinear PDE) | Closed-Loop | NMPC | 878.667s | 0.00007 |
| | | Adjoint-Method | 120.696s | 0.00023 |
| | | SBTO | 59.652s | **0.00001** |
| | | DeepONet-MPC | 98.371s | 0.01392 |
| | | **PDE-OP (ours)** | **0.062s** | 0.00030 |

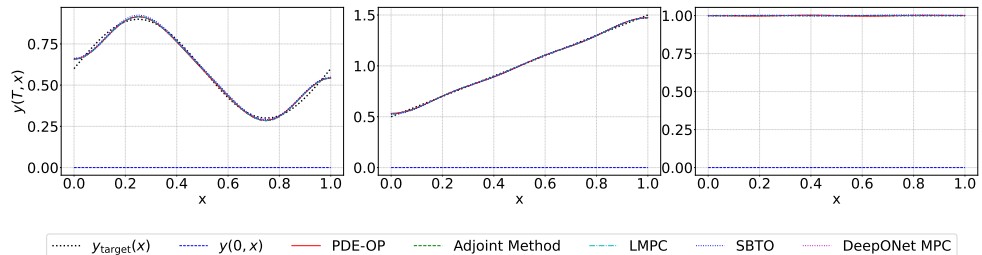

Figure 4: 1D Heat Equation. Comparison of PDE-OP and other methods for three target profiles: $y_{\text{target}}(x) = 0.6 + 0.3(\sin(2\pi x))$(left), $y_{\text{target}}(x) = x + 0.5$(center), and $y_{\text{target}}(x) = 1$(right).

lustrates the comparison of PDE-OP 's predicted terminal state $\hat{y}(T, x)$ and those obtained using the classical methods. As shown in Table 1, LMPC, the Adjoint method, and DeepONet MPC all achieve the same high accuracy (MSE $= 3.5 \times 10^{-4}$), with PDE-OP matching this performance. SBTO yields a slightly higher error (MSE $= 4.0 \times 10^{-4}$). However, notable differences appear in runtime. While the learning-based baselines struggle in this setting—with SBTO requiring 15.706 s and DeepONet MPC taking 144.304 s—PDE-OP is by far the fastest method at 0.124 s. This represents a $3.1\times$ speedup over LMPC (0.384 s) and a $5.3\times$ speedup over the Adjoint method (0.662 s). These results demonstrate that even with time-variant control actions, our method produces solutions qualitatively comparable to classical methods at a significantly reduced cost, avoiding the computational overhead observed in other learning-based baselines.

## 5.3 OPTIMAL CONTROL OF THE 1D BURGERS' EQUATION

Here we consider the problem of controlling the 1D viscous Burgers' equation with Dirichlet boundary condition. This equation is a standard benchmark in physics-based machine learning as it models fluid dynamics phenomena like shock waves and turbulence, arising from the interplay between nonlinear advection and linear diffusion. We refer to Appendix 9.8 for a mathematical description of the problem, datasets, and specifics related. As for the previous task, here the control action is a function of both time and space and expressed through a set of prespecified basis functions.

**Results.** As in the previous tasks, we first verify that PDE-OP's neural operator $\mathcal{Y}_\theta$ accurately captures the system's dynamics even under nonlinearities, with numerical results reported in Appendix 9.9. Figure 5 illustrates the comparison of PDE-OP's predicted terminal state $\hat{y}(T, x)$ and those obtained using the baseline methods. As shown in Table 1, the learning-based baseline SBTO achieves the lowest error (MSE $= 1.0 \times 10^{-5}$) followed by NMPC (MSE $= 7.0 \times 10^{-5}$), but

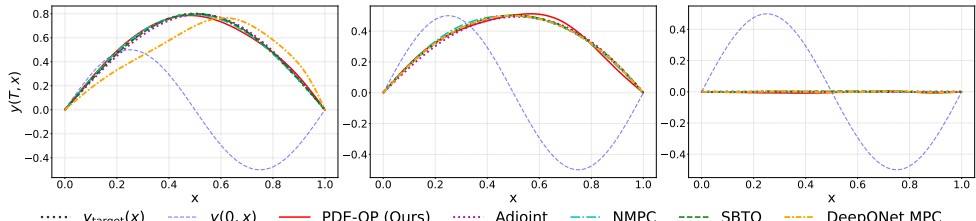

Figure 5: 1D Burgers' Equation. Comparison of PDE-OP and other methods for three target profiles: $y_{\text{target}}(x) = 0.8(\sin(\pi x))$ (left), $y_{\text{target}}(x) = 2x(1-x)$ (center), and $y_{\text{target}}(x) = 0$ (right).

they require significant computation times of $59.652$ s and $878.667$ s, respectively. DeepONet-MPC struggles with accuracy in this setting (MSE $= 1.39 \times 10^{-2}$), while the Adjoint method yields a competitive MSE $= 2.3 \times 10^{-4}$ in $120.696$ s. In contrast, PDE-OP produces qualitatively comparable predictions (MSE $= 3.0 \times 10^{-4}$) while being the fastest at $0.062$ s by a significant margin—*approximately $1.42 \times 10^4 \times$ faster than NMPC and $1.95 \times 10^3 \times$ faster than the Adjoint method!*

## 6 CONCLUSION

This work was motivated by the efficiency requirements associated with solving partial differential equations-constrained optimization problems. It introduced a novel learning-based framework, PDE-OP, which incorporates differential equation constraints into optimization tasks for real-time applications. The approach uses a dual-network architecture, with one approximating the control strategies and another capturing the associated PDEs. This architecture exploits a primal-dual method to ensure that both the dynamics dictated by the PDEs and the optimization objectives are concurrently learned and respected. This integration enables end-to-end differentiation, allowing for efficient gradient-based optimization, and, to our knowledge, solves PDE-constrained optimization problems in real-time for the first time. Empirical evaluations across a set of PDE-constrained optimization tasks illustrated PDE-OP's capability to address these complex challenges adeptly. Our comprehensive results demonstrate the effectiveness of our approach and its broad potential applicability across various scientific and engineering domains where system dynamics arise in optimization or control processes.

## 7 ETHICS STATEMENT

This work does not present any foreseeable ethical concerns. It focuses on methodological contributions to PDE-constrained optimization, and no human subjects, sensitive data, or ethically sensitive applications are involved.

## 8 REPRODUCIBILITY STATEMENT

To reproduce the results shown in this paper, we release the full implementation of PDE-OP and baseline methods in the supplementary material, including training and evaluation scripts, hyperparameter files, as well as data generation and baseline algorithms. The hyperparameters and implementation details of PDE-OP are described in Appendix 9.12 and Section 4.1–4.3, respectively. The parameters and implementation details of the baselines methods are described in Appendix 9.12 and Appendix 9.2–9.5, respectively. The code supports all PDE-constrained optimization tasks studied in this work, namely Burgers', heat, and voltage equations. Upon publication, we will also release a public repository to facilitate future research and extensions.

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

# 9 APPENDIX

## 9.1 RELATED WORK

A central paradigm for decision-making coupled with dynamical systems is *Model Predictive Control (MPC)* (Mayne et al., 2000). It repeatedly solves an optimization problem over a finite time horizon using predicted system states. While MPC provides high-quality control decisions, its reliance on repeatedly solving an optimization problem makes it computationally intractable and limits its applicability in real-time settings. In a similar fashion, exact PDE-constrained optimization methods rely on time–space discretization of the governing equations, after which the resulting high-dimensional system is incorporated into an optimization problem (Hinze et al., 2008; Tröltzsch, 2010). While theoretically well-founded, these approaches become quickly impractical as the problem size grows, especially in the presence of nonlinear dynamics or when real-time feasibility is required. To mitigate these limitations, surrogate models have been developed for reduced-order adjoint solutions (Zahr & Farhat, 2014), sensitivity PDEs (Dihlmann & Haasdonk, 2015), and learning PDE operators (Hwang et al., 2022), offering significant computational advantages. Nevertheless, these methods often face trade-offs between accuracy and efficiency and limited adaptability across operating regimes(Kovachki et al., 2024; Lu et al., 2021; Willcox & Ghattas, 2021).

Differentiable programming frameworks have recently gained traction as an alternative route to PDE-constrained optimization, enabling end-to-end differentiation through numerical solvers using automatic differentiation (see Nzoyem Ngueguin et al. (2023) for a comparative study). These approaches typically follow a discretize-then-optimize paradigm and provide accurate gradients through the PDE solver itself. Our approach, in contrast, avoids differentiation through a full PDE solver by leveraging operator-learning surrogates coupled with a proxy optimizer, which enables real-time inference and amortized optimization.

In parallel, to approximate solutions of constrained optimization problems in real-time setting, machine learning researchers have developed a class of methods known as *proxy optimizers*. These methods employ machine learning models, to learn mappings from problem parameters or system configurations to optimal decisions, thereby significantly accelerating inference (Kotary et al., 2021) compared to classical optimization solvers. Both supervised approaches (Fioretto et al., 2020), which rely on precomputed optimal solutions, and self-supervised ones (Park & Van Hentenryck, 2023), which exploit knowledge of the problem structure, have been proposed, showing promising results. A recurring challenge in this setting is ensuring feasibility: learned solutions may violate problem constraints. This issue has been tackled through penalty-based training (Fioretto et al., 2020), implicit-layer formulations that embed constraints directly into the model architecture (Donti et al., 2020), or post-processing projections to restore feasibility (Kotary et al., 2024).

A large body of recent work has also focused on *learning surrogates for PDE dynamics*. Physics-Informed Neural Networks (Raissi et al., 2019) directly encode PDE residuals into the loss function, enabling solutions that adhere to the governing equations. Neural operator approaches, such as the Fourier Neural Operator (Li et al., 2021) and DeepONet (Lu et al., 2021), learn mappings between infinite-dimensional function spaces, which allows them to approximate solution operators of *families of PDEs*, with strong generalization across different boundary conditions and coefficients (Kovachki et al., 2024). Other operator-learning frameworks (Li et al., 2020; Bhattacharya et al., 2021) and kernel-based methods (Raissi et al., 2021) have also been developed. While these approaches excel at reproducing PDE solutions, they are typically not designed to incorporate decision variables or to directly solve PDE-constrained optimization problems. The closest related work is (Vito et al., 2024), where the authors propose a learning-based surrogate solver to tackle ODE or SDE-constrained optimization tasks. *Our approach goes further, by integrating operator-learning surrogates within the optimization pipeline, thereby coupling the representation of PDE dynamics with proxy optimizers and enabling real-time solutions to PDE-constrained optimization tasks.*

## 9.2 DIRECT METHOD (FINITE-DIFFERENCE APPROACH)

The direct method treats the simulator as a black box mapping the bounded control vector $\boldsymbol{u}$ to the discrete objective $J(\boldsymbol{u})$ and employs a standard optimizer to minimize it. Here the control variable is discretized at each collocation point: $\boldsymbol{u} \in \mathbb{R}^{N_x}$.

**Mechanism.** We use L-BFGS-B(Byrd et al., 1995) with per–coordinate box bounds $u_{\min} \leq u_j \leq u_{\max}$ to minimize (26). No analytic gradients are provided; instead, numerical derivatives are queried via finite differences. With a forward-difference method,

$$\frac{\partial J}{\partial u_j} \approx \frac{J(\boldsymbol{u} + \epsilon_j \boldsymbol{e}_j) - J(\boldsymbol{u})}{\epsilon_j}, \qquad \epsilon_j = \eta \left(1 + |u_j|\right), \tag{13}$$

where $\boldsymbol{e}_j$ is the $j$-th unit vector and $\eta$ is a small scalar (e.g., $\mu = 10^{-6}$). For higher accuracy (to which corresponds a double computational cost), the central-difference method is adopted:

$$\frac{\partial J_h}{\partial u_j} \approx \frac{J_h(\boldsymbol{u} + \epsilon_j \boldsymbol{e}_j) - J_h(\boldsymbol{u} - \epsilon_j \boldsymbol{e}_j)}{2\epsilon_j}. \tag{14}$$

**Computational cost.** Each gradient evaluation requires $N_x + 1$ simulator calls with forward differences (or $2N_x$ with central differences), where $N_x$ denotes the number of spatial collocation points. Since one simulator call is a complete rollout (22) over $N_t$ time steps, the complexity per iteration is $\mathcal{O}\big((N_x+1)\, N_t\, N_x\big)$. This method is robust and simple, but its cost scales linearly with $N_x$, which becomes prohibitive for dense spatial discretizations.

## 9.3 Adjoint Sensitivity Method for Open-Loop Problem

The Adjoint-Sensitivity Method efficiently computes exact gradients of PDE-constrained objectives w.r.t. high-dimensional control variables. Crucially, the cost of one gradient evaluation is *one forward solve + one backward adjoint solve*, regardless of the number of control parameters. We derive the continuous adjoint for the open-loop, time-invariant Voltage Control Optimization (5.1) and then we present the discrete-time adjoint state used in our Crank–Nicolson (CN) implementation.

**Function spaces & BCs.** Consider $u \in L^2(\Omega)$, $y \in L^2([0,T] \times \Omega)$, and impose homogeneous Neumann BCs on $V$ so that the boundary terms vanish in the variational calculus; the adjoint inherits the same Neumann BCs.

The problem consists of finding $u(x)$ minimizing

$$J(u) = \tfrac{1}{2} \int_\Omega (y(T,x) - y_{\text{target}}(x))^2 \, dx \; + \; \tfrac{\gamma}{2} \int_\Omega u(x)^2 \, dx, \tag{15}$$

subject to

$$\frac{\partial y}{\partial t} - D\frac{\partial^2 y}{\partial x^2} + \beta\big(y - y_{\text{ref}}(x)\big) - \alpha u(x) = 0, \qquad \forall (t,x) \in [0,T] \times \Omega, \tag{16a}$$

$$y(0,x) = y_0(x), \qquad \forall x \in \Omega, \tag{16b}$$

$$\partial_x y(t,x) = 0, \qquad \forall (t,x) \in [0,T] \times \partial\Omega, \tag{16c}$$

where $\gamma > 0$ is the regularization weight in the control effort.

**Lagrangian.** We define

$$\mathcal{L}(u,y,p) = J(u) + \int_0^T \int_\Omega p(t,x)\left[\frac{\partial y}{\partial t} - D\frac{\partial^2 y}{\partial x^2} + \beta\big(y - y_{\text{ref}}(x)\big) - \alpha u(x)\right] dx\, dt, \tag{17}$$

where $p(t,x)$ is the adjoint state. By imposing $\nabla_y J = 0$ and integrating by parts in $t$ and $x$ (with Neumann BCs) yields the adjoint system.

**Adjoint PDE.**

$$-\frac{\partial p}{\partial t} - D\frac{\partial^2 p}{\partial x^2} + \beta\, p = 0, \qquad \partial_x p|_{\partial\Omega} = 0. \tag{18}$$

**Terminal condition.**

$$p(T,x) = y(T,x) - y_{\text{target}}(x). \tag{19}$$

**Gradient w.r.t. $u(x)$.** Because $u$ is time-invariant, variation w.r.t. $u$ gives

$$\nabla_{u(x)} J \; = \; \gamma\, u(x) \; - \; \alpha \int_0^T p(t,x)\, dt. \tag{20}$$

We optimize $\boldsymbol{u}$ with L-BFGS-B under elementwise bounds $u_{\min} \leq u_i \leq u_{\max}$; no projection step is needed.

**Semi-discrete forward dynamics.** To simplify the notation, let $\boldsymbol{y}(t) = \begin{bmatrix} y(t, x_1) & \dots & y(t, x_i) & \dots & y(t, x_{N_x}) \end{bmatrix} \in \mathbb{R}^{N_x}$ be the discretized version of $y(t, x)$ on a uniform grid with Neumann BCs, $\boldsymbol{L} \in \mathbb{R}^{N_x \times N_x}$ be the the Neumann Laplacian, and $\boldsymbol{u} \in \mathbb{R}^{N_x}$ the *static* control vector with elements $u(x_i)$. Define $\boldsymbol{A} := D\,\boldsymbol{L} - \beta\,\boldsymbol{I}_{N_x}$ and $\boldsymbol{s} := \beta\,\boldsymbol{y}_{\text{ref}}$. Then

$$\dot{\boldsymbol{y}}(t) = \boldsymbol{A}\,\boldsymbol{y}(t) + \alpha\,\boldsymbol{u} + \boldsymbol{s}. \tag{21}$$

**Crank–Nicolson (CN) step.** With $\Delta t = T/N_t$ and $t_k = k\Delta t$, CN gives

$$\left(\boldsymbol{I}_{N_x} - \tfrac{\Delta t}{2}\boldsymbol{A}\right)\boldsymbol{y}(t_{k+1}) = \left(\boldsymbol{I}_{N_x} + \tfrac{\Delta t}{2}\boldsymbol{A}\right)\boldsymbol{y}(t_k) + \Delta t\,\alpha\,\boldsymbol{u} + \Delta t\,\boldsymbol{s}. \tag{22}$$

Equivalently, the affine one-step map is

$$\boldsymbol{y}(t_{k+1}) = \boldsymbol{A}_{\text{CN}}\,\boldsymbol{y}(t_k) + \boldsymbol{B}_{\text{stat}}\,\boldsymbol{u} + \boldsymbol{d}, \tag{23}$$

$$\boldsymbol{A}_{\text{CN}} = \left(\boldsymbol{I}_{N_x} - \tfrac{\Delta t}{2}\boldsymbol{A}\right)^{-1}\left(\boldsymbol{I}_{N_x} + \tfrac{\Delta t}{2}\boldsymbol{A}\right), \qquad \boldsymbol{B}_{\text{stat}} = \Delta t\left(\boldsymbol{I}_{N_x} - \tfrac{\Delta t}{2}\boldsymbol{A}\right)^{-1}\alpha\,\boldsymbol{I}_{N_x}, \tag{24}$$

$$\boldsymbol{d} = \Delta t\left(\boldsymbol{I}_{N_x} - \tfrac{\Delta t}{2}\boldsymbol{A}\right)^{-1}\boldsymbol{s}. \tag{25}$$

**Discrete objective.** With $\boldsymbol{M} = \Delta x\,\boldsymbol{I}_{N_x}$ we have:

$$J_h(\boldsymbol{u}) = \tfrac{1}{2}\|\boldsymbol{y}_{N_t} - \boldsymbol{y}_{\text{target}}\|_{\boldsymbol{M}}^2 + \tfrac{\gamma}{2}\|\boldsymbol{u}\|_{\boldsymbol{M}}^2 = \tfrac{1}{2}(\boldsymbol{y}_{N_t} - \boldsymbol{y}_{\text{target}})^\top \boldsymbol{M}(\boldsymbol{y}_{N_t} - \boldsymbol{y}_{\text{target}}) + \tfrac{\gamma}{2}\,\boldsymbol{u}^\top \boldsymbol{M}\boldsymbol{u}. \tag{26}$$

**Discrete adjoint & gradient w.r.t. static $\boldsymbol{u}$.**

$$\boldsymbol{\lambda}_{N_t} = \boldsymbol{M}(\boldsymbol{y}_{N_t} - \boldsymbol{y}_{\text{target}}), \qquad \boldsymbol{\lambda}_k = \boldsymbol{A}_{\text{CN}}^\top \boldsymbol{\lambda}_{k+1}, \quad k = N_t - 1, \dots, 0, \tag{27}$$

$$\nabla_{\boldsymbol{u}} J_h = \gamma\,\boldsymbol{M}\,\boldsymbol{u} - \sum_{k=0}^{N_t-1} \boldsymbol{B}_{\text{stat}}^\top \boldsymbol{\lambda}_{k+1} = \gamma\,\boldsymbol{M}\,\boldsymbol{u} - \alpha\,\Delta t\left(\boldsymbol{I}_{N_x} - \tfrac{\Delta t}{2}\boldsymbol{A}\right)^{-\top}\left(\sum_{k=0}^{N_t-1} \boldsymbol{\lambda}_{k+1}\right). \tag{28}$$

**Computational cost.** One forward CN rollout (22) and one backward adjoint sweep (27) yield the exact discrete gradient (28); then, $\boldsymbol{u}$ is updated with L-BFGS-B under box bounds. In 1D, the CN matrix is time-invariant and can be prefactored once, so that each step reduces to cheap triangular-matrix solver call; the totla cost per iteration is $\mathcal{O}(N_t N_x)$.

### 9.4 Time-Varying Adjoint for Closed-Loop Baselines

The adjoint method extends naturally to a time-varying control $u(t, x)$. The continuous adjoint PDE and terminal/boundary conditions remain as per Eq. (18)–(19), while the gradients w.r.t. $u(t, x)$ are calculated as:

$$\nabla_{u(t,x)} J = \gamma\,u(t, x) - \alpha\,p(t, x). \tag{29}$$

We use this time-varying adjoint for the Heat and Burgers experiments (please refer to Section 5.2 and 5.3); the discrete forms below match our CN implementations.

**Heat optimization problem — CN discretization.** With $\boldsymbol{y}(t_{k+1}) = \boldsymbol{A}\boldsymbol{y}(t_k) + \boldsymbol{B}\boldsymbol{u}(t_k) + \boldsymbol{g}$ and a quadratic running/terminal costs, the discrete adjoint equation and per-step gradient are

$$\boldsymbol{\lambda}_{N_t} = \boldsymbol{Q}\,(\boldsymbol{y}_{N_t} - \boldsymbol{y}_{\text{target}}), \qquad \boldsymbol{\lambda}_k = \boldsymbol{A}^\top \boldsymbol{\lambda}_{k+1} + \boldsymbol{T}_Q\,(\boldsymbol{y}(t_k) - \boldsymbol{y}_{\text{target}}), \tag{30}$$

$$\nabla_{\boldsymbol{u}(t_k)} J_h = \boldsymbol{R}\,\boldsymbol{u}(t_k) + \boldsymbol{B}^\top \boldsymbol{\lambda}_{k+1}. \tag{31}$$

This is the gradient used by LMPC in Section 5.2. $\boldsymbol{Q}, \boldsymbol{T}_Q$, and $\boldsymbol{R}$ represent the weight matrices.

**Burgers' optimization problem — implicit CN step.** We write one CN step as the implicit map $\boldsymbol{y}(t_{k+1}) = \mathcal{G}(\boldsymbol{y}(t_k), \boldsymbol{u}(t_k); \Delta t)$ with Dirichlet endpoints enforced at each step. Let

$$A_k = I - \tfrac{\Delta t}{2}\,J(\boldsymbol{y}(t_{k+1})), \qquad B_k = -I - \tfrac{\Delta t}{2}\,J(\boldsymbol{y}(t_k)), \tag{32}$$

where $J(\boldsymbol{y}) = -\boldsymbol{D}\operatorname{diag}(\boldsymbol{y}) + \nu\boldsymbol{D}^2$ is the Jacobian of the semi-discrete residual, and $\boldsymbol{B}$ stacks the actuator shapes. For running cost $L_k = \frac{\Delta t}{2}\|\boldsymbol{y}(t_{k+1}) - \boldsymbol{y}_{\text{target}}\|_2^2 + \frac{\alpha\Delta t}{2}\|\boldsymbol{u}(t_k)\|_2^2$, the closed-loop adjoint/backward sweep is

$$A_k^\top\boldsymbol{q}_k = \Delta t\left(\boldsymbol{y}(t_{k+1}) - \boldsymbol{y}_{\text{target}}\right), \tag{33}$$

$$\boldsymbol{p}_k = \Delta t\left(\boldsymbol{y}(t_k) - \boldsymbol{y}_{\text{target}}\right) - B_k^\top\boldsymbol{q}_k, \tag{34}$$

$$\nabla_{\boldsymbol{u}(t_k)}J_h = \Delta t\left(\alpha\,\boldsymbol{u}(t_k) + \boldsymbol{B}^\top\boldsymbol{q}_k\right), \tag{35}$$

which are the NMPC equations we implemented and related to the experiments in Section 5.3. Here $\boldsymbol{q}_k$ and $\boldsymbol{p}_k$ are adjoint the variables.

## 9.5 MODEL PREDICTIVE CONTROL (MPC)

In what follows, to simplify notation we omit the dependency from the spatial variable $x$. At each time instant $t_k$, MPC solves a finite-horizon problem of $N_p$ steps given inputs $\{\boldsymbol{u}(t_i)\}_{i=k}^{k+N_p-1}$; it applies the first input $\boldsymbol{u}(t_k)$, then applies shifts and iterates(receding horizon).

**Linear MPC for the Heat equation.** With Crank–Nicolson discretization and cosine-basis expansion for the control action, the dynamics are linear:

$$\boldsymbol{y}(t_{i+1}) = \boldsymbol{A}\,\boldsymbol{y}(t_i) + \boldsymbol{B}\,\boldsymbol{c}(t_i) + \boldsymbol{g}, \qquad \boldsymbol{y}(t_i)\in\mathbb{R}^{N_x},\ \ \boldsymbol{c}(t_i)\in\mathbb{R}^M. \tag{36}$$

We solve the convex QP

$$\min_{\{\boldsymbol{c}(t_i)\}_{i=k}^{k+N_p-1}} \sum_{i=k}^{k+N_p-1} \|\boldsymbol{y}(t_i) - \boldsymbol{y}_{\text{target}}\|_{\boldsymbol{T}_Q}^2 + \|\boldsymbol{c}(t_i)\|_{\boldsymbol{R}}^2 + \|\boldsymbol{y}(t_{k+N_p}) - \boldsymbol{y}_{\text{target}}\|_{\boldsymbol{Q}}^2 \tag{37}$$

$$\text{s.t. } (36)$$

$$c_{\min} \le \boldsymbol{c}(t_i) \le c_{\max}.$$

Here, $\boldsymbol{A} = M_L^{-1}M_R$, $\boldsymbol{B} = M_L^{-1}(\Delta t\,\boldsymbol{B}_c)$, $\boldsymbol{g} = M_L^{-1}(\Delta t\,\mathbf{s})$ (Neumann BCs), $\mathbf{B}_c$ stacks $\cos(j\pi x/X)$; (37) is solved with CVXPY/OSQP (Diamond & Boyd, 2016). Here $M_L$ and $M_R$ are the CN matrices of the system, obtained from the semi-discrete PDE operator $A_c$:

$$M_L = I - \frac{\Delta t}{2}A_c, \qquad M_R = I + \frac{\Delta t}{2}A_c. \tag{38}$$

This way one CN step writes as:

$$M_L\,\boldsymbol{y}(t_{k+1}) = M_R\,\boldsymbol{y}(t_k) + \Delta t\,\boldsymbol{B}_c\,\boldsymbol{c}(t_k) + \Delta t\,\mathbf{s}. \tag{39}$$

**Nonlinear MPC for the Burgers equation.** For viscous Burgers, the CN step is nonlinear due to convection term; we write one step as the implicit map

$$\boldsymbol{y}(t_{i+1}) = \mathcal{G}(\boldsymbol{y}(t_i), \boldsymbol{c}(t_i); \Delta t), \tag{40}$$

where $\mathcal{G}$ is the unique CN solution for Burgers with input $\boldsymbol{B}\boldsymbol{c}(t_i)$ (computed via `fsolve` on the CN residual and by imposing Dirichlet endpoints at each step).

$$\min_{\{\boldsymbol{c}(t_i)\}_{i=k}^{k+N_p-1}} \sum_{i=k}^{k+N_p-1} \|\boldsymbol{y}(t_i) - \boldsymbol{y}_{\text{target}}\|_{\boldsymbol{T}_Q}^2 + \|\boldsymbol{c}(t_i)\|_{\boldsymbol{R}}^2 + \|\boldsymbol{y}(t_{k+N_p}) - \boldsymbol{y}_{\text{target}}\|_{\boldsymbol{Q}}^2 \tag{41}$$

$$\text{s.t. } 40,\ \ c_{\min} \le \boldsymbol{c}(t_i) \le c_{\max}.$$

We use single shooting over the full-order model: propagate with (40), optimize with SLSQP(Kraft, 1994) under box bounds, apply $\mathbf{y}(t_k)$, advance one CN step, and iterate. Here $\mathbf{B}$ stacks $\sin((j+1)\pi x/X)$ shapes used in the implementation. Notably, $\mathbf{T}_Q, \mathbf{R}$, and $\mathbf{Q}$ represent the weight matrices, which correspond to the multipliers $\boldsymbol{\lambda}$ in our PDE-OP method.

## 9.6 SURROGATE-BASED TRAJECTORY OPTIMIZATION (SBTO)

We compare against a learning-based trajectory optimization baseline adapted from (Hwang et al., 2022), which we refer to as Surrogate-Based Trajectory Optimization (SBTO). While Hwang et al. (2022) introduced a reconstruction regularizer to prevent adversarial artifacts in high-dimensional control spaces, we omit this term in our implementation. This is because we parameterize the control inputs using low-dimensional smooth basis functions (as described in Section 4.1), which inherently restricts the search space and acts as an implicit regularizer against high-frequency noise. Given an initial state $y(0, x)$ and a target state $y_{\text{target}}(x)$, the goal is to find the optimal sequence of time-varying weights $\boldsymbol{c}(t)$ that minimizes a cost function over the entire time horizon. The optimization problem can be defined as:

$$\min_{\{\boldsymbol{c}(t_i)\}_{i=0}^{N_t-1}} \|\boldsymbol{y}(t_{N_t}) - \boldsymbol{y}_{\text{target}}\|_{\mathbf{Q}}^2 + \sum_{k=0}^{N_t-1} \left( \|\boldsymbol{y}(t_{k+1}) - \boldsymbol{y}_{\text{target}}\|_{\mathbf{T_Q}}^2 + \|\boldsymbol{c}(t_k)\|_{\mathbf{R}}^2 \right)$$

$$\text{(42)}$$

$$\text{s.t.} \quad \boldsymbol{y}(t_{k+1}) = \mathcal{Y}_\theta(\boldsymbol{y}(t_k), \boldsymbol{c}(t_k)), \quad \text{for } k = 0, \dots, N_t - 1$$

$$c_{\min} \le \boldsymbol{c}(t) \le c_{\max},$$

where $\mathbf{T}_Q, \mathbf{R}$, and $\mathbf{Q}$ represent the weight matrices, which correspond to the multipliers $\boldsymbol{\lambda}$ in our PDE-OP method. This method consists of two distinct phases. **In Phase 1** a neural operator, here denoted as $\mathcal{Y}_\theta$, is trained offline to learn a differentiable surrogate for the PDE dynamics, mapping the current state and control to the next state. **In Phase 2** the surrogate model's parameters $\theta$ are frozen. For each new problem instance at test-time, an open-loop trajectory optimization is performed to find the optimal time-varying weights $\boldsymbol{c}(t)$ as described in (42). This problem is solved using a gradient-based optimizer (i.e., Adam) by backpropagating through the full unrolled trajectory estimated by the surrogate $\mathcal{Y}_\theta$.

## 9.7 DEEPONET MPC

DeepONet MPC is a closed-loop, receding-horizon strategy that combines operator learning with MPC. We follow the **Standard DeepONet MPC** formulation established as a benchmark in (de Jong et al., 2025). Like SBTO, this method operates in two phases. **In Phase 1** the same differentiable surrogate $\mathcal{Y}_\theta$ is trained to learn the system dynamics. **In Phase 2**, however, instead of solving one large optimization problem as in SBTO, DeepONet MPC solves a new, smaller optimization problem at *each* time step $t_k$. Given the current measured state $\boldsymbol{y}(t_k)$, the controller plans a sequence of future actions $\boldsymbol{c}(t_k), \dots, \boldsymbol{c}(t_{k+N_p-1})$ over a shorter planning horizon of length $N_p$. The optimization problem solved at step $t_k$ is:

$$\min_{\{\boldsymbol{c}(t_{k+i})\}_{i=0}^{N_p-1}} \|\boldsymbol{y}(t_{k+N_p}) - \boldsymbol{y}_{\text{target}}\|_{\boldsymbol{T_Q}}^2$$

$$+ \sum_{i=0}^{N_p-1} \left( \|\boldsymbol{y}(t_{k+i+1}) - \boldsymbol{y}_{\text{target}}\|_{\boldsymbol{Q}}^2 + \|\boldsymbol{c}(t_{k+i})\|_{\boldsymbol{R}}^2 \right)$$

$$\text{(43)}$$

$$\text{s.t.} \quad \boldsymbol{y}(t_{k+i+1}) = \mathcal{Y}_\theta(\boldsymbol{y}(t_{k+i}), \boldsymbol{c}(t_{k+i})), \quad \text{for } i = 0, \dots, N_p - 1$$

$$c_{\min} \le \boldsymbol{c}(t_{k+i}) \le c_{\max},$$

where $\mathbf{T}_Q, \mathbf{R}$, and $\mathbf{Q}$ represent the weight matrices. After solving for the optimal sequence $\boldsymbol{c}(t_k), \dots, \boldsymbol{c}(t_{k+N_p-1})$, only the first control action, $\boldsymbol{c}(t_k)^*$, is applied to the system. The rest of the sequence is discarded, and this entire optimization process is repeated from the next time step $t_{k+1}$. This formulation is related to DeepONet MPC (de Jong et al., 2025), but differs in that we use a one-step DeepONet surrogate $\mathcal{Y}_\theta$ and obtain multi-step predictions by rolling it out recursively over the horizon, rather than using a multi-step DeepONet that predicts the full trajectory in a single forward pass.

**Choice of controller.** For the voltage optimization task, we use *Direct*, *SBTO* and *Adjoint-method for open loop*. For the heat equation, CN yields a linear PDE, so the finite-horizon problem is a convex QP, which can be solved with *Linear MPC*. For Burgers' equation, the dynamics are nonlinear

and CN produces an implicit nonlinear update, so we use *Nonlinear MPC*, with NMPC optimizing over an implicit CN step. For both the heat and burgers' tasks, we use *Adjoint-method for closed loop*, *SBTO* and *DeepONet MPC*.

## 9.8 EXPERIMENTAL SETTINGS

This section provides the mathematical models and corresponding description of the tasks considered in the main paper in Section 5.

### 9.8.1 VOLTAGE CONTROL OPTIMIZATION

The problem of regulating the voltage magnitude in a leaky electrical transmission line, subject to a reaction-diffusion PDE with Neumann conditions is described as follows:

$$\underset{u(x)}{\text{Minimize}} \quad \int_\Omega (y(T,x) - y_{\text{target}}(x))^2 dx + \gamma \int_\Omega u(x)^2 dx \tag{44a}$$

$$\text{s.t.} \quad \frac{\partial y}{\partial t} = D\frac{\partial^2 y}{\partial x^2} - \beta(y - y_{\text{ref}}(x)) + \alpha u(x), \quad \forall (x,t) \in \Omega \times [0,T] \tag{44b}$$

$$y(0,x) = y_0(x), \quad \forall x \in \Omega, \{\, t = 0\} \tag{44c}$$

$$\frac{\partial y}{\partial x}(t,x) = 0, \quad \forall x \in \partial\Omega, \ t \in [0,T] \tag{44d}$$

$$u_{\min} \leq u(x) \leq u_{\max}. \quad \forall x \in \Omega. \tag{44e}$$

Here, $y(t,x) \in \mathbb{R}$ represents the voltage profile across the space-time domain defined by the cartesian product between $\Omega$ and time horizon $[0,T]$. The system dynamics, expressed by (44b), model the physical phenomena of diffusion ($D\frac{\partial^2 y}{\partial x^2}$), voltage leakage towards a reference $y_{\text{ref}}(x)$ ($-\beta(y - y_{\text{ref}}(x))$) and the applied control action ($\alpha u(x)$). $D$, $\beta$, and $\alpha$ are all constants representing the diffusion rate, leakage rate, and control gain, respectively. The system is initialized from a state $y_0(x)$ according to (44c) and is constrained by zero-flux Neumann boundary conditions (44d), implying no voltage diffusion across the boundaries of the domain. The control decision $u(x)$ is a space-dependent function representing the externally applied voltage, constrained by actuator limits in (44e). The objective (44a) minimizes a combination of terminal cost and control energy.

**Datasets and methods.** In the system dynamics (44b), we set the diffusion rate $D = 0.1$, leakage rate $\beta = 1.0$, and control gain $\alpha = 2.0$, with a uniform reference voltage $y_{\text{ref}}(x) = 1.0$. The ground-truth dynamics are simulated using a second-order implicit Crank-Nicolson finite-difference method(Crank & Nicolson, 1947). To initialize the proposed time-discrete Neural Operator $\mathcal{Y}_\theta$, we generate a dataset by sampling a diverse set of control functions $u(x)$ from a zero-mean Gaussian Random Field (GRF) with a squared-exponential kernel, with mean $m(x)$ and covariance function $k_\ell(x_1, x_2)$ as:

$$m(x) = 0, \qquad k_\ell(x_1, x_2) = \sigma^2 \exp\left(-\frac{\|x_1 - x_2\|_2^2}{2\ell^2}\right), \tag{45}$$

where $\ell > 0$ is the length–scale and $\sigma^2 > 0$ the marginal variance. The corresponding solution $y(t,x)$ is computed using a numerical solver, and then split into 80% training, 10% validation, and 10% test sets.

PDE-OP's model $\mathcal{Y}_\theta$ estimates the voltage profile $y(t,x)$; for $t = T$ this prediction yield $y(T,x)$, which is compared against the target profile $y_{\text{target}}(x)$. PDE-OP's surrogate controller $\mathcal{U}_\omega$ is trained to produce optimal control action $\hat{u}(x)$ while minimizing objective (44a). Our approach is benchmarked against Direct, Adjoint-Method and SBTO, which are described in Appendix 9.2-9.6. Notably, SBTO is defined for closed-loop control problems, here we redefine it for open-loop structure, for the sake of simplicity we present the closed-loop version in Appendix 9.6. At inference time, PDE-OP's surrogate controller $\mathcal{U}_\omega$ is tested on three unseen target profiles of different level of complexities: constant ($y_{\text{target}}(x) = p_1$), linear ramp ($y_{\text{target}}(x) = p_1 x + p_2$), and sine wave ($y_{\text{target}}(x) = p_1 + p_2(\sin f_0 x + p_3)$). For each target, the control $\hat{u}(x)$ is generated in a single forward pass.

### 9.8.2 OPTIMAL CONTROL OF THE 1D HEAT EQUATION

This task involves controlling the evolution of temperature in a one-dimensional domain described by a reaction–diffusion PDE. Given an initial profile, the controller aims to reach a prescribed target state with minimal control effort. The problem is formulated as:

$$\underset{\boldsymbol{c}(t)_{t\in[0,T]}}{\text{Minimize}} \quad \mathcal{L} = \mathcal{L}_{\text{terminal}} + \lambda\mathcal{L}_{\text{running}} + \gamma\mathcal{L}_{\text{effort}} \tag{46a}$$

$$\text{s.t.} \quad \frac{\partial y}{\partial t} = D\frac{\partial^2 y}{\partial x^2} - \beta(y - y_{\text{ref}}(x)) + \alpha u(t,x), \quad \forall(x,t)\in\Omega\times[0,T] \tag{46b}$$

$$u(t,x) = \boldsymbol{c}(t)^\top\boldsymbol{\phi}(x), \qquad\qquad \forall(x,t)\in\Omega\times[0,T] \tag{46c}$$

$$y(0,x) = y_0(x), \qquad\qquad \forall x\in\Omega \tag{46d}$$

$$\frac{\partial y}{\partial x}(t,x) = 0, \qquad\qquad \forall x\in\partial\Omega,\ t\in[0,T] \tag{46e}$$

$$c_{\min} \le \boldsymbol{c}(t) \le c_{\max}, \qquad\qquad \forall\,t\in[0,T]. \tag{46f}$$

The loss components in (46a) defines as follows.

$$\mathcal{L}_{\text{terminal}} := \int_\Omega |y(T,x) - y_{\text{target}}(x)|^2\ dx, \tag{47a}$$

$$\mathcal{L}_{\text{running}} := \int_0^T\int_\Omega |y(t,x) - y_{\text{target}}(x)|^2\ dxdt, \tag{47b}$$

$$\mathcal{L}_{\text{effort}} := \int_0^T ||\boldsymbol{c}(t)||_2^2\ dt. \tag{47c}$$

Here $y(t,x)$ represents the temperature profile. The objective (46a) balances three terms: a terminal loss for final state accuracy, a running loss for temperature tracking, and an effort loss for control efficiency, described by (47a)–(47c).

**Datasets and methods.** The solutions of (46b) are generated using a Crank-Nicolson method, which deal with time-varying source term by averaging the control input over each time step. To train our models, we generate a dataset of diverse trajectories. First, smooth, time-varying control weights $(\boldsymbol{c}(t))$ are generated by sampling each of the $M$ weight trajectories from a GRF over the time domain. The full spatial-temporal control field $u(t,x)$ is then reconstructed from these weights constrained by (46f) using a set of fixed basis functions, as defined in (46c). For each generated control field, the full PDE is solved to generate a ground-truth temperature evolution $y(t,x)$. Each complete simulation yields a training sample, consisting of the control input weight trajectory $\{\boldsymbol{c}(t_k)\}$, and the corresponding state trajectory $\{y(t_k,x)\}$ evaluated at a set of fixed sensor locations. The dataset is then split into 80% training, 10% validation and 10% test.

Our framework uses two neural networks. First discrete-time neural operator, $\mathcal{Y}_\theta$, is trained as a surrogate model described in Section (4.2). It act as a one-step time-advancement operator, learning the mapping $(y(t_k,x), \boldsymbol{c}(t_k)) \to y(t_{k+1},x)$. Second, PDE-OP's surrogate controller $\mathcal{U}_\omega$ is a recurrent neural network that acts as sequential decision-maker. At each time step $t_k$, it takes the current state $y(t_k,x)$ and the final target $y_{\text{target}}(x)$ to produce the control weights $(\boldsymbol{c}(t_k))$. The controller is trained end-to-end by unrolling the trajectory using the surrogate $\mathcal{Y}_\theta$. We benchmark our approach against a highly-tuned Adjoint-Method, Linear MPC(LMPC), SBTO, and DeepONet MPC. Further implementation details are in Appendix 9.4-9.7.

At inference, we evaluate PDE-OP model on three unseen target profiles: constant($y_{\text{target}}(x) = p_1$), linear ramp($y_{\text{target}}(x) = p_1x + p_2$), sine wave($y_{\text{target}}(x) = p_1 + p_2(\sin f_0x + p_3)$), step(discontinuous), high-frequency($y_{\text{target}}(x) = p_1\sin(f_1x) + p_2\sin(f_2x)$), and complex-gaussian (mixed of gaussian profiles).

### 9.8.3 OPTIMAL CONTROL OF THE 2D HEAT EQUATION

To consider a more realistic and complex scenario, we extend the temperature control task introduced in the previous subsection to a two-dimensional spatial domain $\Omega = [0, L_x]\times[0, L_y] \subset \mathbb{R}^2$. Similar

to the 1D case, the controller seeks to drive the system from an initial temperature profile to a target state using minimal control effort. The control input is distributed spatially via a set of 2D basis functions. The optimization problem is formulated as follows:

$$\underset{\boldsymbol{c}(t)_{t\in[0,T]}}{\text{Minimize}} \quad \mathcal{L} = \mathcal{L}_{\text{terminal}} + \lambda \mathcal{L}_{\text{running}} + \gamma \mathcal{L}_{\text{effort}} \tag{48a}$$

$$\text{s.t.} \quad \frac{\partial z}{\partial t} = D\left(\frac{\partial^2 z}{\partial x^2} + \frac{\partial^2 z}{\partial y^2}\right) - \beta(z - z_{\text{ref}}) + \alpha u(t,\mathbf{x}), \quad \forall (\mathbf{x},t) \in \Omega \times [0,T] \tag{48b}$$

$$u(t,\mathbf{x}) = \boldsymbol{c}(t)^\top \boldsymbol{\Phi}(\mathbf{x}), \qquad\qquad\qquad \forall (\mathbf{x},t) \in \Omega \times [0,T] \tag{48c}$$

$$z(0,\mathbf{x}) = z_0(\mathbf{x}), \qquad\qquad\qquad\qquad \forall \mathbf{x} \in \Omega \tag{48d}$$

$$\nabla z \cdot \mathbf{n} = 0, \qquad\qquad\qquad\qquad\quad \forall \mathbf{x} \in \partial\Omega,\ t \in [0,T] \tag{48e}$$

$$c_{\min} \le \boldsymbol{c}(t) \le c_{\max}, \qquad\qquad\qquad \forall\, t \in [0,T]. \tag{48f}$$

Here, $\mathbf{x} = (x,y)$ denotes the spatial coordinates. The boundary condition (48e) represents homogeneous Neumann conditions (which correspond to insulated boundaries), where $\mathbf{n}$ is the outward normal vector. The control input $u(t,\mathbf{x})$ is represented as a linear combination of $M$ basis functions with basis weights $\boldsymbol{c}(t) \in \mathbb{R}^M$, where $M = N_{b,x} \times N_{b,y}$. The spatial basis $\boldsymbol{\Phi}(\mathbf{x})$ is defined as the product of cosine functions: $\phi_{ij}(x,y) = \cos(\frac{i\pi x}{L_x})\cos(\frac{j\pi y}{L_y})$.

The loss components are defined over the 2D domain:

$$\mathcal{L}_{\text{terminal}} := \int_\Omega |z(T,\mathbf{x}) - z_{\text{target}}(\mathbf{x})|^2\ d\mathbf{x}, \tag{49a}$$

$$\mathcal{L}_{\text{running}} := \int_0^T \int_\Omega |z(t,\mathbf{x}) - z_{\text{target}}(\mathbf{x})|^2\ d\mathbf{x}dt, \tag{49b}$$

$$\mathcal{L}_{\text{effort}} := \int_0^T ||\boldsymbol{c}(t)||_2^2\ dt. \tag{49c}$$

**Datasets and methods.** The system in equation (48b) is discretized on a tensor-product grid using a 2D Crank–Nicolson scheme. To efficiently handle spatial derivatives, the discrete Laplacian is constructed via the Kronecker sum of 1D operators. To generate the training dataset we sample diverse control trajectories where the weights $\boldsymbol{c}(t)$ are drawn from a GRF with varying temporal length scales $l$. The resulting control field $u(t,\mathbf{x})$ is computed via Equation (48c), and the PDE is solved to obtain the corresponding temperature evolution $z(t,\mathbf{x})$. The solution is then interpolated onto a fixed sensor grid of size $N_{s,x} \times N_{s,y}$ and the spatial fields are flattened into vectors of length $N_{s,x}N_{s,y}$ to be used as inputs for the neural operator. Each training sample consists of a control weight trajectory $\boldsymbol{c}(t)$ and the corresponding flattened state trajectory $z(t,\mathbf{x})$. The dataset is split into 80% training, 10% validation, and 10% test sets.

Following the 1D setup, we use two neural networks: a discrete-time neural operator $\mathcal{Y}_\theta$ adapted for 2D inputs (via a DeepONet with 2D trunk input), and a recurrent surrogate controller $\mathcal{U}_\omega$ that maps the flattened sensor data to control weight outputs for the 2D basis functions. We benchmark our approach against the Adjoint-Method, Linear MPC(LMPC), SBTO and DeepONet MPC. A description of each of these these methods can be found in Appendix 9.4-9.7.

At test time, we evaluate the PDE-OP model on three unseen target profiles: a *Gaussian peak* ($z_{\text{target}}(\mathbf{x}) = A\exp(-\frac{(x-c_x)^2}{2w_x^2} - \frac{(y-c_y)^2}{2w_y^2})$), a *Sine wave* ($z_{\text{target}}(\mathbf{x}) = \mu + A\sin(2\pi x/L_x)\sin(2\pi y/L_y)$), and a *Complex Gaussian* profile generated by a stochastic superposition of five randomly sampled Gaussian modes.

### 9.8.4 OPTIMAL CONTROL OF 1D BURGERS' EQUATION

To test the ability of PDE-OP to handle nonlinear dynamics, we address the control of the 1D viscous Burgers' equation with Dirichlet boundary condition. The control input and external force $u(t,x)$, is parameterized identically to the heat equation task, using time-varying weights $(\boldsymbol{c}(t_k))$ for a set of basis functions. The optimization goal is to find the optimal trajectory of these weights subject to

Burger's equation (50b), initial (50d) and Dirchlet's boundary condition (50e) and box constraints (50f).

$$\underset{\boldsymbol{c}(t)_{t \in [0,T]}}{\text{Minimize}} \quad \mathcal{L} = \mathcal{L}_{\text{terminal}} + \lambda \mathcal{L}_{\text{running}} + \gamma \mathcal{L}_{\text{effort}} \tag{50a}$$

$$\text{s.t.} \quad \frac{\partial y}{\partial t} + y \frac{\partial y}{\partial x} = \nu \frac{\partial^2 y}{\partial x^2} + u(t,x), \qquad \forall (x,t) \in \Omega \times [0,T] \tag{50b}$$

$$u(t,x) = \boldsymbol{c}(t)^\top \boldsymbol{\phi}(x), \qquad \forall (x,t) \in \Omega \times [0,T] \tag{50c}$$

$$y(0,x) = y_0(x), \qquad \forall x \in \Omega \tag{50d}$$

$$y(t,x) = 0, \qquad \forall x \in \partial\Omega,\ t \in [0,T] \tag{50e}$$

$$c_{\min} \le \boldsymbol{c}(t) \le c_{\max}, \qquad \forall\, t \in [0,T]. \tag{50f}$$

Here $y(t,x)$ is the velocity field and $\nu$ represents the viscosity parameter. The PDE in (50b) contains convective nonlinearity $(y\frac{\partial y}{\partial x})$ which has a significant control challenge not present in the linear heat equation. The objective retrains the same structure, balancing final state accuracy, path tracking, and control efficiency.

**Datasets and methods.** The ground-truth dynamics for the nonlinear Burgers' equation are simulated using a high-fidelity numerical solver. The dataset generation process mirrors that of the heat equation: time-varying weights $\boldsymbol{c}(t)$ are sampled from a GRF, and the force field $u(t,x)$ is reconstructed using spatial basis functions, and the PDE is solved to generate trajectories. Due to the system's high nonlinearity, we benchmark our method against a much more general and computationally expensive Nonlinear MPC (NMPC) and Adjoint-Sensitivity Method. Further implementation details are in Appendix 9.4 - 9.5.

At inference, we test the PDE-OP model on three unseen target profiles: constant $(y_{\text{target}}(x) = p_1)$, parabolic $(y_{\text{target}}(x) = p_1 x (1 - x))$ and sine wave $(y_{\text{target}}(x) = p_1(\sin f_0 x + p_2))$. We don't use a a linear target profile, because of the Dirichlet BCs.

### 9.8.5 OPTIMAL CONTROL OF 1D BURGERS' EQUATION WITH SPATIALLY-VARYING VISCOSITY

To evaluate the ability of PDE-OP to generalize across heterogeneous physical systems, we address the control of the 1D viscous Burgers' equation with a spatially-varying viscosity parameter $\nu(x)$. This introduces a parametric complexity where the controller must adapt to the specific material properties of the domain. The optimization goal remains to find the optimal trajectory of basis function weights $\boldsymbol{c}(t)$ subject to the modified dynamics and Dirichlet boundary conditions:

$$\underset{\boldsymbol{c}(t)_{t \in [0,T]}}{\text{Minimize}} \quad \mathcal{L} = \mathcal{L}_{\text{terminal}} + \lambda \mathcal{L}_{\text{running}} + \gamma \mathcal{L}_{\text{effort}} \tag{51a}$$

$$\text{s.t.} \quad \frac{\partial y}{\partial t} + y \frac{\partial y}{\partial x} = \frac{\partial}{\partial x}\left(\nu(x)\frac{\partial y}{\partial x}\right) + u(t,x), \qquad \forall (x,t) \in \Omega \times [0,T] \tag{51b}$$

$$u(t,x) = \boldsymbol{c}(t)^\top \boldsymbol{\phi}(x), \qquad \forall (x,t) \in \Omega \times [0,T] \tag{51c}$$

$$y(0,x) = y_0(x), \qquad \forall x \in \Omega \tag{51d}$$

$$y(t,x) = 0, \qquad \forall x \in \partial\Omega,\ t \in [0,T] \tag{51e}$$

$$c_{\min} \le \boldsymbol{c}(t) \le c_{\max}, \qquad \forall\, t \in [0,T]. \tag{51f}$$

Here $y(t,x)$ is the velocity field. The diffusive term in (51b) is now governed by $\nu(x)$, which varies across the domain $\Omega$. This setup presents a dual challenge: the controller must handle the convective nonlinearity $(y\partial_x y)$ while simultaneously compensating for regions of high and low dissipation defined by $\nu(x)$.

**Datasets and methods.** The ground-truth dynamics are generated using a numerical Crank-Nicolson solveron a staggered grid, which ensures accurate flux conservation under varying viscosity conditions. To construct the dataset, we randomize both the control inputs and the physical parameters: control weights $\boldsymbol{c}(t)$ are sampled from a temporal GRF, while the spatial viscosity profile $\nu(x)$ is drawn from a spatial GRF and mapped to a bounded positive range $[\nu_{\min}, \nu_{\max}]$. To

account for this variability, both the neural operator and the surrogate controller are designed to be *parameter-conditioned*, which is obtained by concatenating the discretized viscosity profile $\nu$ to their respective inputs.

We benchmark PDE-OP against the classical method Adjoint-Sensitivity Method, which assumes full knowledge of the system dynamics. Additionally, we compare against two learning-based baselines: SBTO, which optimizes control inputs using a pretrained operator model, and DeepONet MPC, which uses the surrogate model for look-ahead receding-horizon planning.

At inference time, we evaluate the models on six target profiles to asses generalization performance: (i) *Sine Mixture* ($y_{\text{target}} \propto 0.4\sin(2\pi x) + 0.4\sin(4\pi x)$); (ii) *Parabola* ($y_{\text{target}} \propto x(L-x)$); (iii) *Step Function* ($y_{\text{target}} = 0.6$ for $x \in [0.3L, 0.7L]$); (iv) *High-Frequency* ($y_{\text{target}} \propto 0.5\sin(2\pi x) + 0.2\sin(12\pi x)$); and (v) *Complex Gaussian*, a stochastic superposition of 5 random Gaussian modes. All evaluations are conducted under randomized viscosity profiles $\nu(x)$ to test the robustness of the controllers to variations in the physical environment.

### 9.8.6   OPTIMAL CONTROL OF THE 1D HEAT EQUATION (BATTERY THERMAL MANAGEMENT)

Here the task consists of controlling the thermal profile of a Lithium-Ion battery module, modeled as a 1D heat equation as described in Section 9.8.2. This task is particularly important in high-performance energy systems, as maintaining an optimal temperature distribution is critical for safety and efficiency.

**Dataset and Physical Parameters.**

To reflect real-world energy system behavior, we adopt the PDE parameters from Liu et al. (2019). Specifically, the thermal diffusivity D is derived from the material properties of a commercial Lithium-Ion polymer battery cell (EiG ePLB-C020), as experimentally validated in that study. Given the density $\rho = 2092\,\text{kg/m}^3$, specific heat capacity $C_p = 678\,\text{J/(kg} \cdot \text{K)}$, and thermal conductivity $k \approx 18.4\,\text{W/(m} \cdot \text{K)}$, the physical thermal diffusivity $D_{\text{phys}}$ is calculated as:

$$D_{\text{phys}} = \frac{k}{\rho \cdot C_p} = \frac{18.4}{2092 \cdot 678} \approx 1.3 \times 10^{-5}\,\text{m}^2/\text{s} = 0.13\,\text{cm}^2/\text{s}. \tag{52}$$

The PDE solutions used to initialize the neural surrogate of PDE-OP and the neural baselines are generated using a Crank-Nicolson method. In particular, we generate a dataset of diverse trajectories by sampling smooth, time-varying control weights $c(t)$ from a Gaussian Random Field. The dataset is divided into 80% training, 10% validation, and 10% test. We compare our approach against the Adjoint-Method, Linear MPC, SBTO, and DeepONet MPC.

### 9.9   EXPERIMENTS

In this section, we first evaluate the ability of the time-discrete neural operator introduced in Section 4.2 in capturing the system dynamics once initialized as described in Section 4.2.
Secondly, we evaluate the runtime and terminal tracking error (MSE) achieved by each method on different target profiles, and report these metrics in Tables 2- 8. All methods use the same spatial/temporal discretization and solver settings as described in Appendix 9.12.

### 9.9.1   CAPTURING THE SYSTEM DYNAMICS WITH PDE-OP'S TIME-DISCRETE NEURAL OPERATOR

Figure 6 shows the predicted voltage (central sub-figure of Fig. 6) of PDE-OP's dynamic component $\mathcal{Y}_\theta$ at test time, which is compared to the solution (left sub-figure) obtained with a numerical PDE-solver. The figure shows that the predicted voltage closely matches the numerical PDE-solver solution, with an absolute error (right sub-figure) on the order of $10^{-2}$ across the space–time domain. Figure 7 shows the temperature (central sub-figure of Fig. 7) estimated by PDE-OP's dynamic component $\mathcal{Y}_\theta$ at test time, which is compared to the solution (left sub-figure) obtained with a numerical PDE-solver. The figure shows that also in this setting the predicted temperature aligns closely with the numerical PDE-solver solution, with an absolute error on the order of $10^{-3}$ across the space–time

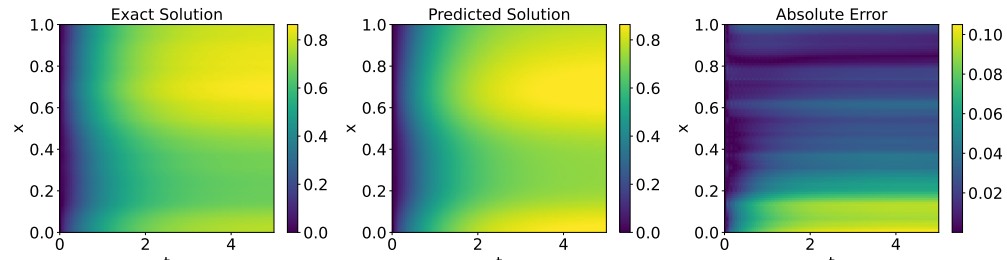

Figure 6: Voltage Equation. PDE-OP's dynamic predictor $\mathcal{Y}_\theta$ solution estimate at test time: numerical solution (left), model prediction (center), and absolute error (right).

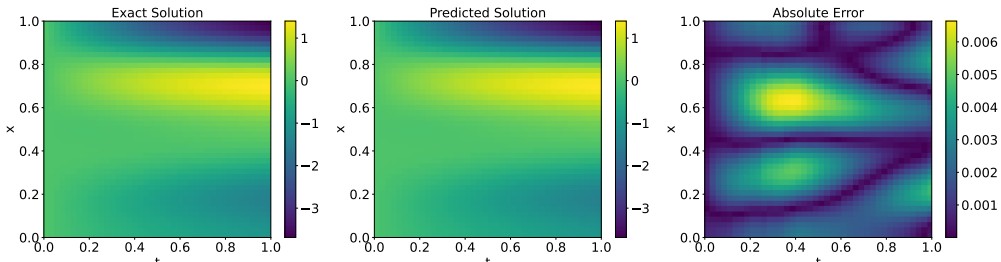

Figure 7: 1D Heat Equation. PDE-OP's dynamic predictor $\mathcal{Y}_\theta$ solution estimate at test time: numerical solution (left), model prediction (center), and absolute error (right).

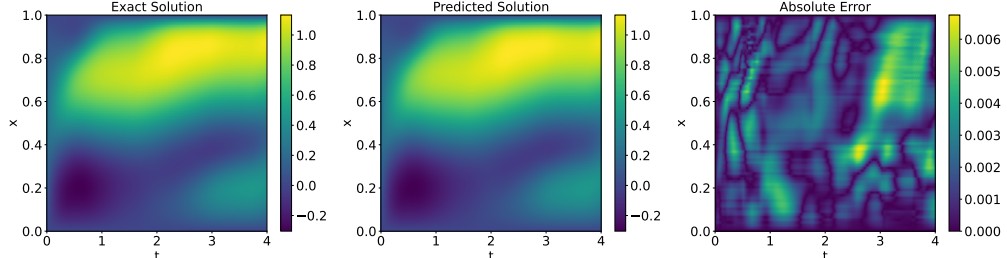

Figure 8: 1D Burgers' Equation. PDE-OP's dynamic predictor $\mathcal{Y}_\theta$ solution estimate at test time: numerical solution (left), model prediction (center), and absolute error (right).

domain. Figure 8 shows the system dynamics (central sub-figure of Fig. 8) estimated by PDE-OP's dynamic component $\mathcal{Y}_\theta$ at test time, which is compared to the solution (left sub-figure) obtained with a numerical PDE-solver. The figure shows that the predicted temperature aligns closely with the numerical PDE-solver solution; with absolute error on the order of $10^{-3}$ across the space–time domain, demonstrating the ability of PDE-OP's dynamic predictor to accurately capture Burgers' dynamics.

### 9.9.2 VOLTAGE CONTROL OPTIMIZATION

We show other targets $y_{\text{target}}(x) = p_1 x + p_2$: (i) *constant*: $(p_1, p_2) = (0, 1)$ so $y_{\text{target}}(x) = 1$; (ii) *ramp*: $(p_1, p_2) = (1, 0.5)$ so $y_{\text{target}}(x) = x + 0.5$. (Equivalently, $y_{\text{target}}(x) = p_1 x + p_2$ with $(p_1, p_2) = (0, 1)$ and $(1, 0.5)$, respectively.) Since a ramp target profile is infeasible under zero-flux (Neumann) boundaries, all methods can only produce approximations.

**Results.** *Constant target $y_{target}(x) = 1$.* The Adjoint method attains the highest accuracy (MSE $\mathbf{2.82 \times 10^{-12}}$) in $0.049$ s. SBTO achieves an MSE of $3.73 \times 10^{-9}$ in $0.418$ s, while the Direct method is significantly slower ($2.422$ s). PDE-OP is the fastest at $\mathbf{0.037}$ s—approximately $11\times$ faster than SBTO and $1.32\times$ faster than the Adjoint method—with a final MSE of $1.53 \times 10^{-6}$.

Table 2: Voltage Optimization Task. For each method and target profile, we report runtime (s) and MSE.

| Methods | Constant | | Ramp | |
|---|---|---|---|---|
| | **Runtime** | **Accuracy** | **Runtime** | **Accuracy** |
| Direct (Finite Diff.) | 2.422s | 4.6498e-11 | 12.810s | 0.00005 |
| Adjoint-Method | 0.049s | **2.8213e-12** | 0.464s | **0.00004** |
| SBTO | 0.418s | 3.7279e-09 | 0.422s | 0.00038 |
| **PDE-OP (ours)** | **0.037s** | 1.5275e-6 | **0.050s** | 0.00007 |

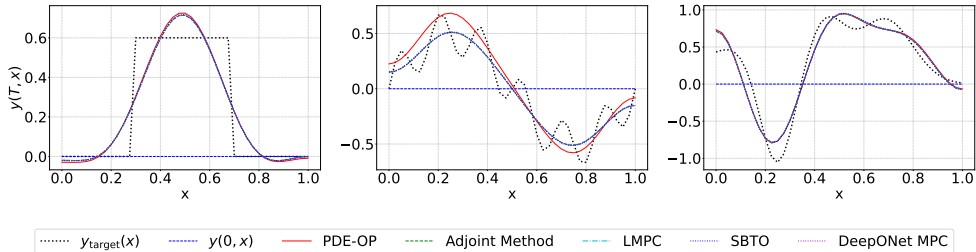

$y_{target}(x)$ ........  $y(0,x)$ - - -  PDE-OP ——  Adjoint Method - - -  LMPC - - -  SBTO ........  DeepONet MPC ........

Figure 9: 1D Heat Equation. Comparison of PDE-OP and baselines for advanced target profiles: *Step* (discontinuous)(left), *High-Frequency* ($y_{target}(x) = 0.5 \sin(2\pi x) + 0.2 \sin(12\pi x)$)(center), and *Complex Gaussian* (stochastic multi-modal profile)(right).

*Ramp target $y_{target}(x) = x + 0.5$.* The Adjoint method yields the lowest error ($\mathbf{4.0 \times 10^{-5}}$), running in $0.464$ s. SBTO trails with an MSE of $3.8 \times 10^{-4}$ in $0.422$ s. PDE-OP runs in $\mathbf{0.050}$ s, about $9.3\times$ faster than the Adjoint method and $8.4\times$ faster than SBTO, producing an MSE of $7.0 \times 10^{-5}$, which is on the same order of magnitude as the optimal baselines.

### 9.9.3 OPTIMAL CONTROL OF THE 1D HEAT EQUATION

For the 1D heat equation, we evaluate the control performance on two categories of target profiles $y_{target}(x)$. **Standard Targets:** (i) *Constant* ($y_{target} = 1$); (ii) *Ramp* ($y_{target} = x + 0.5$); and (iii) *Sine* ($y_{target} \propto 0.6 + 0.3 \sin(2\pi x)$). **Advanced Targets:** (i) *Step* (discontinuous profile); (ii) *High-Frequency* (superposition of low and high frequency sine waves); and (iii) *Complex Gaussian* (stochastic mixture of Gaussian modes). We compare the proposed PDE-OP model against classical baselines (Adjoint Method, LMPC) and neural optimization baselines (SBTO, DeepONet MPC) as shown in Figures 4 and 9.

Table 3: 1D Heat Optimization Task. For each method and standard target profile, we report runtime (s) and MSE.

| Method | Constant | | Ramp | |
|---|---|---|---|---|
| | **Runtime** | **Accuracy** | **Runtime** | **Accuracy** |
| LMPC | 0.443s | **2.3289e-14** | 0.562s | **0.00008** |
| Adjoint Method | 0.485s | 6.6612e-11 | 0.552s | **0.00008** |
| SBTO | 15.647s | 4.1923e-6 | 15.596s | 0.00009 |
| DeepONet MPC | 144.306s | 1.3723e-10 | 143.960s | **0.00008** |
| **PDE-OP (Ours)** | **0.223s** | 1.5210e-6 | **0.132s** | 0.00009 |

**Results.** *Standard Targets (Table 3).* When controlling the temperature profile to a constant target, LMPC achieves machine precision ($2.33 \times 10^{-14}$), whereas PDE-OP produces an MSE of $1.52 \times 10^{-6}$. However, for the ramp target, PDE-OP achieves accuracy ($9.0 \times 10^{-5}$) on the same order of magnitude as the numerical optimizers ($8.0 \times 10^{-5}$) while running significantly faster. Specifically,

Table 4: 1D Heat Optimization Task. For each method and advanced target profile, we report runtime (s) and MSE.

| Method | Step | | High Frequency | | Complex Gaussian | |
|---|---|---|---|---|---|---|
| | **Runtime** | **Accuracy** | **Runtime** | **Accuracy** | **Runtime** | **Accuracy** |
| LMPC | 0.315s | **0.01359** | 0.332s | **0.01868** | 0.564s | **0.02405** |
| Adjoint Method | 0.423s | **0.01359** | 0.416s | **0.01868** | 0.619s | **0.02405** |
| SBTO | 15.595s | **0.01359** | 15.556s | 0.01892 | 15.555s | 0.02408 |
| DeepONet MPC | 144.149s | **0.01359** | 144.198s | **0.01868** | 144.245s | **0.02405** |
| **PDE-OP (Ours)** | **0.033s** | 0.01364 | **0.033s** | 0.02725 | **0.033s** | 0.02415 |

the inference time of PDE-OP is only $0.132$ seconds for the ramp profile, compared to $\approx 0.5$ seconds for LMPC and Adjoint and $> 140$ seconds for DeepONet MPC.

*Advanced Targets (Table 4).* For complex profiles, the performance gap narrows in accuracy but widens in efficiency. On the Step and Complex Gaussian targets, PDE-OP achieves an MSE of $1.36 \times 10^{-2}$ and $2.42 \times 10^{-2}$ respectively, effectively matching the optimal solutions found by LMPC and Adjoint methods. *Notably, PDE-OP computes these controls in $0.03$ seconds, representing a $10 \times$ speedup over LMPC/Adjoint, a $450 \times$ speedup over SBTO, and a $4000 \times$ speedup over DeepONet MPC!*

*Takeaway.* For this experiment, we observe that classical control-based methods achieve the highest precision when targeting simple target profiles. However, PDE-OP offers real-time inference with negligible accuracy loss on complex, spatially heterogeneous targets, demonstrating its suitability for time-critical applications.

### 9.9.4 OPTIMAL CONTROL OF THE 2D HEAT EQUATION

For this task, we evaluate PDE-OP and the baseline methods on two distinct target profiles defined in the spatial domain $\Omega = [0, L_x] \times [0, L_y]$: (i) *Gaussian Peak*, a single centered mode $z_{\text{target}}(\mathbf{x}) = 1.5 \exp\left(-\frac{(x-c_x)^2}{2w_x^2} - \frac{(y-c_y)^2}{2w_y^2}\right)$; (ii) *Sine Wave*, a spatially periodic profile $z_{\text{target}}(\mathbf{x}) = 0.8 + 0.7 \sin(2\pi x/L_x) \sin(2\pi y/L_y)$. We compare the proposed PDE-OP model against the Adjoint-Sensitivity Method, LMPC and two neural baselines: Surrogate-Based Test-Time Optimization (SBTO) and DeepONet MPC as shown in Figure 10 and in Table 5.

Table 5: 2D Heat Equation Task. We report runtime (s) and MSE for both target types.

| Methods | Gaussian Peak | | Sine Wave | |
|---|---|---|---|---|
| | **Runtime** | **Accuracy** | **Runtime** | **Accuracy** |
| LMPC | 68.432s | **0.00066** | 92.879s | **0.00712** |
| Adjoint-Method | 8.621s | **0.00066** | 8.475s | **0.00712** |
| SBTO | 13.321s | 0.00071 | 13.102s | 0.00725 |
| DeepONet MPC | 15.291s | **0.00066** | 15.262s | 0.01900 |
| **PDE-OP (ours)** | **0.032s** | 0.00072 | **0.031s** | 0.00719 |

**Results.** *Gaussian Peak.* Classical methods (LMPC and Adjoint) achieve the lowest MSE ($\mathbf{6.62 \times 10^{-4}}$), though LMPC is computationally prohibitive for real-time applications (runs in 68.4 s). The neural baselines, SBTO and DeepONet MPC, achieve competitive accuracy but require iterative optimization at test time ($\approx 14$ s). PDE-OP matches their accuracy level ($7.25 \times 10^{-4}$) but runs in only $0.03$ s—approximately $444 \times$ faster than SBTO and $287 \times$ faster than the Adjoint method.

*Sine Wave.* The performance trend observed for the Gaussian target remain consistent also for the sine wave profile: Adjoint and LMPC generates a MSE of ($7.12 \times 10^{-3}$), while DeepONet

MPC struggles with this target ($1.90 \times 10^{-2}$), running in (15.2 s). SBTO matches the lowest error ($7.25 \times 10^{-3}$) but runs in 13.1 s. PDE-OP provides the best trade-off, producing a MSE of $7.19 \times 10^{-3}$, comparable to the best baselines, but with a negligible inference time of approximately **0.03**, confirming the speedups improvements obtained in case of Gaussian Peak target.

*Takeaway.* For this experiment, iterative neural baselines (SBTO, DeepONet MPC) suffer from high inference latency due to repeated surrogate evaluations. PDE-OP eliminates this bottleneck by producing control decisions in milliseconds, with accuracy comparable to that of the best baseline methods.

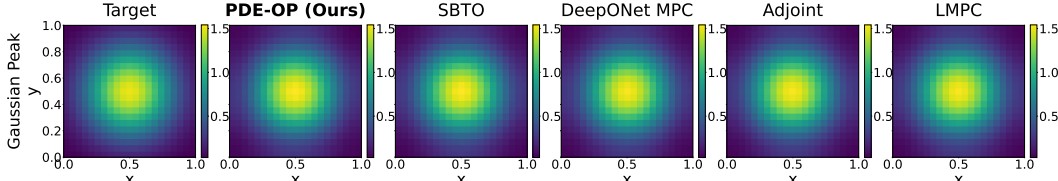

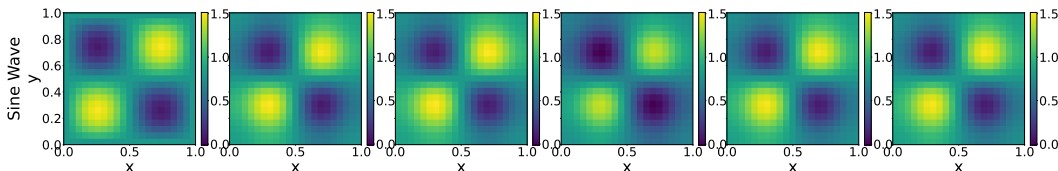

Figure 10: 2D Heat Equation. Comparison of PDE-OP and methods for target profiles: *Gaussian Peak(up), Sine Wave(down).*

### 9.9.5 OPTIMAL CONTROL OF THE 1D BURGERS' EQUATION

For the 1D viscous Burgers equation with Dirichlet boundaries $y(t, 0) = y(t, L) = 0$, admissible steady targets must satisfy these conditions. We therefore consider (i) a *zero* (constant) target $y_{\text{target}}(x) = 0$, and (ii) a *parabolic* target $y_{\text{target}}(x) = p_1 x(1 - x)$ (we use $p_1 = 2.0$ in experiments). We compare our method with NMPC and time-varying Adjoint baseline.

Table 6: Burger Optimization Task. For each method and target profile we report runtime (s) and MSE.

| Methods | Constant | | Parabola | |
|---|---|---|---|---|
| | **Runtime** | **Accuracy** | **Runtime** | **Accuracy** |
| NMPC | 132.243s | **5.6760e-14** | 671.033s | 0.00006 |
| Adjoint-Method | 21.487s | 5.8290e-11 | 122.116s | 0.00013 |
| SBTO | 59.179s | 6.6334e-6 | 58.982s | 7.5214e-6 |
| DeepONet MPC | 98.296s | 3.3844e-6 | 98.208s | **1.8547e-6** |
| **PDE-OP (ours)** | **0.059s** | 0.00002 | **0.073s** | 0.00031 |

**Results.** *Constant target* $y_{\text{target}}(x) = 0$. NMPC reaches nearly machine precision (MSE $\mathbf{5.68 \times 10^{-14}}$) but is slow at 132.243 s. The learning-based baselines, SBTO and DeepONet MPC, achieve MSEs of $6.63 \times 10^{-6}$ and $3.38 \times 10^{-6}$ respectively, but require significantly more time (59.179 s and 98.296 s) than the Adjoint baseline (21.487 s). PDE-OP is the fastest at **0.059** s (about $2.24 \times 10^3 \times$ vs. NMPC and $3.64 \times 10^2 \times$ vs. Adjoint) with a comparable error of $2.0 \times 10^{-5}$. *Parabolic target* $y_{\text{target}}(x) = 2x(1 - x)$. DeepONet MPC attains the lowest error (MSE $\mathbf{1.85 \times 10^{-6}}$), followed by SBTO ($7.52 \times 10^{-6}$), both outperforming NMPC ($6.0 \times 10^{-5}$).

However, their runtimes remain high at $98.208$ s and $58.982$ s, respectively. PDE-OP runs in **0.073** s—about $9.19 \times 10^3 \times$ faster than NMPC and $1.34 \times 10^3 \times$ faster than DeepONet MPC—with an MSE of $3.1 \times 10^{-4}$.

*Takeaway.* For nonlinear Burgers dynamics with Dirichlet BCs, while DeepONet MPC and SBTO can provide superior accuracy on complex profiles, PDE-OP delivers three to four orders of magnitude lower runtime with modest accuracy loss, making it uniquely suitable for real-time applications.

### 9.9.6 OPTIMAL CONTROL OF THE 1D BURGERS' EQUATION WITH SPATIALLY VARYING VISCOSITY

To evaluate the robustness of PDE-OP and the baseline methods against heterogeneous physical parameters and nonlinear dynamics, we test these methods on the 1D Burgers' equation control task (which is formalized in Problem (51)), where the viscosity profile $\nu(x)$ is a function of the space variable $x$. Due to its high-computational cost, NMPC is not included as a baseline in our evaluation.

Each method is evaluated on the following target profiles $y_{\text{target}}(x)$:

1. **Standard Targets:** Smooth profiles satisfying the boundary conditions.
   - *Sine Mixture*: $y_{\text{target}}(x) = 0.4 \sin(2\pi x/L) + 0.4 \sin(4\pi x/L)$.
   - *Parabola*: $y_{\text{target}}(x) = 2.0x(L-x)/L^2$.
   - *Zero constant*: $y_{\text{target}}(x) = 0$ (stabilization task).

2. **Advanced Targets:** Discontinuous, high-frequency, or stochastic profiles.
   - *Step*: $y_{\text{target}}(x) = 0.6$ for $x \in [0.3L, 0.7L]$, else 0.
   - *High-Frequency*: $y_{\text{target}}(x) = 0.5 \sin(2\pi x/L) + 0.2 \sin(12\pi x/L)$.
   - *Complex Gaussian*: A stochastic superposition of five random Gaussian modes, $y_{\text{target}}(x) = \sum_{i=1}^{5} A_i \exp(\frac{-(x-\mu_i)^2}{2\sigma_i^2})$.

We compare PDE-OP against the Adjoint Method, SBTO and DeepONet MPC as shown in Figures 11-12.

Table 7: Burger Optimization Task with varying coefficients. For each method and standard target profile we report runtime (s) and MSE.

| Method | Sine Mixture | | Parabola | | Zero | |
| | Runtime | Accuracy | Runtime | Accuracy | Runtime | Accuracy |
| --- | --- | --- | --- | --- | --- | --- |
| Adjoint Method | 194.271s | 0.03440 | 143.632s | 0.00034 | 41.93s | **5.7012e-8** |
| SBTO | 122.242s | 0.00990 | 121.589s | 0.00021 | 121.581s | 0.00020 |
| DeepONet MPC | 200.005s | 0.03550 | 199.594s | **0.00016** | 199.967s | 0.00019 |
| **PDE-OP (Ours)** | **0.49s** | **0.00918** | **0.356s** | 0.00051 | **0.342s** | 0.00124 |

Table 8: Burger Optimization Task with varying coefficients. For each method and advanced target profile we report runtime (s) and MSE.

| Method | Step | | High Frequency | | Complex Gaussian | |
| | Runtime | Accuracy | Runtime | Accuracy | Runtime | Accuracy |
| --- | --- | --- | --- | --- | --- | --- |
| Adjoint Method | 177.382s | 0.01400 | 159.489s | 0.02080 | 147.412s | 0.11900 |
| SBTO | 120.641s | **0.01070** | 121.745s | 0.04240 | 121.35s | 0.10100 |
| DeepONet MPC | 200.012s | 0.01470 | 199.885s | **0.02050** | 199.791s | 0.11500 |
| **PDE-OP (Ours)** | **0.342s** | 0.01510 | **0.343s** | 0.02230 | **0.342s** | **0.09840** |

**Results.** *Standard Targets.* As shown in Table 7 and Figure 11, PDE-OP achieves the lowest error on the Sine Mixture target (MSE $\mathbf{9.18 \times 10^{-3}}$), significantly outperforming the Adjoint Method

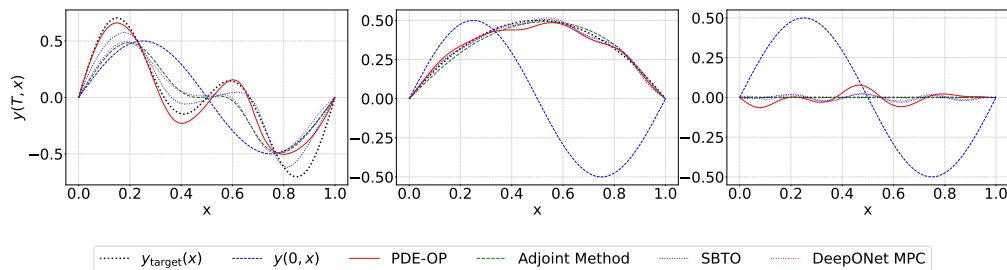

Figure 11: 1D Burgers' Equation with spatially varying coefficient. Comparison between PDE-OP and each baseline method's solution on three different target profile: $y_{\text{target}}(x) = 0.4(\sin(2\pi x)) + 0.4(\sin(4\pi x))$ (left), $y_{\text{target}}(x) = 2x(1-x)$ (center), and $y_{\text{target}}(x) = 0$ (right).

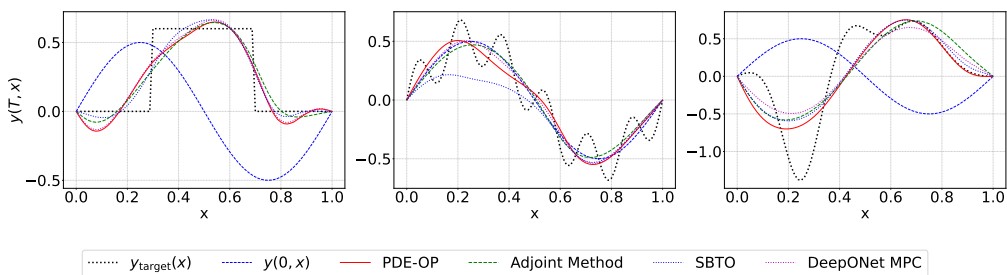

Figure 12: 1D Burgers' Equation with spatially varying coefficient. Comparison of PDE-OP and baselines for advanced target profiles: *Step* (discontinuous)(left), *High-Frequency* ($y_{\text{target}}(x) = 0.5\sin(2\pi x) + 0.2\sin(12\pi x)$)(center), and *Complex Gaussian* (stochastic multi-modal profile)(right).

$(3.44 \times 10^{-2})$. While the Adjoint method achieves near-machine precision on the trivial Zero target, PDE-OP maintains competitive accuracy across all shapes while running $\approx 400\times$ faster than the best baseline.

*Advanced Targets.* Table 8 reports the inference time of PDE-OP and the baseline methods, along with the corresponding MSE between the desired target and the final state trajectory, across different advanced target profiles. For a *Complex Gaussian* target, PDE-OP achieves the best accuracy (MSE **$9.84 \times 10^{-2}$**), outperforming all the baseline methods as depicted in Figure 12. For the Step and High-Frequency targets, PDE-OP matches the accuracy of the Adjoint method and DeepONet MPC (MSE $\approx 10^{-2}$) but requires only **0.34** seconds compared to $> 150$ seconds for the baselines. SBTO struggles with the High-Frequency target, likely due to convergence to local minima.

*Takeaway.* For the nonlinear Burgers' equation with heterogeneous viscosity, PDE-OP delivers robust control performance across diverse target geometries. It matches or exceeds the accuracy of computationally expensive iterative optimizers on complex tasks while offering three orders of magnitude acceleration in inference speed.

### 9.9.7 OPTIMAL CONTROL OF 1D HEAT EQUATION(BATTERY THERMAL MANAGEMENT)

For this task, we evaluate the performance of each method on two categories of target profiles $y_{\text{target}}(x)$. **Standard Targets:** (i) *Sine* ($y_{\text{target}} \propto 0.6 + 0.3\sin(2\pi x)$); (ii) *Ramp* ($y_{\text{target}} = x + 0.5$); and (iii) *Constant* ($y_{\text{target}} = 1$). **Advanced Targets:** (i) *Step* (discontinuous profile); (ii) *High-Frequency* (superposition of low and high frequency sine waves); and (iii) *Complex Gaussian* (stochastic mixture of Gaussian modes). We compare the proposed PDE-OP model against classical baselines (Adjoint Method, LMPC) and neural-based baselines (SBTO, DeepONet MPC). Figures 13 and 14 show the target profile and the terminal state produced by each method. From Figure 13, we observe that each method produces terminal states that closely align with the desired terminal state. As the target profiles become more complex, Figure 14 shows that all methods tend to capture only the overall trend, while high-frequency components are more challenging to approximate accurately.

Table 9: 1D Heat Optimization Task (Battery Physics). For each method and standard target profile, we report runtime (s) and MSE.

| Method | Constant | | Sine | | Ramp | |
|---|---|---|---|---|---|---|
| | Runtime | Accuracy | Runtime | Accuracy | Runtime | Accuracy |
| LMPC | 0.442s | **2.33e-14** | 0.413s | **0.00035** | 0.458s | 0.00008 |
| Adjoint Method | 0.318s | 1.05e-10 | 0.683s | **0.00035** | 0.499s | 0.00008 |
| SBTO | 13.260s | 0.00010 | 14.265s | 0.00077 | 13.315s | 0.00020 |
| DeepONet MPC | 124.322s | 1.66e-09 | 124.263s | 0.00039 | 124.301s | **0.00008** |
| **PDE-OP (Ours)** | **0.030s** | 9.56e-06 | **0.031s** | 0.00035 | **0.030s** | 0.00009 |

Table 10: 1D Heat Optimization Task (Battery Physics). For each method and advanced target profile, we report runtime (s) and MSE.

| Method | Step | | High Frequency | | Complex Gaussian | |
|---|---|---|---|---|---|---|
| | Runtime | Accuracy | Runtime | Accuracy | Runtime | Accuracy |
| LMPC | 0.359s | **0.01372** | 0.397s | **0.02164** | 0.533s | **0.11625** |
| Adjoint Method | 0.404s | **0.01372** | 0.505s | **0.02164** | 0.438s | **0.11625** |
| SBTO | 13.312s | 0.01759 | 13.309s | 0.02403 | 13.264s | 0.12829 |
| DeepONet MPC | 124.198s | 0.01462 | 124.180s | 0.02238 | 124.260s | 0.12037 |
| **PDE-OP (Ours)** | **0.032s** | 0.01382 | **0.031s** | **0.02164** | **0.031s** | **0.11625** |

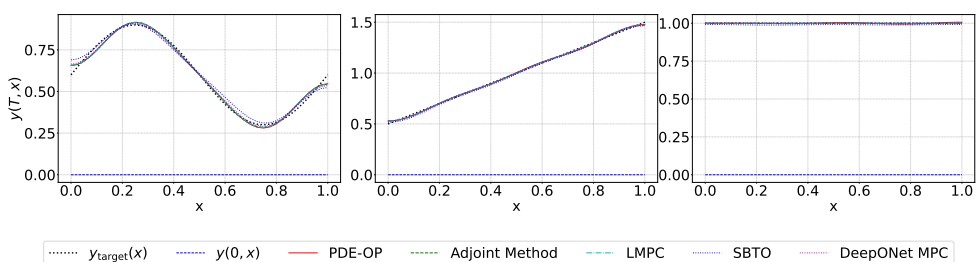

Figure 13: 1D Heat Equation (Battery Physics). Comparison of PDE-OP and other methods for three target profiles: $y_{\text{target}}(x) = 0.6 + 0.3(\sin(2\pi x))$(left), $y_{\text{target}}(x) = x + 0.5$(center), and $y_{\text{target}}(x) = 1$(right).

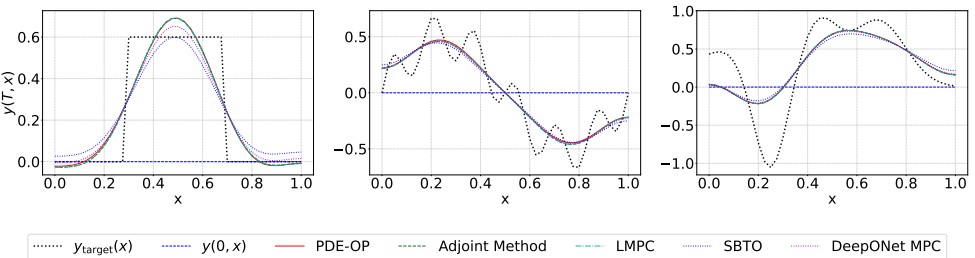

Figure 14: 1D Heat Equation (Battery Physics). Comparison of PDE-OP and baselines for advanced target profiles: *Step* (discontinuous)(left), *High-Frequency* ($y_{\text{target}}(x) = 0.5\sin(2\pi x) + 0.2\sin(12\pi x)$)(center), and *Complex Gaussian* (stochastic multi-modal profile)(right).

**Results.** We evaluate each method on the Lithium-Ion battery thermal management task with $D = 0.13$, which is $13\times$ higher than in the previous setting ($D = 0.01$), causing the heat to dissipate significantly faster and requiring more energy for control.

*Standard Targets.* Table 3 reports the MSE and computational time of each method given the Constant, Sine, and Ramp profiles. For the sinusoidal target, PDE-OP achieves an MSE of $3.52 \times 10^{-4}$,

matching the accuracy of the Adjoint Method and LMPC ($3.50 \times 10^{-4}$). For the Ramp target, PDE-OP yields an MSE of $9.78 \times 10^{-5}$, which is also comparable to the best baselines. In terms of speed, PDE-OP generates solutions in $0.030s$, approximately $15\times$ faster than the Adjoint method ($0.499s$) and LMPC ($0.458s$).

*Advanced Targets.* Table 4 reports the MSE and computational time of each method in presence of complex target profiles. For the Step and High-Frequency targets, PDE-OP performs identically to the best baselines, namel Adjoint method and LMPC, with an MSE of $0.02164$. When the target is a *Complex Gaussian*, all methods incur a higher error (MSE $\approx 1.16 \times 10^{-1}$). This is due to the high diffusivity ($D = 0.13$), which prevents the formation of sharp, multi-modal peaks under the given constraints. PDE-OP matches the error of the Adjoint method ($1.16 \times 10^{-1}$), but runs approximately $15$ faster then the best baselines.

*Takeaway.* On this physically realistic battery task, PDE-OP achieves the same solution quality as the optimal Adjoint and LMPC methods, even in presence of complex target profiles. Consistent with the other tasks, PDE-OP achieves a speedup of $1$ to $2$ orders of magnitude compared to the most efficient baseline methods, while running $3$ and $4$ order of magnitude faster than the neural-based models, namely SBTO and DeepONet MPC, respectively.

## 9.10 DISCRETE-TIME NEURAL OPERATOR WITH DIRECT CONTROL FIELD $u(x, t)$

In this section we describe a variant of our discrete-time neural operator that learns *directly* the spatio–temporal control field $u(t, x)$ rather than the weights of the basis function as introduced in Section 4.2. Similarly to the original DeepONet (Lu et al., 2021), PDE-OP's model $\mathcal{Y}_\theta$ factors into a *branch* network that encodes input functions and a *trunk* network that encodes query coordinates. At each time $t_k$, given the current state $y(t_k, x)$ and the applied control $u(t_k, x)$, this model predicts the next state at coordinate $x$:

$$\hat{\boldsymbol{y}}(t_{k+1}, x) \; = \; \mathcal{Y}_\theta\big(t, \, x, \, \hat{\boldsymbol{y}}(t_k, x), \, \hat{\boldsymbol{u}}(t_k, x)\big) \tag{53}$$

**Branch network.** The branch net encodes the estimated system state $\hat{\boldsymbol{y}}(t_k, x)$ sampled at $n$ fixed sensor locations together with estimated control signals $\hat{\boldsymbol{u}}(t_k, x)$, and maps this concatenated input to a $d$-dimensional vector of coefficients $\boldsymbol{b}(\hat{\boldsymbol{y}}(t_k, x), \hat{\boldsymbol{u}}(t_k)) \in \mathbb{R}^d$.

**Trunk network.** The trunk net encodes the spatial query coordinate $x$, producing a latent feature vector $\boldsymbol{\gamma}(x) \in \mathbb{R}^d$.

The estimated subsequent (in time) state $\hat{\boldsymbol{y}}(t_{k+1}, x)$ at location $x_i$ is obtained by combining the branch coefficients and trunk features through an inner product:

$$\hat{y}(t_{k+1}, x_i) \; = \; \sum_{j=1}^{d} b_j(\hat{y}(t_k, x_i), \hat{\boldsymbol{u}}(t_k, x_i)) \, \gamma_j(x_i), \;\; i = 1, \ldots, n. \tag{54}$$

**Model initialization and training (direct $u$).** In this setting the dynamic predictor $\mathcal{Y}_\theta$ takes as input the control action *sampled on the spatial grid*, together with the current state. $\mathcal{Y}_\theta$ is initialized in a supervised fashion by generating the target trajectories with a numerical PDE solver under diverse time-varying controls $u(t, x)$.

We sample admissible control sequences $\{u(t_k, x_i)\}$ on the grid $\{x_i\}_{i=1}^{n}$ from a zero-mean Gaussian random field (GRF) with prescribed spatial covariance (and optional temporal correlation), after which a clipping operation is applied to meet the actuator bounds $u_{\min} \leq u(t_k, x_i) \leq u_{\max}$. Rolling the solver with these inputs produces states $y(t_k, \cdot)$ satisfying (1b)-(1f). Each trajectory is converted into one-step training pairs

$$\big(y(t_k, x_i), \, u(t_k, x_i)\big) \; \longmapsto \; y(t_{k+1}, x_i), \qquad k = 0, \ldots, T-1, \quad i = 1, \ldots, n$$

Formally, the dataset consists of triples $\mathcal{D} = \Big\{ \big(\boldsymbol{y}(t_k, x), \, \boldsymbol{u}(t_k, x), \, \boldsymbol{y}(t_{k+1}, x)\big) \Big\}$, with $\boldsymbol{y}(t_k, x) = [y(t_k, x_1), \ldots, y(t_k, x_n)]^\top$ and $\boldsymbol{u}(t_k, x) = [u(t_k, x_1), \ldots, u(t_k, x_n)]^\top$. $\mathcal{Y}_\theta$ is trained to minimize the following loss

$$\min_\theta \; \mathbb{E}_{(\boldsymbol{u}, y) \sim \mathcal{D}} \Big[ \big\| \hat{y}(t, x) - y(t, x) \big\|^2 \Big], \tag{55}$$

### 9.10.1 MPC FOR DIRECT CONTROL FIELD $u(x, t)$

Here we describe a "direct-$u$" variant for viscous Burgers with Dirichlet boundaries, where the control action is parameterized *at every spatial collocation point*. At each time $t_k$ a vector, the optimization task consists of finding $\boldsymbol{u}(t_k, x) \in \mathbb{R}^{N_x}$ (e.g., for a given time instant $t_k$ we have $N_x$ spatial collocation point), and applying an implicit Crank–Nicolson step to propagate the state in time from $t_k$ to $t_{k+1}$.

$$\min_{\{\boldsymbol{u}(t_i)\}_{i=k}^{k+N_p-1}} \sum_{i=k}^{k+N_p-1} \|\boldsymbol{y}(t_i) - \boldsymbol{y}_{\text{target}}\|_{\boldsymbol{T}_Q}^2 + \|\boldsymbol{u}(t_i)\|_{\boldsymbol{R}}^2 + \|\boldsymbol{y}(t_{k+N_p}) - \boldsymbol{y}_{\text{target}}\|_{\boldsymbol{Q}}^2 \tag{56}$$

$$\text{s.t. } 40, \quad u_{\min} \leq \boldsymbol{u}(t_i) \leq u_{\max}.$$

In case of closed loop (NMPC), a horizon of length $N_p$ yields $N_p N_x$ decision variables per iteration; we warm start across receding horizons and use bound-constrained quasi-Newton (e.g., L-BFGS-B/SLSQP). Here $\mathbf{T}_Q, \mathbf{R}$, and $\mathbf{Q}$ represent the weight matrices, which correspond to identical with the Lagrange multipliers $\boldsymbol{\lambda}$ of our PDE-OP's training algorithm. To compute the gradients, we either backpropagate through the discrete CN integrator using automatic differentiation or use the fully discrete adjoint; both methods are exact but require one forward rollout and one backward sweep over the horizon. This grid-wise formulation removes any basis bias and shows the attainable tracking when the controller has full spatial freedom, but it is While direct modeling of the control action doesn't require any basis representation, its computational cost *higher* w.r.t. to the case in which a basis representation is adopted: the per-step cost scales roughly with $N_p N_t N_x$, due to line-search/gradient evaluation entailing multiple CN solves. We therefore include it as a comprehensive baseline to contextualize the efficiency of our reduced-parameter (mode-based) controllers.

### 9.10.2 EXPERIMENT RESULTS FOR DIRECT CONTROL FIELD $u(t, x)$

When using the direct control field modeling for $u(t, x)$, we evaluate each method on the nonlinear, time-variant optimal control of the 1D Burgers' equation, using the same target profiles presented in Section 5.3. Tables 11-12 report the runtime (s) and terminal tracking error (MSE) of each method. All methods use the same spatial/temporal discretization and solver settings as in 9.12. Figure 15 shows the predicted voltage (central sub-figure of Fig. 15) of PDE-OP's dynamic component $\mathcal{Y}_\theta$ at test time, which is compared to the solution (left sub-figure) obtained with a numerical PDE-solver. Figures 16 and 17 show the comparison of PDE-OP's predicted terminal state $\hat{y}(T, x)$ and those obtained using the baseline methods, over the same target profiles introduced in the main text. In figure 16, we set $N_p = 1$ for NMPC and $N_p = 10$ for the Adjoint-method, while in figure 17, we set $N_p = 4$ for NMPC and $N_p = 20$ for the Adjoint-method.

Table 11: 1D Burgers' Equation with direct control field. PDE-OP vs. classical baselines with short horizon $N_p$. Here we use $N_p$=1 for NMPC and $N_p$=10 for adjoint. We report runtime (s) and MSE (lower is better).

| Method | Parabola | | Sine | | Constant | |
|---|---|---|---|---|---|---|
| | Runtime | MSE | Runtime | MSE | Runtime | MSE |
| NMPC | 39.254s | **0.00002** | 76.368s | 0.00014 | 36.483s | 7.9141e−06 |
| Adjoint-Method | 47.805s | 0.00124 | 79.644s | 0.00317 | 24.956s | **2.7618e−08** |
| **PDE-OP (ours)** | **0.267s** | 0.00003 | **0.256s** | **0.00008** | **0.257s** | 0.00008 |

**Results.** Across all targets, PDE-OP is *two orders of magnitude faster* than classical methods, between $100$–$300\times$ faster then the NMPC and the Adjoint. In terms of accuracy, for the *Sine* target, PDE-OP achieves the lowest MSE; for the *Parabola*, NMPC attains the best results with PDE-OP close behind; for the *Zero* target, the adjoint method achieves the lowest MSE with PDE-OP producing an MSE of the order of $10^{-5}$.

**Results.** With $N_p$=4 (NMPC) and $N_p$=20 (adjoint), PDE-OP remains dramatically faster, three to four orders of magnitude—than classical solvers. For the *Parabola* target, PDE-OP is $\sim 3250\times$

Table 12: 1D Burgers' Equation with direct control field. PDE-OP vs. classical baselines with longer horizons. Here we use $N_p=4$ for NMPC and $N_p=20$ for adjoint. We report runtime (s) and MSE (lower is better).

| Method | Parabola Runtime | Parabola MSE | Sine Runtime | Sine MSE | Constant Runtime | Constant MSE |
|---|---|---|---|---|---|---|
| NMPC | 826.176s | 0.00016 | 3143.735s | 0.00032 | 470.851s | 0.00061 |
| Adjoint-Method | 186.910s | 0.00129 | 263.891s | 0.00316 | 51.702s | **4.9213e−09** |
| **PDE-OP (ours)** | **0.254s** | **0.00003** | **0.277s** | **0.00008** | **0.956s** | 0.00008 |

faster than NMPC and $\sim 740\times$ faster than adjoint, while also reporting the best MSE ($3\times10^{-5}$). For the *Sine* target, the speedups over classical methods is $\sim 11300\times$ vs. NMPC and $\sim 950\times$ vs. adjoint, while attaining (again) the lowest accuracy ($8\times10^{-5}$). For the *Zero* target, the Adjoint method attains near-perfect regulation ($4.9\times10^{-9}$), with PDE-OP is still fastest, running in $0.956\,s$ and approximately $54\times$ faster than the Adjoint method and $495\times$ faster than NMPC, while reporting a very small tracking error ($8\times10^{-5}$).

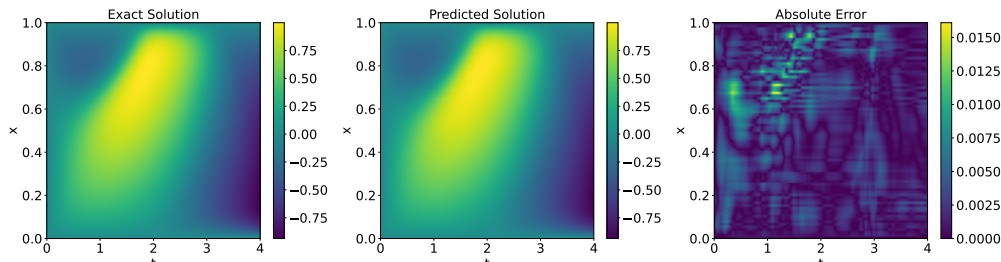

Figure 15: 1D Burgers' Equation with direct control field. PDE-OP's dynamic predictor $\mathcal{Y}_\theta$ solution estimate at test time: numerical solution (left), model prediction (center), and absolute error (right).

### 9.10.3 ABLATION STUDY: EFFECT OF THE NUMBER OF BASIS FUNCTIONS $M$ ON PDE-OP'S PERFORMANCE

The parameter $M$ (number of spatial basis functions) governs the spatial resolution and expressivity of the control input $u(t,x) = \sum_{i=1}^{M} c_i(t)\phi_i(x)$. A small $M$ acts as a low-pass filter, restricting the controller to generate only smooth and low-frequency control signals. Conversely, a large $M$ allows for fine-grained spatial control, which is theoretically necessary to approximate high-frequency or discontinuous target profiles (e.g., Step or Complex Gaussian). To quantify the trade-off between control expressivity and computational complexity, we train the PDE-OP controller with varying number of basis functions $M \in \{2, 4, 6, 8, 10\}$ to solve the optimal control problem for the 1D Heat Equation.

Table 13: Ablation study on basis dimension $M$. We report the Final MSE (and inference runtime in parentheses) across three distinct target profiles. Increasing $M$ significantly reduces error for complex targets without increasing computational cost.

| Basis Dim ($M$) | Sine Accuracy (Runtime) | Step Accuracy (Runtime) | Complex Gaussian Accuracy (Runtime) |
|---|---|---|---|
| 2 | 0.01390 (0.029s) | 0.08360 (0.029s) | 0.26800 (0.030s) |
| 4 | 0.00116 (0.029s) | 0.02190 (0.029s) | 0.11700 (0.031s) |
| 6 | 0.00041 (0.029s) | 0.01260 (0.029s) | 0.02420 (0.031s) |
| 8 | 0.00023 (0.029s) | 0.01150 (0.029s) | 0.00608 (0.030s) |
| 10 | **0.00010** (0.029s) | **0.00723** (0.029s) | **0.00034** (0.030s) |

**Results.** As shown in Table 13 and Figure 18, increasing $M$ results in a monotonic and significant reduction in the tracking error across all targets. For the smooth *Sine* target, accuracy improves by two orders of magnitude as $M$ increases from 2 to 10. The effect is most pronounced for the

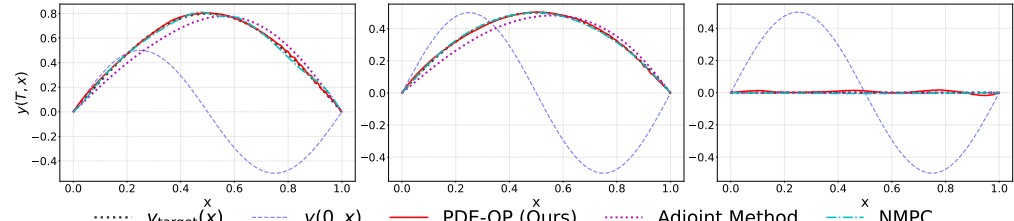

Figure 16: 1D Burgers' Equation with direct control field. Comparison between PDE-OP and each baseline method's solution on three different target profile: $y_{\text{target}}(x) = 0.8(\sin(\pi x))$ (left), $y_{\text{target}}(x) = 2x(1-x)$ (center), and $y_{\text{target}}(x) = 0$ (right). We set $N_p = 1$ for NMPC, and $N_p = 10$ for adjoint-method.

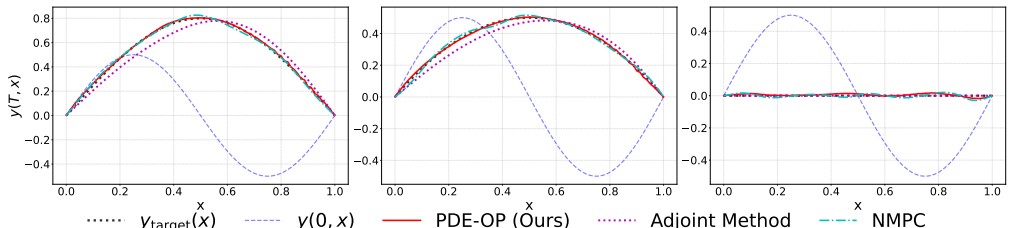

Figure 17: 1D Burgers' Equation with direct control field. Comparison between PDE-OP and each baseline method's solution on three different target profile: $y_{\text{target}}(x) = 0.8(\sin(\pi x))$ (left), $y_{\text{target}}(x) = 2x(1-x)$ (center), and $y_{\text{target}}(x) = 0$ (right). We set $N_p = 4$ for NMPC, and $N_p = 20$ for adjoint-method.

*Complex Gaussian* target, where the MSE drops from $0.268$ ($M = 2$) to $3.4 \times 10^{-4}$ ($M = 10$). This suggests that higher-dimensional basis are essential for capturing multi-modal shapes that contain significant high-frequency energy.

Crucially, the increase in $M$ incurs with **negligible computational overhead**. The inference time remains stable at $\approx 0.029$–$0.031$s regardless of $M$. This is because the neural operator processes inputs in parallel on the GPU, and the slight increase in the dimension of the basis projection matrix has minimal impact on the forward-pass latency. Thus, unlike classical optimization where cost scales with decision variables, PDE-OP allows for high-resolution control ($M = 10$) in real-time.

### 9.11 ABLATION STUDY ON RUNTIME EVALUATION

To evaluate the trade-off between computational runtime and accuracy, we implemented an adaptive benchmarking to determine the minimum computational time required by each baseline method to achieve a solution quality comparable to those of our proposed method, PDE-OP.

To do so, we first run PDE-OP to set a baseline error threshold $E_{\text{thresh}}$. Let $\hat{u}_{\text{PDE-OP}}(t, x)$ be the control action generated by the surrogate controller $\mathcal{U}_\omega$, and let $y_{\text{PDE-OP}}(T, x)$ be the resulting terminal state computed using the discrete-time neural operator $\mathcal{Y}_\theta$. This error threshold is defined as the terminal tracking error (MSE):

$$E_{\text{thresh}} = \|y_{\text{PDE-OP}}(T, x) - y_{\text{target}}(x)\|_2^2, \tag{57}$$

where $y_{\text{target}}(x)$ represents the prescribed target profile (as defined in Section 5). We then constrain each baseline to achieve a final error $\mathcal{L}_{\text{terminal}} \leq E_{\text{thresh}}$. In what follows, we describe the stopping criteria adopted by each method to match PDE-OP's error threshold $E_{\text{thresh}}$ defined by (57).

**1. Iterative Learning Baselines (Adaptive Early Stopping)** For methods relying on iterative gradient-based optimization—specifically **SBTO** and **DeepONet MPC**—we implemented an online convergence check, detailed as follows.

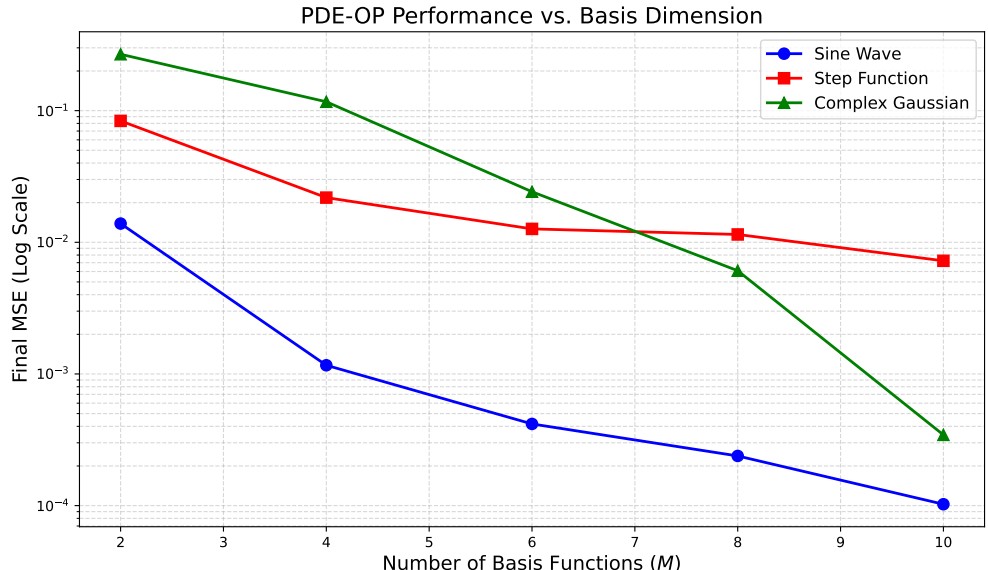

Figure 18: PDE-OP Performance vs. Basis Dimension($M$)

- **SBTO:** The global optimization loop (optimizing the control weights **c**) is terminated at iteration $k$ if, given the predicted state $y_{\text{SBTO}}(T, x)$, the terminal loss satisfies the following:

$$\|y_{\text{SBTO}}(T, x) - y_{\text{target}}(x)\|_2^2 \leq E_{\text{thresh}} \tag{58}$$

- **Neural MPC:** The inner optimization loop at each time step $t_k$ (planning the local horizon) is terminated early if the predicted trajectory error falls below $E_{\text{thresh}}$.

**2. Classical Baselines (Tolerance Relaxation)** To reduce the runtime of time-stepping methods like the **Adjoint Method** and **Linear/Nonlinear MPC**, we tuned their solver tolerances, since they require terminal target access and must integrate all the way to the final time step. We treated the solver tolerance $\epsilon$ (e.g., `ftol` for L-BFGS-B, `eps` for OSQP) as a hyperparameter and performed a logarithmic grid search to find the fastest setting satisfying the condition:

$$\min_{\epsilon} \quad \text{Runtime}(\epsilon) \quad \text{subject to} \quad \|y_\epsilon(T, x) - y_{\text{target}}(x)\|_2^2 \leq E_{\text{thresh}}. \tag{59}$$

Next, we report the runtime of the baseline methods run with the stopping criteria described above.

Table 14: **Runtime Evaluation (Constant Target).** Comparison of runtime time (in seconds) required to match the target accuracy threshold ($E_{\text{thresh}}$) set by PDE-OP. Baseline methods run using the respective stopping criteria. In parenthesis we report the runtime time of each method running without the corresponding stopping criteria.

| Method | Voltage Control 
 *(Time in s)* | 1D Heat Eq. 
 *(Time in s)* | 1D Burgers' Eq. 
 *(Time in s)* |
|---|---|---|---|
| **PDE-OP (Ours)** | **0.037** | 0.223 | **0.059** |
| Direct / Linear / Nonlinear MPC[†] | 1.567(2.422) | 0.301(0.443) | 59.463(132.243) |
| Adjoint Method | 0.046(0.049) | **0.194**(0.485) | 12.904(21.487) |
| SBTO (Early Stop) | 0.123(0.418) | N/A(15.647) | 2.461(59.179) |
| DeepONet MPC (Early Stop) | N/A | 88.327(144.306) | 46.342(98.296) |

[†] *Direct Method used for Voltage; Linear MPC for Heat; Nonlinear MPC for Burgers.*

**Results: Speed-Accuracy Trade-off Analysis.** We conducted the ablation study using the *constant profile* setting, as it resulted in the most significant accuracy gap between PDE-OP and the baseline methods. Table 14 reports the runtime required for each method to match the solution quality ($E_{\text{thresh}}$) of PDE-OP on the constant target profiles (i.e., $y_{\text{target}}(x) = 1$ for Voltage and 1D Heat equation, and $y_{\text{target}}(x) = 0$ for 1D Burgers' equation). When adopting the stopping criteria described above, we observe a modest reduction in runtime for each method. Despite so, PDE-OP continue to demonstrate a significant computational advantage, particularly in nonlinear settings, which is also the most complex. For the **1D Burgers' Equation**, where nonlinearities pose a major challenge for classical solvers, the speedup is dramatic: PDE-OP (0.059s) is approximately $218\times$ **faster than the Adjoint method** and $1000\times$ **faster than Nonlinear MPC**, even when those baselines are tuned to the loosest valid tolerance.

Notably, some entries in the table are marked as N/A, indicating that the corresponding method failed to pass the error threshold ($E_{\text{thresh}}$) achieved by PDE-OP (see Table 3).

### 9.12 HYPERPARAMETER SETTINGS

In Tables 15 - 18, we provide the values of the hyperparameters of PDE-OP model and the baseline methods related to the voltage optimization experiments; Tables 19 - 23 report the values of the hyperparameters of PDE-OP model and the baselines related to the 1D heat equation experiments;Tables 24 - 28 report the values of the hyperparameters of PDE-OP model and the baselines related to the 2D heat equation experiments; Tables 29 - 33 report the values of the hyperparameters of PDE-OP model and baseline methods related to the 1D Burgers' equation experiments; Tables 34 - 37 report the values of the hyperparameters of PDE-OP model and the baselines related to the 1D burger with varying coefficient equation experiments; For a fair evaluation, each method adopts the same PDE, grid, and horizon parameters $n, N_x, N_t, [u_{\min}, u_{\max}], [c_{\min}, c_{\max}]$ .

Table 15: Voltage equation: simulation parameters and model/training hyperparameters. *MLP* (Goodfellow et al., 2016) a fully connected feedforward network.

| Category | Hyperparameter | Value |
|---|---|---|
| PDE & Simulation | Domain length $L$ | 1.0 |
| | Final time $T$ | 5.0 |
| | Diffusion $D$ | 0.1 |
| | Leakage $\beta$ | 1.0 |
| | Control gain $\alpha$ | 2.0 |
| | Initial state $V(x,0)$ | 0.0 |
| | Reference $V_{\text{ref}}(x)$ | 1.0 |
| | Control bounds $[u_{\min}, u_{\max}]$ | $[-1.0,\ 1.0]$ |
| Discretization | Solver grid $(n)$ | 101 |
| | Solver steps $(N_t)$ | 101 |
| | Number of sensors $(N_x = n)$ | 101 |
| PDE-OP's $\mathcal{Y}_\theta$ | Number of Train Trajectories | 1000 |
| | Number of Test/Val Trajectories | 100 |
| | Learning Rate | $5 \times 10^{-4}$ |
| | Number of epochs | 1000 |
| | Optimizer | Adam |
| | Batch size | 1024 |
| | Activation function | SiLU |
| | Lagrange Multiplier $\rho$ | 0.05 |
| | Branch net $(b)$: # layers | 4 |
| | Branch net $(b)$: hidden sizes | $[512, 512, 512, 512]$ |
| | Trunk net $(\gamma)$: # layers | 4 |
| | Trunk net $(\gamma)$: hidden sizes | $[512, 512, 512, 512]$ |
| | Feature dimension $(d)$ | 512 |
| PDE-OP's $\mathcal{U}_\omega$ | Learning rate | $1 \times 10^{-3}$ |
| | Number of epochs | 1000 |
| | Optimizer | AdamW |
| | Batch size = Number of Train Trajectories | 256 |
| | Activation function | RELU |
| | MLP: # layers | 3 |
| | MLP: hidden sizes | $[256, 256, 256]$ |

Table 16: Voltage equation: Direct(finite-difference) method parameters.

| Category | Hyperparameter | Value / Note |
|---|---|---|
| Optimizer | Algorithm | L-BFGS-B (box bounds $[u_{\min}, u_{\max}]$) |
| | Max iterations | 100 |
| | Tolerances | `ftol` $= 10^{-6}$ (and/or `gtol`) |
| | Initialization | $\boldsymbol{u}(t_0) = \boldsymbol{0}$ (uniform) |
| Finite diff. | Stencil | forward (optionally central) |
| | Step size | $\epsilon_j = \eta(1 + |u_j|), \eta = 10^{-6}$ |
| Cost eval | Rollout solver | CN |
| | Runtime note | $N_x{+}1$ evals/grad (or $2N_x$ for central) |

Table 17: Voltage equation: Adjoint-method parameters.

| Category | Hyperparameter | Value / Note |
|---|---|---|
| Optimizer | Algorithm | L-BFGS-B (box bounds $[u_{\min}, u_{\max}]$) |
| | Max iterations | 100 |
| | Tolerances | $\mathtt{ftol} = 10^{-6}$ (and/or $\mathtt{gtol}$) |
| | Initialization | $\boldsymbol{u}(t_0) = \mathbf{0}$ |
| Adjoint solves | Forward stepper | CN |
| | Backward sweep | discrete adjoint (one pass) |
| Numerics | Linear solves | direct (LU) or iterative tol $10^{-10}$ |

Table 18: Voltage equation: SBTO (Surrogate-Based) method parameters.

| Category | Hyperparameter | Value / Note |
|---|---|---|
| Optimizer | Algorithm | Adam |
| | Learning Rate | $1 \times 10^{-2}$ |
| | Max iterations | 200 |
| | Initialization | Latent parameters $\mathbf{0}$ (maps to $\boldsymbol{u} = \mathbf{0}$) |
| Constraints | Method | Soft saturation ($\tanh(\cdot) \times u_{\max}$) |
| Loss | Formulation | MSE (tracking) + $L_2$ penalty |
| | Effort weight | $1 \times 10^{-5}$ |
| Surrogate | Model | DeepONet (Frozen parameters) |
| | Gradient | Automatic Differentiation |

Table 19: 1D Heat equation: simulation parameters and model/training hyperparameters. *LSTM* (Hochreiter & Schmidhuber, 1997) denotes a Long Short-Term Memory network.

| Category | Hyperparameter | Value |
|---|---|---|
| PDE & Simulation | Domain length $L$ | 1.0 |
| | Final time $T$ | 1.0 |
| | Diffusion $D$ | 0.01 |
| | Leakage $\beta$ | 0.5 |
| | Control gain $\alpha$ | 2.0 |
| | Initial state $V(x, 0)$ | 0.0 |
| | Reference $V_{\text{ref}}(x)$ | 0.0 |
| | Weight bounds $[c_{\min}, c_{\max}]$ | $[-1.0,\ 1.0]$ |
| Discretization | Solver grid $(n)$ | 41 |
| | Solver steps $(N_t)$ | 41 |
| | Number of sensors $(N_x = n)$ | 41 |
| | Number of basis functions $(M)$ | 6 |
| PDE-OP's $\mathcal{Y}_\theta$ | Learning Rate | $1 \times 10^{-3}$ |
| | Number of Train Trajectories | 5000 |
| | Number of Test/Val Trajectories | 1000 |
| | Number of epochs | 2000 |
| | Optimizer | AdamW |
| | Batch size | 128 |
| | Activation function | RELU |
| | Branch net $(b)$: # layers | 4 |
| | Branch net $(b)$: hidden sizes | $[512, 512, 512, 512]$ |
| | Trunk net $(\gamma)$: # layers | 4 |
| | Trunk net $(\gamma)$: hidden sizes | $[512, 512, 512, 512]$ |
| | Feature dimension $(d)$ | 512 |
| PDE-OP's $\mathcal{U}_\omega$ | Learning rate | $1 \times 10^{-4}$ |
| | Number of epochs | 5000 |
| | Optimizer | AdamW |
| | Batch size = Number of Train Trajectories | 2048 |
| | Activation function | RELU |
| | LSTM: # layers | 2 |
| | LSTM: hidden sizes | $[256, 256]$ |

Table 20: 1D Heat equation: Linear MPC (LMPC) parameters. $(^*)$ The basis function are used by each method.

| Category | Parameter | Value / Note |
|---|---|---|
| Horizon | Prediction length $N_p$ | 10 |
| Dynamics | State update | $\boldsymbol{y}(t_{k+1}) = \boldsymbol{A}\boldsymbol{y}(t_k) + \boldsymbol{B}\boldsymbol{u}(t_k) + \boldsymbol{g}$ |
| | Discretization | Crank–Nicolson (linear) |
| Actuation | Basis$^*$ | $\cos(j\pi x/L),\ j = 0, \ldots, m-1$ |
| Solver | QP solver | OSQP (CVXPY), warm start |
| Stopping | Tolerances | default OSQP (primal/dual res.) |

Table 21: 1D Heat equation: Adjoint method parameters.

| Category | Parameter | Value / Note |
|---|---|---|
| Horizon | Prediction length $N_p$ | 10 |
| Forward | Integrator | Crank–Nicolson (linear), same $\boldsymbol{A}, \boldsymbol{B}, \boldsymbol{g}$ |
| Adjoint | Backward recursion | $\boldsymbol{\lambda}_{N_t} = \boldsymbol{Q}\left(\boldsymbol{y}_{N_t} - \boldsymbol{y}_{\text{target}}\right)$ |
| | | $\boldsymbol{\lambda}_k = \boldsymbol{A}^\top \boldsymbol{\lambda}_{k+1} + \boldsymbol{T}_Q\left(\boldsymbol{y}(t_k) - \boldsymbol{y}_{\text{target}}\right)$ |
| Gradient | Per step | $\nabla_{\boldsymbol{u}(t_k)} J_h = \boldsymbol{R}\,\boldsymbol{u}(t_k) + \boldsymbol{B}^\top \boldsymbol{\lambda}_{k+1}.$ |
| Optimizer | Algorithm | L-BFGS-B, warm start |
| Stopping | Max iters / tolerances | 100; $\texttt{ftol} = 10^{-6}$ |

Table 22: 1D Heat equation: SBTO (Surrogate-Based) method parameters.

| Category | Hyperparameter | Value / Note |
|---|---|---|
| Optimizer | Algorithm | Adam |
| | Learning Rate | $1 \times 10^{-2}$ |
| | Max iterations | 200 |
| Loss | Formulation | Terminal MSE + Running MSE + Control Effort |
| | Weights | $\lambda_{term} = 1.0, \lambda_{run} = 0.1, \lambda_{effort} = 1 \times 10^{-5}$ |
| Constraints | Method | Soft saturation $(\tanh(\cdot))$ |
| Surrogate | Model | Propagator DeepONet (Frozen) |
| | Rollout | Recurrent (Full Horizon $N_t$) |

Table 23: 1D Heat equation: DeepONet MPC parameters.

| Category | Hyperparameter | Value / Note |
|---|---|---|
| Horizon | Prediction length $N_p$ | 10 |
| Optimizer | Algorithm | Adam (per time step) |
| | Learning Rate | $2 \times 10^{-2}$ |
| | Steps per solve | 200 |
| Loss | Weights | $\lambda_{term} = 1.0, \lambda_{run} = 0.1, \lambda_{effort} = 1 \times 10^{-5}$ |
| Control | Strategy | Receding Horizon (Apply 1st action) |
| | Parameterization | Basis functions (optimized via backprop) |
| Surrogate | Model | Propagator DeepONet (Frozen) |

Table 24: 2D Heat equation: simulation parameters and model/training hyperparameters. *LSTM* (Hochreiter & Schmidhuber, 1997) denotes a Long Short-Term Memory network.

| Category | Hyperparameter | Value |
|---|---|---|
| PDE & Simulation | Domain sizes $L_x, L_y$ | $1.0, 1.0$ |
| | Final time $T$ | $1.0$ |
| | Diffusion $D$ | $0.01$ |
| | Leakage $\beta$ | $0.5$ |
| | Control gain $\alpha$ | $2.0$ |
| | Initial state $V(x, y, 0)$ | $0.0$ |
| | Reference $V_{\text{ref}}$ | $0.0$ |
| | Weight bounds $[c_{\min}, c_{\max}]$ | $[-1.0,\ 1.0]$ |
| Discretization | Solver grid $(N_x \times N_y)$ | $21 \times 21$ |
| | Solver steps $(N_t)$ | $41$ |
| | Number of sensors $(N_{\text{sens}})$ | $441\ (21 \times 21)$ |
| | Number of basis functions $(M)$ | $16\ (4 \times 4)$ |
| PDE-OP's $\mathcal{Y}_\theta$ | Learning Rate | $1 \times 10^{-3}$ |
| | Number of Train Trajectories | $5000$ |
| | Number of Test/Val Trajectories | $1000$ |
| | Number of epochs | $2000$ |
| | Optimizer | AdamW |
| | Batch size | $64$ |
| | Activation function | RELU |
| | Branch net $(b)$: # layers | $4$ |
| | Branch net $(b)$: hidden sizes | $[512, 512, 512, 512]$ |
| | Trunk net $(\gamma)$: # layers | $2$ |
| | Trunk net $(\gamma)$: hidden sizes | $[128, 128]$ |
| | Feature dimension $(d)$ | $128$ |
| PDE-OP's $\mathcal{U}_\omega$ | Learning rate | $1 \times 10^{-5}$ |
| | Number of epochs | $5000$ |
| | Optimizer | AdamW |
| | Batch size = Number of Train Trajectories | $1024$ |
| | Activation function | RELU |
| | LSTM: # layers | $2$ |
| | LSTM: hidden sizes | $[256, 256]$ |

Table 25: 2D Heat equation: Adjoint method parameters.

| Category | Parameter | Value / Note |
|---|---|---|
| Horizon | Prediction length $N_p$ | $10$ |
| Forward | Integrator | Crank–Nicolson (2D linear) |
| | Operator $\mathbf{A}$ | Derived from $\mathbf{I} - \frac{D\Delta t}{2}\mathbf{L}_{2D}$ |
| Adjoint | Backward recursion | $\boldsymbol{\lambda}_{N_t} = \boldsymbol{Q}\left(\boldsymbol{y}_{N_t} - \boldsymbol{y}_{\text{target}}\right)$ |
| | | $\boldsymbol{\lambda}_k = \mathbf{A}^\top \boldsymbol{\lambda}_{k+1} + \boldsymbol{T}_Q\left(\boldsymbol{y}(t_k) - \boldsymbol{y}_{\text{target}}\right)$ |
| Gradient | Per step | $\nabla_{\boldsymbol{u}(t_k)} J_h = \boldsymbol{R}\,\boldsymbol{u}(t_k) + \mathbf{B}^\top \boldsymbol{\lambda}_{k+1}.$ |
| Optimizer | Algorithm | L-BFGS-B, warm start |
| Stopping | Max iters / tolerances | $30$; `ftol` $= 10^{-6}$ |

Table 26: 2D Heat equation: Linear MPC (LMPC) parameters.

| Category | Parameter | Value / Note |
|---|---|---|
| Horizon | Prediction length $N_p$ | 10 |
| Dynamics | State update | $\boldsymbol{y}(t_{k+1}) = \mathbf{A}\boldsymbol{y}(t_k) + \mathbf{B}\boldsymbol{u}(t_k) + \boldsymbol{g}$ |
| | Discretization | Crank–Nicolson (2D Kronecker) |
| Actuation | 2D Basis$^*$ | $\cos(\frac{i\pi x}{L_x})\cos(\frac{j\pi y}{L_y})$ |
| Solver | QP solver | OSQP (CVXPY), warm start |
| Stopping | Tolerances | default OSQP (primal/dual res.) |

Table 27: 2D Heat equation: SBTO (Surrogate-Based) method parameters.

| Category | Hyperparameter | Value / Note |
|---|---|---|
| Optimizer | Algorithm | Adam |
| | Learning Rate | $1 \times 10^{-2}$ |
| | Max iterations | 200 |
| Loss | Weights | $\lambda_{term} = 1.0, \lambda_{run} = 0.1, \lambda_{effort} = 1 \times 10^{-5}$ |
| Constraints | Method | Soft saturation ($\tanh(\cdot)$) |
| Surrogate | Model | Propagator DeepONet (Frozen) |
| | Inputs | 2D Coordinates $(x, y)$ + Basis Weights |

Table 28: 2D Heat equation: DeepONet MPC parameters.

| Category | Hyperparameter | Value / Note |
|---|---|---|
| Horizon | Prediction length $N_p$ | 10 |
| Optimizer | Algorithm | Adam (per step) |
| | Learning Rate | $2 \times 10^{-2}$ |
| | Steps per solve | 25 |
| Loss | Weights | $\lambda_{term} = 1.0, \lambda_{run} = 0.1, \lambda_{effort} = 1 \times 10^{-5}$ |
| Control | Strategy | Receding Horizon (Apply 1st action) |
| Surrogate | Model | Propagator DeepONet (Frozen) |

Table 29: Burgers' equation: simulation parameters and model/training hyperparameters. *LSTM*(Hochreiter & Schmidhuber, 1997) denotes a Long Short-Term Memory network.

| Category | Hyperparameter | Value |
|---|---|---|
| PDE & Simulation | Domain length $L$ | 1.0 |
| | Final time $T$ | 4.0 |
| | Viscosity $\nu$ | 0.03 |
| | Weight bounds $[c_{\min}, c_{\max}]$ | $[-1.0,\ 1.0]$ |
| Discretization | Solver grid $(n)$ | 81 |
| | Solver steps $(N_t)$ | 201 |
| | Number of sensors $(N_x = n)$ | 81 |
| | Number of basis functions $(M)$ | 4 |
| PDE-OP's $\mathcal{Y}_\theta$ | Learning Rate | $1 \times 10^{-3}$ |
| | Number of Train Trajectories | 5000 |
| | Number of Test/Val Trajectories | 1000 |
| | Number of epochs | 2000 |
| | Optimizer | AdamW |
| | Batch size | 512 |
| | Activation function | RELU |
| | Branch net $(b)$: # layers | 3 |
| | Branch net $(b)$: hidden sizes | $[256, 256, 256]$ |
| | Trunk net $(\gamma)$: # layers | 5 |
| | Trunk net $(\gamma)$: hidden sizes | $[256, 256, 256, 256, 256]$ |
| | Feature dimension $(d)$ | 128 |
| PDE-OP's $\mathcal{U}_\omega$ | Learning rate | $1 \times 10^{-4}$ |
| | Number of epochs | 5000 |
| | Optimizer | AdamW |
| | Batch size = Number of Train Trajectories | 2048 |
| | Activation function | RELU |
| | LSTM: # layers | 2 |
| | LSTM: hidden sizes | $[256, 256]$ |

Table 30: Burgers' equation: Nonlinear MPC (NMPC) parameters. ($^*$) The basis function are used by each method.

| Category | Parameter | Value / Note |
|---|---|---|
| Horizon | Prediction length $N_p$ | e.g. 10 (receding horizon) |
| Dynamics | Implicit step (CN) | $\boldsymbol{y}(t_{k+1}) = \mathcal{G}(\boldsymbol{y}(t_k), \boldsymbol{c}(t_k))$ |
| Actuation | Basis$^*$ | $\sin\big((j+1)\pi x / X\big), j = 0, \dots, M-1$ |
| Solver | Nonlinear optimizer | SLSQP (single-shooting), box bounds; warm start by shift |
| CN inner | Newton solve per step | tolerance $10^{-10}$, max iters 25 |

Table 31: Burgers' equation: Adjoint method parameters.

| Category | Parameter | Value / Note |
|---|---|---|
| Horizon | Prediction lentgh $N_p$ | 10 |
| Forward | Implicit CN dynamics | as in NMPC, Dirichlet enforced each step |
| Jacobian | Semi-discrete residual | $J(\boldsymbol{y}) = -\boldsymbol{D}\operatorname{diag}(\boldsymbol{y}) + \nu\,\boldsymbol{D}^2$ |
| Adjoint | Backward sweep | $A_k^\top \boldsymbol{q}_k = \Delta t\,(\boldsymbol{y}(t_{k+1}) - \boldsymbol{y}_{\text{target}});$ |
| | | $\boldsymbol{p}_k = \Delta t\,(\boldsymbol{y}(t_k) - \boldsymbol{y}_{\text{target}}) - B_k^\top \boldsymbol{q}_k$ |
| Gradient | Per-step control gradient | $\nabla_{\boldsymbol{u}(t_k)} J = \Delta t\big(\alpha\,\boldsymbol{u}(t_k) + \boldsymbol{B}^\top \boldsymbol{q}_k\big),$ |
| Optimizer | Outer optimizer | L-BFGS-B, warm start |

Table 32: Burgers' equation: SBTO (Surrogate-Based) method parameters.

| Category | Hyperparameter | Value / Note |
|---|---|---|
| Optimizer | Algorithm | Adam |
| | Learning Rate | $1 \times 10^{-2}$ |
| | Max iterations | 150 |
| Loss | Weights | $\lambda_{term} = 1.0, \lambda_{run} = 0.1, \lambda_{effort} = 1 \times 10^{-5}$ |
| Constraints | Method | Soft saturation ($\tanh(\cdot) \times c_{\max}$) |
| Surrogate | Model | Propagator DeepONet (Frozen) |
| | Rollout | Recurrent (Full Horizon $N_t$) |

Table 33: Burgers' equation: DeepONet MPC parameters.

| Category | Hyperparameter | Value / Note |
|---|---|---|
| Horizon | Prediction length $N_p$ | 10 |
| Optimizer | Algorithm | Adam (per time step) |
| | Learning Rate | $2 \times 10^{-2}$ |
| | Steps per solve | 25 |
| Loss | Weights | $\lambda_{term} = 1.0, \lambda_{run} = 0.1, \lambda_{effort} = 1 \times 10^{-5}$ |
| Control | Strategy | Receding Horizon (Apply 1st action) |
| Surrogate | Model | Propagator DeepONet (Frozen) |

Table 34: Burgers' equation with varying coefficient: simulation parameters and model/training hyperparameters. *LSTM* (Hochreiter & Schmidhuber, 1997) denotes a Long Short-Term Memory network.

| Category | Hyperparameter | Value |
|---|---|---|
| PDE & Simulation | Domain length $L$ | 1.0 |
| | Final time $T$ | 4.0 |
| | Viscosity $\nu(x)$ range | $[0.01, 0.05]$ |
| | Correlation length (GRF) | 0.3 |
| | Control scale | 2.0 |
| Discretization | Solver grid $(N_x)$ | 81 |
| | Solver steps $(N_t)$ | 401 |
| | Number of sensors $(N_{\text{sens}})$ | 81 |
| | Number of basis functions $(M)$ | 4 |
| PDE-OP's $\mathcal{Y}_\theta$ | Learning Rate | $1 \times 10^{-3}$ |
| | Number of Train Trajectories | 1000 |
| | Number of Test/Val Trajectories | 200 |
| | Number of epochs | 2000 |
| | Optimizer | AdamW |
| | Batch size | 512 |
| | Activation function | RELU |
| | Branch net $(b)$: # layers | 3 |
| | Branch net $(b)$: hidden sizes | $[256, 256, 256]$ |
| | Trunk net $(\gamma)$: # layers | 5 |
| | Trunk net $(\gamma)$: hidden sizes | $[256, 256, 256, 256, 256]$ |
| | Feature dimension $(d)$ | 256 |
| PDE-OP's $\mathcal{U}_\omega$ | Learning rate | $1 \times 10^{-3}$ |
| | Number of epochs | 5000 |
| | Optimizer | Adam |
| | Batch size = Number of Train Trajectories | 1024 |
| | Activation function | RELU |
| | LSTM: # layers | 2 |
| | LSTM: hidden sizes | $[256, 256]$ |

Table 35: Burgers' equation with varying coefficient: Adjoint method parameters.

| Category | Parameter | Value / Note |
|---|---|---|
| Horizon | Prediction length $N_p$ | 10 |
| Forward | Integrator | Implicit Crank–Nicolson (Newton solver) |
| | Operator $\mathbf{L}_\nu$ | Spatially varying viscosity $\nu(x)$ |
| Adjoint | Backward recursion | Linearized adjoint equation |
| | | $\boldsymbol{p}_k = \Delta t(\boldsymbol{y}_k - \boldsymbol{y}_{\text{ref}}) - \mathbf{B}_k^\top \boldsymbol{q}_k$ |
| Gradient | Per step | $\nabla_{\boldsymbol{u}} J = \Delta t(\alpha \boldsymbol{u}_k + \mathbf{B}^\top \boldsymbol{q}_k)$ |
| Optimizer | Algorithm | L-BFGS-B, warm start |
| Stopping | Max iters / tolerances | 40; `ftol` $= 10^{-6}$ |

Table 36: Burgers' equation with varying coefficient: SBTO (Surrogate-Based) method parameters.

| Category | Hyperparameter | Value / Note |
| --- | --- | --- |
| Optimizer | Algorithm | Adam |
| | Learning Rate | $1 \times 10^{-2}$ |
| | Max iterations | 150 |
| Loss | Weights | $\lambda_{term} = 1.0, \lambda_{run} = 0.1, \lambda_{effort} = 1 \times 10^{-5}$ |
| Constraints | Method | Soft saturation ($\tanh(\cdot) \times 2.0$) |
| Surrogate | Model | Propagator DeepONet (Frozen) |
| | Inputs | Viscosity profile $\nu(x)$ included |

Table 37: Burgers' equation with varying coefficient: DeepONet MPC parameters

| Category | Hyperparameter | Value / Note |
| --- | --- | --- |
| Horizon | Prediction length $N_p$ | 10 |
| Optimizer | Algorithm | Adam (per time step) |
| | Learning Rate | $2 \times 10^{-2}$ |
| | Steps per solve | 25 |
| Loss | Weights | $\lambda_{term} = 1.0, \lambda_{run} = 0.1, \lambda_{effort} = 1 \times 10^{-5}$ |
| Control | Strategy | Receding Horizon (Apply 1st action) |
| Surrogate | Model | Propagator DeepONet (Frozen) |