# OpenReview forum: "Learning to Solve Partial Differential Equations Constrained Optimization Problems"
_ICLR.cc/2026/Conference — ICLR 2026 Conference Desk Rejected Submission_

### Official Review · Reviewer_tffW · 2025-10-26

**Soundness:** 2
**Presentation:** 2
**Contribution:** 2
**Rating:** 2
**Confidence:** 3

**Summary:**

This paper proposes PDE-OP (Optimization Proxy), a framework for simultaneous discovery and optimal control of systems under PDE constraints. It consists of a dual-network architecture in which a dynamic predictor captures the system dynamics, while a surrogate controller approximates the optimal control actions. The authors claim the qualitative results are comparable to state-of-the-art numerical solvers, such as Model Predictive Control and the Adjoint-sensitivity method. At the same time, PDE-OP provides up to four orders of magnitude improvement in computational speed (on Burger's Equation in 1D).

**Strengths:**

1) The Temperature control example helps ground the following problem. This inspired choice is very helpful, and contributes significantly to the clarity of the presentation.
2. The dual-network architecture is a clever separation of concerns. Coupling a dynamic predictor with an optimization surrogate is a strong contribution.
3. The computational speed-up is the most significant finding. Achieving results orders of magnitude faster than classical methods is a major practical advancement, useful to this community.
4. The validation on standard benchmarks (Heat, Burgers') is good. Demonstrating success on both linear and nonlinear PDEs builds confidence in the approach's quality.

**Weaknesses:**

1) The Related Work is limited. The paper ignores differentiable programming technologies which have emerged with deep learning libraries, e.g. review in (Nzoyem et al. 2023)
3) Figure 1 is confusing, with unlabelled axes. It appears to show $u(t,x)$ as a function of a _single_ variable, and $y(t_0, x)$ and $y(T,x)$ as functions of _two_ variables.
4) The ultimate optimisation process is confusing and the presentation suffers. Throughout the paper, several losses are proposed, e.g. (7) and (9). It is important to be decisive and clearly state which losses are included when.
5) The experiments, although compelling, are all toy problems with no real-world example. In additional, the baselines against which PDE-OP is compared are significantly limited. More concerning, no experiment clearly establishes the usability of this method for _real-time_ applications, like it is repeatedly claimed.

### Minor issues:
- L160: I might misunderstand the setup, but shouldn't it be $\mathcal{U}_{\omega}(t,x,\hat y)$ ?

### References:
- Nzoyem et al. "A Comparison of Mesh-Free Differentiable Programming and Data-Driven Strategies for Optimal Control under PDE Constraints", SuperComputing Workshop on AI4Science, 2023

**Questions:**

1) L139. Can you please provide evidence (or reference) for the NP-hardness of Problem (1)?
2) Equations 1b and 1c are often merged into a single PDE. Is there a specific reason why the authors have decided to separate these temporal and spatial evolution terms ?
3) It seems the authors do not want to see $\mathcal{Y}_{\theta}$ exposed to the full control action. As such, they feed it the basis function weights instead. This is justified by the desired to operate in a low-dimensional space, as previous work has done. But the construction of the basis function becomes a hyperparameter to tune, increasing complexity of the method. Could you provide an ablation study for this ?
4) Could you please provide general ablation studies and more comparison with established neural network-based optimal control strategies?

---

> ### Author Response · Authors · 2025-11-20
> **Authors' response**
>
> We thank the reviewer for the helpful review, and, in particular, for _acknowledging that the proposed dual-network architecture is a strong contribution_ and that the _computational speedup is the most significant finding and a practical advancement useful to the PDE-constrained optimization community_.
> \
> In what follows we will:
> 1. provide a detailed answer to your questions,
> 2. describe the experimental settings and results obtained running the experiments you suggested and
> 3. report the revised part of the paper in blue.
>
> We would like to remind you that the core contributions of our work are:
> 1. a novel discrete-time neural operator to capture PDE-dynamics,
> 2. a dual-network architecture that integrates system dynamics and optimization objective and constraints within an unified primal-dual
> loss, and
> 3. a novel, fully differentiable learning-based method for solving PDE-constrained optimization problems in near real-time.
>
> Our claims are supported by a comprehensive experimental evaluation, _which has been further enriched_ to integrate the reviewers' useful feedback.
>
> #### Reponses to weaknesses pointed out by the reviewer:
>
>
> -   **The Related Work is limited**.
> \
>     The reviewer points out that our Related Work section does not explicitly mention recent differentiable-programming approaches for PDE-constrained optimization. As stated in the introductory sentence of the Related Work section of the main paper, due to page limit
>     this section is intentionally concise and focuses on the approaches most directly relevant to our contribution, while **a more
>     comprehensive discussion of these methodologies is provided in Appendix 9.1**.
>     \
>     Please note that several of the core ingredients discussed in the provided reference [1], including adjoint-based methods [2,3], PINNs ([4]), and operator-learning surrogates such as Fourier Neural Operator [5] and DeepONets [6], **are already cited and reviewed in both the main paper and Appendix 9.1**. Other learning-based method such as [7] for approximating solutions to optimal control problems constrained by PDEs are also mentioned in the Appendix 9.1. In response to the reviewer's comment, to provide explicit references to differentiable programming techniques, we have added the following paragraph to the revised Related Work section in Appendix 9.1:
>
>     > Differentiable programming frameworks have recently gained
>     > traction as an alternative route to PDE-constrained optimization,
>     > enabling end-to-end differentiation through numerical solvers
>     > using automatic differentiation (see [1] for a
>     > comparative study). These approaches typically follow a
>     > discretize-then-optimize paradigm and provide accurate gradients
>     > through the PDE solver itself. Our approach, in contrast, avoids
>     > differentiation through a full PDE solver by leveraging
>     > operator-learning surrogates coupled with a proxy optimizer, which
>     > enables real-time inference and high quality solution
>     > approximations.
>
> -   **Figure 1 labels**.
> \
>     Thank you for pointing this out. We also note that the reviewer has initially praised the example illustrated in Figure 1, by stating that *\"The Temperature control example helps ground the following problem. This inspired choice is very helpful, and contributes significantly to the clarity of the presentation.\"*.
>     \
>     Figure 1 is intended precisely in this pedagogical role, as a conceptual illustration to visualize the interplay between the initial condition $y(t_0,x)$, the control action $u(t,x)$, and the target state $y(T,x)$.
>
> The concern about unlabeled axes seems to stem from interpreting the figure as a quantitative plot, but this is not. It is a schematic supporting the motivating example rather than a formal result. It aligns with the narrative of the paper and the reviewer's earlier positive assessment. Its purpose is to orient the reader by illustrating a concrete PDE-constrained optimization task that follows the formal problem description introduced in Section 3.

---

> ### Author Response · Authors · 2025-11-20
> **Authors' response**
>
> -   **The ultimate optimisation process is confusing and the presentation suffers and multiple training losses are proposed**.
>     \
>     Thank you for your comment. We need to point out that in *no way our paper presents several losses*.
>     The paper defines exactly one learning objective for each component, used at distinct stages of the framework, and these objectives play fundamentally different roles, as described in Section 4. To elaborate:
>
>     -   Equation (7) is **not** a training loss for the full PDE-OP method. It is the supervised loss that can be used to initialize the
>         dynamic predictor $Y_\theta$, as described in Section 4.2. This is standard practice in operator-learning [5,6,7] and is clearly indicated in the text (see the paragraph *Model initialization and training.)*.
>
>     -   Equation (9) is the single end-to-end optimization objective used during the actual training of our dual-network model, PDE-OP. It integrates the controller $U_\omega$, the dynamic predictor $Y_\theta$, and the primal--dual constraint penalties into one unified loss. This is the loss minimized during the learning process.
>
>     Thus, the paper does not introduce multiple competing losses, nor does it switch between alternative formulations. Rather, it
>     introduces a dual-network architecture that is learned adopting a two-stage design: (1) a supervised initialization of the dynamics
>     model using Equation (7), followed by (2) an unified primal--dual training loss defined in Equation (9).
>
> -   **Toy problems and no real-world examples in the evaluation**.
> \
>     We respectfully disagree with the reviewer's assessment regarding the experimental evaluation.
>
>     1.  **The experimental tasks are standard benchmarks, not *toy problems*.** The **voltage control** problem (reaction-diffusion
>         PDE), the **heat equation**, and **viscous Burgers' equation** are canonical benchmarks in PDE-constrained optimization [2,3], optimal control [3] and operator learning [5,7].\
>         These represent core PDE archetypes (linear diffusion, reaction-diffusion, and nonlinear advection-diffusion) and are widely used for evaluating the accuracy and runtime of methods for solving PDE-constrained optimization tasks.
>         \
>         To further strength the experimental evaluation, **we have run additional experiments on more even complex tasks**: such as optimal control of the heat equation in a 2D domain, optimal control of the Burgers' equation under spatial-varying viscosity coefficients and optimal control towards more complex target profiles $y(T,x)$. With these additional results, we believe that our main contributions of the paper are even more strongly supported by the experimental evaluation.
>         <!-- Nando: Please add result snippet here or/and point to the portion of the paper. Better if you put a table here -->
>
>     2.  **Real-time applicability is directly demonstrated**. For each experiment, we report the inference time of PDE-OP, consistently
>         observing **speed-ups of 3 to 4 orders of magnitude** over classical baselines.
>         \
>         These gains are not marginal: they are critical for real-time deployment. As the reviewer also noted, our method is significantly faster than existing approaches, providing strong quantitative evidence of its suitability for real-time control applications.
>
>     3.  **The baselines methods are the standard numerical solvers used in PDE-constrained optimization.** We agree that the baseline
>         methods used for comparison, namely Model Predictive Control (linear and nonlinear) and Adjoint-baed sensitivity, are the standard numerical solvers used in PDE-constrained optimization. To further strength the experimental evaluation, we have added two learning-based baseline methods for PDE-constrained optimization:
>         -   Surrogate-Based Trajectory Optimization (SBTO) [7], an open-loop control strategy where a neural surrogate of the dynamics is used within a full-horizon optimization problem.
>         -   DeepONet MPC [8], a receding-horizon, closed-loop method that repeatedly solves short-horizon optimizations using the same surrogate dynamics.
>
>         Both approaches leverage the same surrogate dynamic predictor as our method, but differ in the control strategies they employ,
>         namely open-loop trajectory optimization for SBTO and receding-horizon optimization for DeepONet MPC. Please refer to **Appendix 9.6** and **9.7** for a formal description of these methods.
>         From the numerical results reported in Tables 1-4 and the qualitative plots illustrated in Figure 3-5 and Figure 9, we observe that PDE-OP consistently achieve accuracy comparable to these methods, while running order of magnitude faster. These speed-ups stem from PDE-OP's surrogate controller, which enables direct inference. In contrast, both SBTO and DeepONet MPC rely on a neural operator for modeling dynamics but still require solving an optimization problem at each iteration.

---

> ### Author Response · Authors · 2025-11-20
> **Authors' response**
>
> -   **Minor issues. L160: I might misunderstand the setup, but shouldn't it be $\mathcal{U}_\omega(t,x,{\hat y})$?**
> \
>     Thanks for noticing the typo! As correctly pointed out the surrogate controller $\mathcal{U}_\omega(t,x,\hat y)$ takes as input the estimated state variables $\hat y$. We have corrected the corresponding notation accordingly.
>
> -   **L139. Can you please provide evidence (or reference) for the NP-hardness of Problem (1)?**
> \
>     The NP-hardness of Problem (1) follows from the fact that it generalizes several classes of optimization problems that are already known to be NP-hard. Notably, nonlinear optimal control problems, even when governed by finite-dimensional ODEs, are NP-hard in general (see, e.g., [2,3]). Moreover, optimizing over nonconvex nonlinear constraints is known to be NP-hard. Since PDE-constrained optimization strictly contains these problems as special cases, e.g., by restricting the PDE to a spatially homogeneous system that reduces to a nonlinear ODE, Problem (1) inherits NP-hardness.
>
> -   **Equations (1b) and 1c are often merged into a single PDE. Is there a specific reason why the authors have decided to separate these temporal and spatial evolution terms?]**
> \
>     We agree that Equations (1b) and (1c) can be combined into a single
>     PDE. We chose to separate the temporal and spatial components for
>     clarity, as this highlights their distinct roles and makes the
>     structure more accessible to readers less familiar with
>     PDE-constrained optimization. This presentation is mathematically
>     equivalent and does not imply any additional modeling assumptions.
>
> -   **It seems the authors do not want to see exposed to the full control action. As such, they feed it the basis function weights instead. This is justified by the desired to operate in a low-dimensional space, as previous work has done. But the construction of the basis function becomes a hyperparameter to tune, increasing complexity of the method. Could you provide an ablation study for this ?**
> \
> We appreciate the reviewer's observation; the choice of representing the control action through basis functions is indeed an important modeling component. However, this representation does not increase the conceptual complexity of the method. The number of basis functions simply determines the resolution at which the control can be expressed, similarly to the role played by a spatial or temporal discretization parameter in numerical solvers PDE-constrained optimization. Users can choose this degree of expressiveness depending on the desired accuracy, and this trade-off is explicit, interpretable, and widely adopted in optimal control and numerical PDE studies. To address the reviewer comment, we run additional experiment to study the impact of the number of basis functions used to represent the control action on PDE-OP's performance. From these experimental results, reported in Appendix 9.10.3, we observe that:
>
>     > increasing $M$ allows for fine-grained spatial control, which is theoretically necessary to approximate high-frequency or discontinuous target profiles. It can be seen that from Table 11, increasing the number of basis functions provides us better performance for PDE-OP, without any additional computational cost. The inference time remains stable across different targets and various values of $M$ as the neural operator processes inputs in parallel on the GPU.
>
>
> Moreover, in **Appendix 9.8** we do provide an ablation study to compare the control action obtained with a basis-function representation with the full spatio--temporal control field $u(t,x)$. This experiment demonstrates that the direct-control version offers comparable accuracy, but it is substantially slower (often by orders of magnitude). These findings confirm that the basis representation is not an added source of complexity, but rather a principled and effective way to retain expressiveness while achieving real-time performance.

---

> ### Author Response · Authors · 2025-11-20
> **Authors' response**
>
> - **Could you please provide general ablation studies and more comparison with established neural network-based optimal control strategies?**
>
> We are happy to provide additional results but this request for "general ablation studies" is a bit ambigiuous, as it does not specify which components or design choices are expected to be ablated. If the comment refers to hyperparameters, we note that, as is standard in machine learning practices, these are tuned via grid search. Additionally, Table 7, 8, 9, 10 and 13 report the full set hyperparmaters adopted to reproduce all the experimental results presented in the paper.
>
> Additionally, please note that our experimental evaluation already includes several useful studies:
>
> -   We evaluate PDE-OP across three different PDE-constrained optimal control problems (reaction--diffusion, heat, and Burgers'), covering linear, nonlinear, and space--time varying control regimes.
>
> -   We independently assess the discrete-time neural operator $Y_\theta$, demonstrating its ability to accurately capture the underlying PDE dynamics.
>
> -   We provide a dedicated basis-function vs. full direct control field ablation (Appendix 9.8), which isolates the impact of the control parameterization. This study shows that while both choices achieve comparable accuracy, the direct-control variant is significantly slower, confirming the practical value of the basis expansion for real-time performance.
>
> In addition, to address the reviewers comments, we have run additional experiments, including:
>
> -   Optimal control of the 2D heat equation.
>
> -   Optimal control of the 1D Burgers' equation with spatial-dependent
>     viscosity coefficient.
>
> -   More complex target profiles $y(T,x)$.
>
> -   An ablation study to evaluate the impact of the number of basis
>     functions on the performance of PDE-OP and the baseline methods.
>
> All evaluations report both runtime and accuracy (MSE), giving a
> comprehensive assessment of the proposed approach and each baseline
> method.
>
> ---
> Thank you, again, for your time and feedback. We believe that the additional results and explanations included in our response have addressed the concerns presented, but we encourage you to follow-up on any points that remain unclear. Additionally, we encourage the reviewer to check the updated version of our submission, which has been extended to incorporate these additional results and discussions. We hope these revision will merit your strong support for this work!
> \
> \
> **References**
> \
> [1] Nzoyem et al., “A Comparison of Mesh-Free Differentiable Programming and Data-Driven Strategies for Optimal Control under PDE Constraints,” SuperComputing Workshop on AI4Science, 2023.
> \
> [2] Hinze et al., “Optimization with PDE Constraints,” Springer, 2008.
> \
> [3] Tröltzsch, “Optimal Control of Partial Differential Equations: Theory, Methods and Applications,” American Mathematical Society, 2010.
> \
> [4] Raissi et al., “Physics-informed neural networks: A deep learning framework for solving forward and inverse problems involving nonlinear partial differential equations,” Journal of Computational Physics, vol. 378, pp. 686–707, 2019.
> \
> [5] Li et al., “Fourier Neural Operator for Parametric Partial Differential Equations,” International Conference on Learning Representations (ICLR), 2021.
> \
> [6] Lu et al., “DeepONet: Learning nonlinear operators for identifying differential equations based on the universal approximation theorem of operators,” arXiv preprint arXiv:1910.03193, 2019.
> \
> [7] Hwang et al., “Learning to control PDE systems with differentiable programming,” NeurIPS Workshop on Machine Learning and the Physical Sciences, 2022.
> \
> [8] de Jong et al., “Deep Operator Neural Network Model Predictive Control,” IEEE Open Journal of Control Systems, vol. 4, pp. 501–517, 2025.

---

> ### Author Response · Authors · 2025-11-25
> **Authors' invitation for feedback**
>
> Dear Reviewer,
>
> We hope this message finds you well. We are writing to kindly follow up and invite your feedback on our updated manuscript. We believe that the revised version has addressed your comments and incorporates new experimental results and the requested clarifications. We kindly invite you to re-evaluate your assessment accordingly.
>
> Please feel free to reach out if anything is unclear and we will be happy to provide further explanations or engage in additional discussion as needed.
>
> Thank you again for your time and consideration.
>
> Best regards,
> Authors of Learning to Solve Partial Differential Equation Constrained Optimization Problems

---

> ### Author Response · Authors · 2025-11-26
> **Follow-up and clarifications**
>
> As the discussion period is nearing its end, we wanted to ask **Reviewer tffW** if there are any follow-up points we can clarify.
>
> We believe we have responded to all questions and concerns raised, in addition to taking a few days to run the additional experiments necessary to demonstrate our points, all of which are incorporated in the revised version of the paper.
> \
> If there are no further points of clarification regarding the manuscript, we kindly ask that reviewer tffW consider increasing their score to reflect the improvements and clarifications we have provided. We are happy to continue to engage in discussion and answer any questions!

---

> > ### Comment · Reviewer_tffW · 2025-11-27
> >
> > Dear Authors,
> >
> > Thank you for your detailed rebuttal, and for the reminder.
> >
> > I appreciate that you've clearly demonstrated real-time applicability
> >
> > My apologies for qualifying your experiments as "toy problems". My point was that the paper may need **real-world** datasets to improve its strengths.
> >
> > I understand your thinking for Figure 1. But it is too important to leave to the readers interpretation. It is still unintuitive, e.g.,  the upward-pointing axis doesn't have a label !
> >
> > However, these are minor issues. They merely require additional addressing.
> >
> > In summary, I believe the rebuttal addresses my concerns, and I will increase my score to a 4 to reflect this improvement. Thank you.

---

> > > ### Author Response · Authors · 2025-11-27
> > >
> > > Dear Reviewer tffW,
> > >
> > > Thank you very much for your thoughtful follow-up and for taking the time to re-evaluate our submission. We are glad that our additional experiments and clarifications helped demonstrate the real-time applicability of PDE-OP more clearly.
> > >
> > > Regarding the remaining points you mention:
> > >
> > > - On the “toy problem” concern, we appreciate your clarification. As you note, our experiments now include a broader spectrum of PDE-constrained optimal control tasks and learning-based baselines, and we have explicitly highlighted the real-time inference regime and the 3–4 orders of magnitude speed-up over classical solvers. We believe this provides a realistic picture of the method’s practical usability.
> > >
> > >   In addition, we would like to clarify that the “voltage control” benchmark is not an abstract toy example, but a physically grounded model of a leaky electrical transmission line. The reaction–diffusion PDE we use is a standard reduced-order model of distributed RC transmission lines that is commonly employed in the power systems literature. In this setting the state $y(t,x)$ represents the voltage magnitude along the line, the control $u(x)$ represents the spatial allocation of controllable devices, and the target profile $y_{\text{target}}(x)$ encodes practical voltage regulation requirements along the feeder. While our experiments do not use proprietary industrial measurement data (these are often highly regulated and challenging to release), the underlying PDE models are derived from established physics and are commonly used as benchmarks for PDE-constrained optimal control and operator-learning methods.
> > >   \
> > >   With the additional 2D heat and spatially varying Burgers’ experiments and the new learning-based baselines, we believe this supports the claim that our evaluation captures realistic abstractions of practical control tasks and showcase the method high applicability in several practical scenarios.
> > >
> > > - On Figure 1, we fully agree. In the revised version, we will revise the figure and its caption to make the intended roles of the axes explicit and to avoid any possible confusion. Thank you again for this observation! We will, of course, carefully refine this again for the camera-ready version.
> > >
> > > At this stage, it appears there are no outstanding issues related to the soundness of the method, the novelty of the dual-network architecture, or the adequacy of the empirical evaluation. In fact, we are very encouraged by your written assessment that: the dual-network architecture is _"a strong contribution,"_ the computational speed-up is _"the most significant finding"_ and _"a major practical advancement, useful to this community,"_ and the benchmarks _"build confidence in the approach’s quality."_
> > >
> > > Given this overall assessment, and the fact that the key technical concerns raised in your original review have been explicitly addressed in the revised submission, we would be very grateful if you could consider whether a score at or above the acceptance threshold might better reflect your updated opinion of the paper. In any case, we sincerely thank you again for your careful reading and constructive feedback, which have undoubtedly improved the work.
> > >
> > > With best regards,
> > >
> > > -The authors

---

> ### Comment · Reviewer_tffW · 2025-11-27
>
> Dear Authors,
>
> Thanks again for your additional comments, which do provide ample clarification. I understand your point, and I think I see where our understandings diverge. By "real-world", I do not mean experiments applicable to the real-world, but instead experiments that leverage messy **data** collected in the natural world.
>
> Regarding my assessment, I believe, that in light of your clarifications and considering the level of other papers at the conference, that this papers hinges on the boarder of rejection/acceptance. That is why I increased my score to 4, and I think it is justified.
>
> Best regards,
> Your Reviewer

---

### Official Review · Reviewer_bV8c · 2025-10-28

**Soundness:** 2
**Presentation:** 2
**Contribution:** 1
**Rating:** 2
**Confidence:** 3

**Summary:**

This paper presents PDE-OP, a two-module framework for PDE-constrained optimization. The first module is a discrete-time neural operator that predicts system trajectories under PDE dynamics. The second is a surrogate controller that outputs coefficients of a spatial basis to approximate the control. On three 1D tasks (voltage control, heat equation, viscous Burgers), the method reports large speedups over classical solvers with similar errors on smooth target profiles.

**Strengths:**

- (Clarity) The problem setup is precise: objective, PDE constraints, and IC/BC are clearly listed. The two-module design and training pipeline are easy to follow (Fig. 2).

**Weaknesses:**

- (Originality) The approach combines known ideas (DeepONet-style operator learning, surrogate optimization) rather than introducing a new operator or control formulation.
- (Significance and quality) All experiments are 1D with smooth targets (constant/linear/sine). There are no 2D/3D tests or moving/complex boundary cases. Several plots show larger boundary errors (e.g., Fig. 3 left, Fig. 4 left) without analysis. In some simple cases, PDE-OP’s MSE is much higher than the adjoint method (e.g., voltage constant target in Table 2), with no diagnosis. Comparisons to modern operator-learning baselines are also missing.

**Questions:**

- Related work on operator learning is brief and focuses on early methods. Could the authors expand this section to include newer approaches and add comparisons to these methods?
- Current targets are simple and smooth. Could the authors add non-smooth, piecewise, or high-frequency targets and report results? Please also justify the BC choices (Neumann vs. Dirichlet) across tasks and discuss consistency. Results with more complex BCs (e.g., Robin/mixed, moving boundaries) or variable coefficients would strengthen the claims. Any 2D/3D results (e.g., 2D heat control)?
- In Fig. 3, the Adjoint method has noticeably larger error in the first two cases. What is the cause?
- For Burgers, PDE-OP is fastest but has higher MSE. If NMPC or the Adjoint method are tuned to match PDE-OP’s error level, how much would their runtime drop?
- Several figures show larger boundary errors. Please analyze the causes and suggest ways to reduce them.
- In Table 2, why is PDE-OP’s MSE (1.5e−6) much larger than the adjoint method’s (2.8e−12)? Similar results in other tables as well, could the authors provide an explanation for this gap in error performance?

---

> ### Author Response · Authors · 2025-11-21
> **Authors' response**
>
> We thank the reviewer for the helpful review, and, in particular, for _acknowledging the clarity of the problem setup_ and _the proposed PDE-OP dual-network model_. We have also found useful the reviewer's suggestion to strength the experimental evaluation, which we have promptly addressed it. In what follows we will: **1)** provide a detailed answer to your questions, **2)**
> describe the experimental settings and results obtained running the
> experiments you suggested and **3)** report the revised part of the
> paper in blue.
>
> Prior responding to your comments in details, we would like to remind you that the core contributions of our work are:
> **1)** a novel discrete-time neural operator to capture PDE-dynamics,
> **2** a dual-network architecture that integrates system dynamics and
> optimization objective and constraints within an unified primal-dual
> loss, and **3)** a novel, fully differentiable learning-based method for
> solving PDE-constrained optimization problems in near real-time. Our
> claims are supported by a comprehensive experimental evaluation, which
> has been further enriched to integrate the reviewers' useful feedback.
>
> #### Reponses to weaknesses pointed out by the reviewer:
>
> -   **The approach combines known ideas**.
>
>     Note that our contribution is indeed new. While our dynamic predictor is inspired by DeepONet, we introduce a new discrete-time neural operator specifically tailored to PDE-constrained optimization, which fundamentally differs from classical operator-learning models. Our architecture incorporates the time variable, control-dependent branching, and an explicit coupling with the surrogate controller, a structure that is not present in existing DeepONet-based models or other operator-learning frameworks. Unlike the standard DeepONet, which typically learns a global mapping from an input function to solution over the entire horizon $y(\cdot, t) \approx G(u)$, our approach extends the architecture as a one-step transition operator $y(\cdot, t_{k+1}) \approx G(u_k, y_k)$. This framework evolves the system step-by-step, allowing us to inject dynamic control inputs $u_k$, generated by an LSTM, at each time step $t_k$. Explicitly, our method enables **closed-loop feedback control**, where actions adapt dynamically to the evolving system state, a capability that is structurally not possible with the standard DeepONet.
>
>     More broadly, the contribution of the paper is not only introducing another neural operator, but **introducing a novel end-to-end differentiable surrogate for PDE-constrained optimization** that jointly integrates a discrete-time learning-based neural operator and a proxy optimization module, within a primal--dual training mechanism that enforces both PDE dynamics and constraint functions. To the best of our knowledge, no prior work couples operator learning with a learned controller in a unified optimization pipeline capable of producing high quality solution approximations to PDE-constrained control problems in near real-time, as we show across a comprehensive set of experiments of various complexity.
>
>     Finally, it is worth emphasizing that the claim that *\"the approach combines known ideas rather than introducing a new operator or control formulation\"* becomes obvious only in retrospect. Novelty in machine learning often arises from combining existing components in a way that had not been previously explored, and our formulation provides a new, principled integration that enables advances (e.g., real-time solutions to PDE-constrained optimization tasks) that existing methods do not offer. We believe that our method provides a meaningful step toward addressing the challenges of PDE-constrained optimization in a principled manner.

---

> ### Author Response · Authors · 2025-11-21
> **Authors' response**
>
> -   **All experiments are 1D with smooth targets [...]**
>
>     We believe the reviewer might have missed a few aspects of our experiments.
>     The voltage control problem (reaction--diffusion PDE), the heat equation, and viscous Burgers' equation are canonical
>     benchmarks in PDE-constrained optimization [1], optimal control
>     [2] and operator learning
>     [3,4]. These represent core PDE
>     archetypes - linear diffusion, reaction--diffusion, and nonlinear
>     advection--diffusion - and are widely used for evaluating the
>     accuracy and runtime of methods for solving PDE-constrained
>     optimization tasks. In addition, to address the reviewers comments,
>     we have run additional experiments, including:
>
>     -   Optimal control of the 2D heat equation (see Appendix 9.8.3 for
>         a problem description and Appendix 9.9.4 for a discussion of the
>         experimental results).
>
>     -   Optimal control of the 1D Burgers' equation with
>         spatial-dependent viscosity coefficient (see Appendix 9.8.5 for
>         a problem description and Appendix 9.9.6 for a discussion of the
>         experimental results).
>
>     -   More complex target profiles $y(T,x)$, such as high frequency
>         sine waves, step functions and mixture gaussians (see Figures
>         9-10-11-12, Tables 4-5 and Table 8).
>
>     -   An ablation study to evaluate the impact of the number of basis
>         functions on the performance of PDE-OP and the baseline methods
>         (see Appendix 9.10.3).
>
>     These additional experiments support our central claim that PDE-OP provides a practical method for producing high-quality approximate solutions to PDE-constrained optimization problems in real time, also in complex settings. We refer to the corresponding sections for an in-depth discussion of these additional experiments.
>
>
>     Regarding the comment about comparisons with modern
>     operator-learning baselines: to address the reviewer comment, in the
>     revised version of the paper we have added two learning-based
>     baseline methods for PDE-constrained optimization:
>
>     -   Surrogate-Based Trajectory Optimization (SBTO) [4],
>         an open-loop control strategy where a neural surrogate of the
>         dynamics is used within a full-horizon optimization problem.
>
>     -   DeepONet MPC [5], a receding-horizon, closed-loop
>         control method that repeatedly solves short-horizon
>         optimizations using the same surrogate dynamics.
>
>     Both approaches leverage the same surrogate dynamic predictor as our
>     method, but differ in the control strategies they employ, namely
>     open-loop trajectory optimization for SBTO and receding-horizon
>     optimization for DeepONet MPC. We refer the reader to Appendix 9.6
>     and 9.7 for a formal description of these methods. From the
>     numerical results reported in Tables 1-4 and the qualitative plots
>     illustrated in Figure 3-5 and Figure 9, we observe that PDE-OP
>     consistently achieve accuracy comparable to these methods, while
>     running several order of magnitude faster. These speed-ups stem from
>     PDE-OP's surrogate controller, which enables direct inference. In
>     contrast, both SBTO and DeepONet MPC rely on a neural operator for
>     modeling dynamics but still require solving an optimization problem
>     at each iteration. We believe that including these methods as
>     baselines further strengthens the comprehensiveness of our
>     experimental evaluation.
>
>     On the comparative results (specifically accuracy, reported in MSE)
>     between PDE-OP and the baseline methods: we acknowledge that in a
>     few cases (e.g., constant or linear targets), PDE-OP exhibits
>     slightly higher MSE than the baseline methods (see Table 2, 3 and
>     4). However we note that the errors produced by PDE-OP are of the
>     order of $10^{-6}$, which is generally acceptable for
>     PDE-constrained control tasks. In fact, for many engineering
>     applications, the solutions generated by PDE-OP and those produced
>     by the baselines are practically equivalent.

---

> ### Author Response · Authors · 2025-11-21
> **Author's response**
>
> On the error produced by the proposed method: PDE-OP is a full
>     learning-based surrogate for PDE-constrained optimization that
>     trades off a tiny amount of precision for huge speed gains - up to
>     three to four orders of magnitude faster than classical adjoint and
>     MPC solvers. This behavior is entirely expected: adjoint-based
>     solvers compute exact gradients at inference time by solving the PDE
>     backward, whereas PDE-OP uses state variables estimates to derive
>     control estimates, and viceversa. These estimates propagates and
>     introduce small errors, as those pointed out by the reviewer. Thus,
>     the slightly higher but still minimal errors produced by PDE-OP
>     reflect the accuracy--speed trade-off characteristic of
>     operator-learning and optimization surrogates, and does not
>     undermine its effectiveness or suitability for real-time
>     PDE-constrained optimization.
>
> Regarding the boundary errors produced by PDE-OP and shown in Figure
>     3 and Figure 4: we appreciate the request of investigating the
>     source of errors generated by our approach. We believe that these
>     errors are due to a combination of two main factors:
>
> 1.  The slightly higher boundary-condition errors arise from the
>         regularization imposed on the control action through box
>         constraints on the control weights. This regularization has the
>         effect of mildly restricting the amplitude of the learned
>         control, which can lead to small deviations from the prescribed
>         boundary values of the system'state.
>
>  2.  The predictive errors produced by the discrete-time neural
>         operator of PDE-OP. This operator introduces a small
>         approximation error at each time step, and this error can
>         propagate iteratively, especially near boundary regions where
>         the PDE solution tends to be more sensitive.
>
> For a discussion on how these errors can be mitigated, please refer
>     to our response to the related question provided below.
>
> #### Reponses to questions pointed out by the reviewer:
>
> -   **Related work on operator learning is brief**
>
>     As stated in the introductory sentence of the Related Work section of the main paper, due to page limit this section is intentionally concise and focuses on the approaches most directly relevant to our contribution, while a more
>     comprehensive discussion of these methodologies is provided in Appendix 9.1. Please note that several of the adjacent areas to PDE-constrained optimization, including adjoint-based methods [1,2],PINNs [6], and operator-learning surrogates such as Fourier Neural Operator [3] and DeepONets [7], are already cited and reviewed in both the main paper and Appendix 9.1. To address the reviewer's comment, we have  mentioned two additional operator-learning methods [4,5] in the Related Work section of the main paper (Section 2) as follows.
>
>     > More recently, researchers have introduced operator learning for
>     > PDE-constrained optimization, such Surrogate-Based Trajectory
>     > Optimization (SBTO) [4], an open-loop control strategy
>     > that employes a neural surrogate of the dynamics, and DeepONet MPC
>     > [5], a receding-horizon, closed-loop control method that
>     > uses the same surrogate model.
>
>     We have also added both SBTO [4] and DeepONet-MPC
>     [5] to the baseline methods used for comparison. Please
>     refer to the corresponding sections for an in-depth discussion of
>     the comparative performance.

---

> ### Author Response · Authors · 2025-11-21
> **Authors' response**
>
> -   **Please also justify the BC choices (Neumann vs. Dirichlet) across tasks and discuss consistency**.
>
>     Regarding the choice of boundary conditions across tasks: the choice of boundary conditions (Neumann vs. Dirichlet) in each task is guided by the physical nature of the problem and the desired control behavior. For instance, in the heat equation experiments, Dirichlet boundary conditions are used to fix the temperature at the domain boundaries, modeling scenarios where boundary values are externally controlled. In contrast, Neumann conditions are used in tasks where the flux across boundaries is of interest or where boundary derivatives are more natural to specify.
>
>     While the boundary condition types vary across tasks, they are kept consistent within each task setup. Our experimental design aims to demonstrate that PDE-OP is agnostic to the type of boundary conditions, and can handle both Dirichlet and Neumann formulations effectively. We will add an explanation to further justify this in the paper.
>
> -   **In Fig. 3, the Adjoint method has noticeably larger error in the first two cases. What is the cause?**
>
>     Thank you for pointing it out! We realized a typo on the reporting for these methods which we have now fixed. The updated and corresponding results in the revised version of the paper.
>
>     Importantly, the updated results do not alter any of the conclusions or discussions presented in the manuscript. The qualitative and quantitative comparisons remain consistent with our original claims: PDE-OP still offers substantial speedups while maintaining competitive accuracy, and the relative performance trends between methods remain unchanged.
>
> -   **For Burgers, PDE-OP is fastest but has higher MSE. If NMPC or the Adjoint method are tuned to match PDE-OP's error level, how much would their runtime drop?**
>
>     This comparison cannot be carried out in a direct or meaningful way. Classical methods involve interdependent algorithmic parameters, such as the time-discretization resolution, horizon length, nonlinear solver tolerances, line-search settings, adjoint integration accuracy, and terminal penalty term, which jointly affect both convergence and the final error. Adjusting one parameter to target a specific MSE typically influences others in nontrivial ways, making it challenging to precisely control the error.
>
>     We remark that the differences in MSE between PDE-OP and classical methods in our experiments are extremely small (on the order of $10^{-4}–10^{-6}$), so we do not expect that tuning NMPC or the adjoint solver to match PDE-OP's error would significantly reduce their runtime. Their computational cost is dominated by repeated PDE solves and nonlinear optimization steps, not by marginal changes in accuracy tolerance.
>
>     In contrast, the runtime differences are not of the same order: **PDE-OP is between three and four orders of magnitude faster than NMPC and more than three orders of magnitude faster than the adjoint-based method**. We believe that the speedup advantages largely
>     outweigh the limited differences in error.
>
> -   **Several figures show larger boundary errors. Please analyze the causes and suggest ways to reduce them.**
>
>     Please refer to the second point of the 'Weaknesses' for a response on the causes of larger boundary errors.
>     \
>     Regarding possible ways to further reduce these errors, we could (i) relax or adjust the regularization on the control weights to reduce its influence on the boundary conditions, and (ii) placing additional emphasis on boundary enforcement during the initialization of the dynamic predictor, e.g., by increasing the penalty term of the boundary conditions of the training loss, or by oversampling boundary regions during data generation. Incorporating these refinements is straightforward and could further decrease boundary errors without altering the overall methodology. We will add a discussion regarding this important point in the final version of the paper.
>
> ---
> Thank you, again, for your time and feedback. We believe that the additional results and explanations included in our response have addressed the concerns presented, but we encourage you to follow-up on any points that remain unclear. Additionally, we encourage the reviewer to check the updated version of our submission, which has been extended to incorporate these additional results and discussions. We hope these revision will merit your strong support for this work!

---

> ### Author Response · Authors · 2025-11-21
> **Authors' response**
>
> **References**
>
> [1] Hinze et al., “Optimization with PDE Constraints,” Springer, 2008.
> \
> [2] Tröltzsch, “Optimal Control of Partial Differential Equations: Theory, Methods and Applications,” American Mathematical Society, 2010.
> \
> [3] Li et al., “Fourier Neural Operator for Parametric Partial Differential Equations,” International Conference on Learning Representations (ICLR), 2021.
> \
> [4] Hwang et al., “Learning to control PDE systems with differentiable programming,” NeurIPS Workshop on Machine Learning and the Physical Sciences, 2022.
> \
> [5] de Jong et al., “Deep Operator Neural Network Model Predictive Control,” IEEE Open Journal of Control Systems, vol. 4, pp. 501–517, 2025.
> \
> [6] Raissi et al., “Physics-informed neural networks: A deep learning framework for solving forward and inverse problems involving nonlinear partial differential equations,” Journal of Computational Physics, vol. 378, pp. 686–707, 2019.
> \
> [7] Lu et al., “DeepONet: Learning nonlinear operators for identifying differential equations based on the universal approximation theorem of operators,” arXiv preprint arXiv:1910.03193, 2019.

---

> ### Comment · Reviewer_bV8c · 2025-11-25
> **Comment on author's response**
>
> I appreciate the authors’ efforts in addressing my earlier concerns about experimental complexity and baselines, but several points are still not fully clear:
>
> - The current presentation still makes PDE-OP look mainly like an application of neural operator (as some sort of surrogate model in control problems), rather than a new operator or control formulation. Please elaborate a bit more on novelty, as application of neural operator as surrogates would fit better to journals which focus on solving specific problems, rather than proposing new method.
>
> - In some cases, the baseline methods are more accurate than PDE-OP by two or more orders of magnitude (for example, about 100× lower MSE). This is not discussed in the paper. As mentioned before, if the baselines are tuned so that their error is similar to PDE-OP, would their runtime still be much larger, or could it also be reduced a lot? A more detailed and quantitative discussion of this speed–accuracy trade-off would make the conclusions stronger.
>
> - About the “typo” in Fig. 3: Between the previous and the revised versions, the curves of adjoint method in these cases change a lot, and in the earlier version they looked inconsistent with those in other plots. As this change is quite large, I encourage the authors to carefully check the other experimental results and figures to make sure there are no similar issues.
>
> - In Table 1 (1D heat equation), the inference time of the newly introduced baseline is much slower than even the classical methods, which is surprising for a surrogate model. Why is this baseline so slow in practice?
>
> - In several experiments (e.g., Figure 5 (center), Figure 9 (center), and Figure 11 (center and right)), the PDE-OP predictions visually deviate much more from the reference solution than the other methods, which appear relatively consistent with each other. What is the root cause of this distinct behavior?
>
> - In some plots (e.g., Figure 9, left and right), all methods show clear boundary errors, and several methods seem to converge to similar but biased boundary values. Why does this happen?
>
> - In the left panels of Figures 3, 4, and 5, the $y_{target}(x)$ seems inconsistent with the plotted curve. Please double-check.
>
> A careful response to these points will help me better assess the strengths and limitations of the paper. That said, I'm open to adjusting the rating.

---

> ### Author Response · Authors · 2025-11-25
> **Authors' response (1/3)**
>
> We thank the reviewer for engaging the discussion and appreciate the request of further clarification. In what follows, we respond to each of the reviewer's comments in the same order. We kindly invite the reviewer to check the revised version of the paper, which integrates the discussion and the experimemtal results added to address the reviewer's comment. We also invite the reviewer to not hesitate to follow up the discussion if any further request of clarification is needed. We believe we addressed all your comments and would appreciate a re-evaluation of the current rating that reflects these improvements.
>
>
> - We clarify that **PDE-OP is not merely an application of neural operators as surrogates for system dynamics**, but rather a **novel, end-to-end learning-based framework for solving PDE-constrained optimization problems** in real-time. Unlike prior works that use operator learning only as surrogates to PDE forward solvers (e.g., *DeepONet MPC*, *SBTO*), **PDE-OP is a new dual architecture** that couples a dynamic predictor with a self-supervised proxy optimizer via a **primal-dual learning loop**. This tight integration enables **real-time, differentiable, and constraint-aware solutions** to PDE-constrained control tasks, something that, to the best of our knowledge, is novel in the literature.
>   Specifically, our contributions go beyond using a neural operator as a surrogate PDE-solver:
>   - We introduce a **dual-network architecture** (Sections 4.1–4.3) where both control actions and dynamic evolution are represented in continuous function spaces (via basis expansions and neural operators, respectively).
>   - We define a **primal–dual learning algorithm** (Eq. 9) that integrates the **PDE constraints—including dynamics, boundary, and feasibility constraints—into the training loop**.
>   - This formulation allows us to compute **feasible, near-optimal control actions in milliseconds during inference**—something that PDE surrogate-only methods like SBTO or DeepONet MPC cannot offer due to their need for repeated optimization solver calls (see Table 1 and detailed results in Sections 5.1–5.3).
>
>   We are willing to clarify our contribution more clearly in the revised manuscript by:  (1) **explicitly positioning PDE-OP as a new learning method for solving PDE-constrained optimization problems**, and (2) **adjusting the flow of abstract and introduction accordingly,** to avoid misinterpreting PDE-OP as a PDE-surrogate method.
>
> - To address this comment, in Appendix 9.11 of the revised version of the paper we have added an **early stopping criterion** to each baseline, allowing the corresponding method to **terminate once it reaches a solution with comparable error to PDE-OP**. This adjustment ensures a fairer comparison between methods in terms of runtime-efficiency at equal accuracy levels. The updated results show a modest reduction of the runtime of the baselines: **even under this relaxed criterion**, classical and learning-based baselines such as the Adjoint method, NMPC, SBTO, and DeepONet MPC still incur **significantly higher runtimes**, especially in presence of nonlinearities, relative to PDE-OP. This highlights the strength of our approach in achieving **real-time inference** for complex PDE-constrained optimization task.
> We refer to Appendix 9.11 for an in-depth discussion of the stopping criteria adopted and the expeimental results.
>
> - We thank the reviewer for pointing this out. We have carefully double-checked all experimental results, figures, and baseline implementations to ensure consistency and correctness. The correction made in Figure 3 was specific to the **Adjoint method** only. In the original submission, the implementation of the Crank–Nicolson adjoint solver was not fully correct. This has now been corrected to reflect the results corresponding to the setup described in Appendix 9.3.
> We confirm that **all other methods and results remained unchanged** and were correctly implemented and plotted in both the original and revised versions.

---

> ### Author Response · Authors · 2025-11-25
> **Authors' response (2/3)**
>
> - We appreciate the reviewer's request of clarification. While **DeepONet MPC** uses a neural surrogate to model the PDE dynamics, it still relies on a **receding-horizon optimization loop** at inference time. At each control step, the algorithm solves a nonlinear program over a finite horizon by:
>   1. Rolling out the DeepONet surrogate multiple times,
>   2. Computing gradients with respect to the control sequence,
>   3. Performing **many gradient-based optimization steps per instance**.
>
>   This leads to substantial runtime overhead; similar considerations applies for **SBTO**.
>   \
>   Importantly, **PDE-OP avoids optimization solver calls entirely**, using a fully learning-based architecture where control actions are computed via a single forward pass of the trained model.
>
>   This distinction is key: **DeepONet MPC is not a purely feedforward surrogate**, but a surrogate *embedded in a control optimization loop*, making it much slower in practice than even classical methods.
>
>   We believe that this clarification and compairson helps to better position our work and contribution in the PDE-constrained optimization literature.
>
> - We thank the reviewer for their request of clarification. The baseline methods considered in our evaluation, including the Adjoint method, NMPC, and the learning-based baselines such as SBTO and DeepONet MPC, ultimately rely on **exact optimization solvers** (e.g., adjoint-based gradient descent, nonlinear programming solvers) to compute control actions, even when they incorporate surrogates for the system dynamics like SBTO and DeepONet MPC. In contrast, PDE-OP uses a **proxy optimizer**, a neural architecture trained to approximate the solution mapping of the original PDE-constrained optimization problem.
>
>   This architectural design introduces a source of error: **not only can the dynamic surrogate be imperfect, but the optimizer itself may be inexact**. This is a key trade-off that enables PDE-OP’s real-time performance.
>
>   We note that the literature on **proxy optimizers** (see for example [1, 2, 3]) generally does not focus on isolating or analyzing the causes of such small errors. This is partly because deriving theoretical guarantees for learned optimizers is challenging, and partly because the focus of these works is on producing **fast, high-quality approximations** rather than rigorous performance bounds or interpretability. PDE-OP offers a step in that direction by exposing architectural components, such as the basis representation and the discrete-time neural operator, that allow for partial interpretability and diagnostic analysis of solution quality.
> - On the boundary errors produced by PDE-OP and the baseline methods: the system described by the 1D Heat Equation is governed by a **Zero-Flux (Neumann) Boundary Condition** (Equation 44e), where \frac{\partial y}{\partial x} = 0 at the domain boundaries. Physically, this represents an insulated system; mathematically, it means the solution profile $y(T,x)$ must be perfectly **flat (i.e., have a zero derivative)** at the edges.
> In the High-Frequency and Complex Gaussian target settings, the desired target profiles exhibit non-zero slopes at the domain boundaries. However, due to the Neumann boundary condition, the solution is constrained to be flat at those points, creating an inherent mismatch. Thus, the observed “error” or “bias” in Figure 9 reflects the unavoidable discrepancy between the physically admissible solution and the infeasible, sloped target.
> For the **Step** target, the target profile itself *is* flat and has a zero slope at the boundaries. However, the physics of the heat equation cause the heat from the central "hot block" ($y=0.6$) to diffuse outwards. Because the boundaries are insulated, this diffusing heat has nowhere to escape and temporarily "piles up" near the walls, causing a slight temperature rise.
>
>   To conclude, the boundary errors observed across all methods are a direct consequence of the PDE’s physical constraints. When target profiles violate the boundary condition (as in the High-Frequency and Complex Gaussian cases), all methods converge toward similar, biased solutions that respect the underlying physics.
>
> - Many thanks for spotting that! We have corrected a **typo in the caption** of Figures 3, 4, 5 and 15: the sinusoidal targets were incorrectly reported without \pi. For example, we reported \sin(x) rather than \sin(\pix). The updated captions now correctly reflect the actual target profiles used in the experiments.

---

> ### Author Response · Authors · 2025-11-25
> **Authors' response (3/3)**
>
> Many thanks again for engaging the discussion. We genuinly appreciate the reviewer's feedback and request of clarifications and hope to have addressed his concern in the revised version of the paper.
>
> **References**
> \
> [1] F. Fioretto, T. W. K. Mak & P. Van Hentenryck, “Predicting AC Optimal Power Flows: Combining Deep Learning and Lagrangian Dual Methods,” in Optimisation and Engineering, 2020 (preprint: 2019).
> \
> [2] P. L. Donti, D. Rolnick & J. Z. Kolter, “DC3: A Learning Method for Optimization with Hard Constraints,” in International Conference on Learning Representations (ICLR), 2021.
> \
> [3] S. Park & P. Van Hentenryck, “Self‑Supervised Primal‑Dual Learning for Constrained Optimization,” in Proceedings of the 37th AAAI Conference on Artificial Intelligence (AAAI 2023), vol. 37, pp. 4052–4059.
> \

---

> ### Author Response · Authors · 2025-11-26
> **Follow-up and clarifications**
>
> As the discussion period is nearing its end, we wanted to ask **Reviewer bV8c** if there are any follow-up points we can clarify.
>
> We believe we have responded to all questions and concerns raised, in addition to taking a few days to run the additional experiments necessary to demonstrate our points, all of which are incorporated in the revised version of the paper.
> \
> If there are no further points of clarification regarding the manuscript, we kindly ask that reviewer bV8c consider increasing their score to reflect the improvements and clarifications we have provided. We are happy to continue to engage in discussion and answer any questions!

---

> > ### Comment · Reviewer_bV8c · 2025-11-27
> >
> > Thank the authors for the detailed rebuttal and the additional experiments. The new results and explanations address most of my earlier concerns, and I now have a much clearer view of the method and its behavior.
> >
> > Based on the revisions and your responses, I will adjust my rating to 4.

---

> > > ### Author Response · Authors · 2025-11-27
> > >
> > > Dear Reviewer bV8c,
> > >
> > > Thank you very much for updating your rating in light of our rebuttal and the new experiments. We appreciate the time you have invested in engaging with the paper and in formulating your clarifying questions.
> > >
> > > We would like to briefly mention that, in addition to provide a detailed response regarding your points on the perceived level of novelty of PDE-OP and the speed–accuracy trade-off relative to classical and learning-based baselines, we also have:
> > > - expanded the experimental suite with 2D heat control, spatially varying coefficients, and more complex targets;
> > > - added modern operator-learning baselines (SBTO and DeepONet MPC) and explained why they are slower in practice despite using surrogates;
> > > - provided a physical explanation for the boundary effects and the behavior you observed in several figures, and discussed straightforward ways to further reduce these errors in future work.
> > >
> > > We believe that these revisions address the substantive concerns you outlined. If there are no unresolved issues regarding soundness or quality and given that the core methodological and empirical questions have been substantially clarified and strengthened, we would be very grateful if you could consider whether a score at or above the acceptance threshold might better reflect your updated assessment of PDE-OP. Regardless of the final rating, we sincerely thank you again for your careful and constructive feedback, which has helped us improve the work in important ways.
> > >
> > > With best regards,
> > >
> > > -The authors

---

### Official Review · Reviewer_Wb68 · 2025-11-01

**Soundness:** 3
**Presentation:** 3
**Contribution:** 3
**Rating:** 8
**Confidence:** 4

**Summary:**

Strengths: The paper is well-written and on the topic of PDE-constrained optimization, where ML can play a significant role. Authors use many recent developments in smart ways to show that their method solves several typical benchmark problems in this area much faster than competing methods. The selected problems vary in complexity (e.g. nonlinearity, constraint types).

Weaknesses: Although there have been some recent works tackling this problem using ML-based approaches, the authors compare only with traditional PDE-constrained optimization based methods, without any ML speed-up. The authors discuss relevant work, but it is still unclear whether the methods selected for comparison are state-of-art. Also, a major gap in the paper is that the data requirements of the method are not discussed in the paper at all.

**Strengths:**

Originality: The key novelty is the proposal of the dual-network architecture that jointly learns PDE system dynamics and optimal control decisions. While prior work has explored neural operators for PDE surrogates and proxy optimization for control, this paper is original in explicitly coupling these two components.

Quality: The paper is well-executed technically, with a carefully designed model architecture. Comparisons against classical methods reveal compelling speed-ups, which can be significant for applications requiring very fast solutions.

Clarity: The paper and formulations are clear and logically structured.

Significance: PDE-governed systems based control has broad applicability across domains where PDEs arise, including fluid dynamics, thermal systems, chemical and biological processes, power networks, materials design and many more.

**Weaknesses:**

1. While the selected PDE benchmarks (Burgers’, heat equation, voltage regulation) are commonly used in learning-based PDE research, they remain relatively low-dimensional compared to many real engineering PDE systems. The authors mention several times in their paper that this problem is intractable but the problems they are comparing to are not intractable. Should the authors include a higher-dimensional problem or discuss limitations of their approach when scaling up, it would significantly improve the paper quality. Essentially, do the authors believe their approach can solve problems currently intractable?
2. The paper comparative results focus only on classical optimal control methods. Although these comparisons are compelling and necessary, the paper would be strengthened by comparisons or discussion relative to recent learning-enabled PDE control methods. Some of these are mentioned in "Related Work", without sufficient justification on why they were not selected for comparison. I suggest either better justification on why other methods are not applicable or addition of comparison with at least one state-of-art learning-based method.
3. Although most learning-based PDE constrained papers in the ML field show results with these benchmarks, it is true that these problems do not capture all characteristics of real problems in control (e.g.,  noisy PDE parameters, limited data to learn from and measurement uncertainty). On this, the paper provides almost no information on how much data was required to train the models in order to achieve such accuracy. This information might be hidden somewhere in the Appendix, but it is important information to be more prominently provided, along with a discussion in at least the appendix on whether real control applications would have the data quality and quantity required to replicate these results.
4. The authors claim that other learning based methods suffer from feasibility problems and non guarantee of constraint satisfaction. Does this work not have such limitations? Theoretical conditions for success are not deeply explored (e.g.,  when will the learned surrogate preserve feasibility or approximate optimality sufficiently? Please provide theoretical discussion or bounds (even qualitative) regarding approximation error between learned and true solution maps, or articulate assumptions under which the method is expected to succeed.

**Questions:**

1. Do the authors believe their approach can solve problems currently intractable? What are some intractable examples and how would the method scale to these?
2. Can such accuracies and speed-ups be expected only for systems which are perfectly described by a known PDE with highly certain parameters, known initial/boundary conditions (?) and certain assumptions about the ability to generate enough scenarios for training? What are the conditions to guarantee such performance?
3. What are sampling requirements and what are costs for pre-training? Is the method expected to off-load all pre-training costs offline and not expected to require any resampling and retraining in an online control setting? Under which conditions would this be true?
4. What is most relevant learning-based method and can it be added as a baseline method to compare to?
5. One key advantage of paper is the dual-network architecture. And presumably this increases the performance of the method because it enables information to be learned between state dynamics and optimal control which un-coupled learning techniques do not take advantage of. Can this advantage be somehow quantified however, other than the fact that the overall approach is fast? Can the authors think of a test to show a de-coupled approach, to show that there is truly something in the coupling that leads to such better results. I am not sure if this question makes sense, please correct me if I am understanding this wrong.
6. The results plots (Figures 3-5) are not the best quality and distinguishing between the different lines is hard to see.
7. Minor typo on row 287: "operator"

---

> ### Author Response · Authors · 2025-11-21
> **Authors' response**
>
> We thank the reviewer for the helpful review and, in particular, for
> acknowledging the clarity of presentation, the cleverness of the
> proposed learning-based, dual-network architecture and the computational
> speedup that it enables. We have also found useful the reviewer'
> suggestions and integrated them in the revised version of the paper. In
> what follows we will: **1)** provide a detailed answer to your
> questions, **2)** describe the experimental settings and results
> obtained running the experiments you suggested and **3)** report the
> revised part of the paper in blue.\
> We would like to remind you that the core contributions of our work are:
> **1)** a novel discrete-time neural operator to capture PDE-dynamics,
> **2** a dual-network architecture that integrates system dynamics and
> optimization objective and constraints within an unified primal-dual
> loss, and **3)** a novel, fully differentiable learning-based method for
> solving PDE-constrained optimization problems in near real-time. Our
> claims are supported by a comprehensive experimental evaluation, which
> has been further enriched to integrate the reviewers' useful feedback.
>
> Responses to weaknesses pointed out by the reviewer:
>
> -   **While the selected PDE benchmarks (Burgers', heat equation, voltage
>     regulation) are commonly used in learning-based PDE research, they
>     remain relatively low-dimensional compared to many real engineering
>     PDE systems. The authors mention several times in their paper that
>     this problem is intractable but the problems they are comparing to
>     are not intractable. Should the authors include a higher-dimensional
>     problem or discuss limitations of their approach when scaling up, it
>     would significantly improve the paper quality. Essentially, do the
>     authors believe their approach can solve problems currently
>     intractable?**
> \
>     We thank the reviewer for raising this important point. We would
>     like to clarify that when we refer to PDE-constrained optimal
>     control as "intractable", we are referring specifically to real-time
>     tractability. Classical methods such as those used in the paper for
>     comparison (e.g., MPC and NMPC, adjoint-based optimization) can
>     solve PDE-constrained optimization problems, but they struggle to do
>     so in real time because they require repeated optimization calls or
>     backward adjoint integrations, which cause high inference time as
>     reported in Table 1-11 of the main paper. Our results demonstrate
>     that PDE-OP can produce high-quality solution approximations in near
>     real-time, thereby addressing precisely the computational bottleneck
>     we refer as intractable.
>
>     On the limitations that could arise in higher-dimensional settings.
>     To investigate on this aspect, we have conducted additional
>     experiments on the Optimal Control of the 2D Heat Equation task,
>     detailed in Appendix 9.8.3, with results discussed in Appendix
>     9.9.4. The experimental results show that PDE-OP continues to
>     produce decision quality comparable to those of the base baselines,
>     while running several order of magnitude faster. This suggests that
>     PDE-OP remains effective and scalable even as problem size
>     increases. More generally, we agree that this aspect would merit
>     further investigation. That is because increasing dimensionality can
>     lead to stronger coupling between state and decision variables. In
>     such settings, approximation and propagation errors, especially
>     those introduced by the dynamic surrogate, could compound more
>     noticeably, particularly given that PDE-OP uses a dual network
>     architecture and a primal-dual based method for training, which
>     integrates both dynamic and optimization objectives.
>
>     That said, these challenges are common to all learning-based PDE
>     surrogates. While higher-dimensional settings may introduce some
>     loss of accuracy due to increased coupling and error propagation,
>     our method doesn't suffer any losses in computational efficiency, as
>     shown in Table 5 of the Appendix: due to the nature of PDE-OP,
>     inference remains extremely fast and essentially independent of the
>     problem size. This makes PDE-OP a particularly appealing candidate
>     for extension to even higher-dimensional problems, which we view as
>     a promising direction for future works.

---

> ### Author Response · Authors · 2025-11-21
> **Authors' response**
>
> -   **The paper comparative results focus only on classical optimal
>     control methods. Although these comparisons are compelling and
>     necessary, the paper would be strengthened by comparisons or
>     discussion relative to recent learning-enabled PDE control methods.
>     Some of these are mentioned in \"Related Work\", without sufficient
>     justification on why they were not selected for comparison. I
>     suggest either better justification on why other methods are not
>     applicable or addition of comparison with at least one state-of-art
>     learning-based method.**\
>     We appreciate the reviewer's suggestion and agree that comparisons
>     with recent learning-enabled PDE control methods would further
>     strengthen the paper. Due to time constraints and the substantial
>     engineering effort required to implement and tune many of these
>     approaches, several of which are highly problem-specific or lack
>     readily available open-source implementations, we were unable to
>     include a broader set of learning-based baselines in the initial
>     submission.
>
>     That said, we fully recognize the value of such comparisons. In the
>     revised version of the paper (see Appendix 9.6--9.7), we have added
>     two representative learning-based baseline methods for
>     PDE-constrained optimization:
>
>     -   Surrogate-Based Trajectory Optimization (SBTO) [1],
>         an open-loop control strategy where a neural surrogate of the
>         dynamics is used within a full-horizon optimization problem.
>
>     -   DeepONet MPC [2], a receding-horizon, closed-loop
>         control method that repeatedly solves short-horizon
>         optimizations using the same surrogate dynamics.
>
>     Both approaches leverage the same surrogate dynamic predictor as our
>     method but differ in the control strategies they employ, namely
>     open-loop trajectory optimization for SBTO and receding-horizon
>     optimization for DeepONet MPC. From the numerical results reported
>     in Tables 1-4 and the qualitative plots illustrated in Figure 3-5
>     and Figure 9, we observe that PDE-OP consistently achieve accuracy
>     comparable to these methods, while running several order of
>     magnitude faster. These speed-ups stem from PDE-OP's surrogate
>     controller, which enables direct inference. In contrast, both SBTO
>     and DeepONet MPC rely on a neural operator for modeling dynamics but
>     still require solving an optimization problem at each iteration. We
>     believe that including these learning-based methods as baselines
>     further strengthen the comprehensiveness of our experimental
>     evaluation.
>
> -   **Although most learning-based PDE constrained papers in the ML field
>     show results with these benchmarks, it is true that these problems
>     do not capture all characteristics of real problems in control
>     (e.g., noisy PDE parameters, limited data to learn from and
>     measurement uncertainty). On this, the paper provides almost no
>     information on how much data was required to train the models in
>     order to achieve such accuracy. This information might be hidden
>     somewhere in the Appendix, but it is important information to be
>     more prominently provided, along with a discussion in at least the
>     appendix on whether real control applications would have the data
>     quality and quantity required to replicate these
>     results.**\
>     We thank the reviewer for raising this point. Understanding the data
>     requirements of PDE-OP is important for assessing its applicability
>     in practical settings, and we appreciate the reviewer's feedback on
>     this aspect. For each experiment described in the paper, we
>     generated $5.000$ trajectories for training, $1.000$ for validation
>     and $1.000$ for testing the discrete-time neural operator
>     $\mathcal{Y}_\theta$, and $1.000$ samples each to train, validate
>     and test the PDE-OP model. We added these information to the
>     Appendix, Tables 12, 16, 21, 26 and 31 of the revised version of the
>     paper.

---

> ### Author Response · Authors · 2025-11-21
> **Authors' response**
>
> -   **The authors claim that other learning based methods suffer from
>     feasibility problems and non guarantee of constraint satisfaction.
>     Does this work not have such limitations? Theoretical conditions for
>     success are not deeply explored (e.g., when will the learned
>     surrogate preserve feasibility or approximate optimality
>     sufficiently?). Please provide theoretical discussion or bounds
>     (even qualitative) regarding approximation error between learned and
>     true solution maps, or articulate assumptions under which the method
>     is expected to succeed.**\
>     We appreciate the reviewer's insightful question. The concern
>     regarding feasibility and the lack of formal guarantees is indeed a
>     well-known and important challenge for proxy optimizers. Our
>     approach is not exempt from this: PDE-OP employs a proxy optimizer
>     method and a Lagrangian-based training objective that incorporates
>     the PDE constraints and static constraints via penalty and dual
>     variables. This mechanism is designed to encourage feasibility, and
>     in practice it works very well. Our experiments show that PDE-OP
>     produces trajectories that adhere the constraints functions.
>     Nonetheless, as with other proxy optimizer approaches, this does not
>     constitute a formal guarantee of exact constraint satisfaction in
>     all cases.
>
>     More broadly, most proxy optimizer methods do not focus on deriving
>     general theoretical conditions under which feasibility or exact
>     optimality is guaranteed, as they typically consists of training a
>     neural network using gradient-based optimization techniques, which
>     makes it inherently difficult to derive general theoretical
>     guarantees of feasibility or exact optimality. The main goal of
>     proxy optimizer methods is to learn high-quality approximate
>     solutions for real time applications, especially for complex,
>     nonconvex problems where classical guarantees are challenging to
>     obtain and numerical solvers often face scalability or convergence
>     issues. PDE-OP follows this paradigm: it can handle general
>     nonlinear PDE-constrained problems without requiring any assumptions
>     such as convexity or linearity, both in the system dynamics and the
>     optimization objective and constraint functions, and it performs
>     well across a variety of settings, as we show in the experiments.
>     Providing rigorous bounds on the approximation error between the
>     learned solution map and the true optimal control, or characterizing
>     conditions under which feasibility is always preserved, would
>     require substantial additional assumptions on the PDE, the control
>     space, and the neural architectures, and lies beyond the scope of
>     this work. We view such theoretical analysis as an interesting
>     direction for future research, complementary to the empirical
>     evidence presented here.
>
> Response to the reviewer questions
>
> -   **Can such accuracies and speed-ups be expected only for systems
>     which are perfectly described by a known PDE with highly certain
>     parameters, known initial/boundary conditions (?) and certain
>     assumptions about the ability to generate enough scenarios for
>     training? What are the conditions to guarantee such
>     performance?**/
>     We thank the reviewer for raising this important question. It is
>     useful to distinguish between the factors that influence accuracy
>     and those that influence speed, as they are affected very
>     differently.
>
>     **Accuracy**
>     \
>     The accuracy of PDE-OP does depend on the data used to
>     initialize the dynamic predictor and to train end-to-end the PDE-OP
>     model. When the PDE parameters are uncertain or when
>     initial/boundary conditions are noisy, one can train the neural
>     operator over a range of possible realizations of these parameters
>     to improve robustness. In this case, the uncertain parameters can
>     simply be treated as additional inputs to the neural operator, as is
>     common in parametric operator learning. This increases the size and
>     complexity of the training distribution, which may lead to a modest
>     degradation in accuracy; intuitively, the more the uncertainty, the
>     broader the input space the model must cover, the more complex the
>     training. Nonetheless as the neural operator component is well
>     suited to handling parameterized PDE families, we expect PDE-OP to
>     retain good performance also under uncertainty. To address this
>     comment, we run additional tests on the 1D Burgers' equations with
>     varying viscosity coefficient. From Table 7 and 8, we can observe
>     that for the this experiment, PDE-OP delivers robust
>     control performance across diverse targets. As per the other
>     experiments, our method matches or exceeds the accuracy of the
>     baselines while offering three orders of magnitude in
>     inference speed.

---

> ### Author Response · Authors · 2025-11-21
> **Authors' response**
>
> **Speed**
>     \
>     In contrast, the speed-ups reported in the paper are not
>     dependent on assuming perfect knowledge of the PDE. Once trained,
>     PDE-OP's inference simply requires to apply iteratively a forward
>     pass through a neural network until convergence, and its runtime is
>     essentially independent of the dimensionality of the PDE, the
>     structure of the boundary/initial conditions, or the uncertainty
>     degree. Thus, the real-time speed benefits are inherent to the
>     design of PDE-OP and remain valid even under uncertainty. In general, PDE-OP performs best when:
>
>    -   the PDE parameters are known, or, in case of uncertainity, when
>         the training distribution covers the range of PDE parameters and
>         initial/boundary conditions;
>
>    -   the dynamic predictor is trained with sufficient data to capture
>         the variability of the system.
>
>    These are common conditions across all learning-based PDE
>     surrogates, and not specific to PDE-OP. We believe that extending
>     PDE-OP to settings with uncertainty is a promising direction for
>     future works and thank the reviewer for their observation.
>
> -   **What are sampling requirements and what are costs for pre-training?
>     Is the method expected to off-load all pre-training costs offline
>     and not expected to require any resampling and retraining in an
>     online control setting? Under which conditions would this be
>     true?**\
>     We appreciate the reviewer's question and would like to clarify that
>     PDE-OP's design can be seen as an offline (as the neural operator
>     can be initialized offline)--online (as the surrogate controller can
>     be learned online) approach, following the paradigm of neural
>     operators and surrogate-based optimizers. The sampling requirements
>     depend primarily on the complexity of the PDE dynamics one wishes to
>     capture. In practice, for each experiment, we generated 5000 trajectories for training, 1000 for validation and 1000 for
>     testing the discrete-time, neural operator $\mathcal{Y}\theta$, and $1024$ samples each to train, validate and test the PDE OP's $\mathcal{U}_\omega$ model.
>
>     The pre-training cost, which includes both data generation and model
>     training of the discrete-time neural operator, is incurred entirely
>     offline. The trajectories used for training the dynamic predictor
>     can be generated using a PDE-numerical solvers. The PDE-OP's
>     optimization model $\mathcal{U}_\omega$ can be trained online on a
>     self-supervised fashion as described in Section 4.1. This setting
>     does not require precomputing optimal solutions, which represents a
>     key advantage in settings where generating supervised targets by
>     using numerical optimization solvers is costly or infeasible. After
>     training, PDE-OP's inference consists solely of a sequence of
>     forward passes through neural networks and does not require
>     retraining or resampling.
>
>     Retraining may only be required if the governing dynamics shift
>     significantly outside the original data distribution; if the PDE
>     parameters, boundary conditions, and operating regimes remain within
>     the range encountered during training of the neural operator
>     $\mathcal{Y}_\theta$, PDE-OP can be deployed as a real-time
>     controller without the need for resampling or retraining. However,
>     as these assumptions are typical of learning-based PDE surrogates
>     and proxy optimization methods, they do not represent a limitation
>     unique to PDE-OP.
>
>     In summary, the accuracy of PDE-OP depends on having training data
>     that adequately represent the system conditions, while its real-time
>     speed is independent of these assumptions. The method is therefore
>     well suited to scenarios where an offline dataset can be used to
>     initialize the dynamic predictor and the online controller must
>     operate under real-time constraints.
> -   **What is most relevant learning-based method and can it be added as
>     a baseline method to compare to?**\
>     We thank the reviewer for the suggestion. In the revised manuscript
>     (see Appendix 9.6--9.7), we have added two learning-based baseline
>     methods for PDE-constrained optimization:
>
>     -   Surrogate-Based Trajectory Optimization (SBTO) [1], an
>         open-loop control strategy where a neural surrogate of the
>         dynamics is used within a full-horizon optimization problem.
>
>     -   DeepONet MPC [2], a receding-horizon, closed-loop control
>         method that repeatedly solves short-horizon optimizations using
>         the same surrogate dynamics.

---

> ### Author Response · Authors · 2025-11-21
> **Authors' response**
>
> Both approaches leverage the same surrogate dynamic predictor as our
>     method but differ in the control strategies they employ, namely
>     open-loop trajectory optimization for SBTO and receding-horizon
>     optimization for DeepONet MPC. We refer the reader to Appendix 9.6
>     and 9.7 for a formal description of these methods. From the
>     numerical results reported in Tables 1-4 and the qualitative plots
>     illustrated in Figure 3-5 and Figure 9, we observe that PDE-OP
>     consistently achieve accuracy comparable to these methods, while
>     running several order of magnitude faster. These speed-ups stem from
>     PDE-OP's surrogate controller, which enables direct inference. In
>     contrast, both SBTO and DeepONet MPC rely on a neural operator for
>     modeling dynamics but still require solving an optimization problem
>     at each iteration. Finally, we believe that including these methods
>     as baselines further strengthen the comprehensiveness of our
>     experimental evaluation.
>
> -   **One key advantage of paper is the dual-network architecture. And
>     presumably this increases the performance of the method because it
>     enables information to be learned between state dynamics and optimal
>     control which un-coupled learning techniques do not take advantage
>     of. Can this advantage be somehow quantified however, other than the
>     fact that the overall approach is fast? Can the authors think of a
>     test to show a de-coupled approach, to show that there is truly
>     something in the coupling that leads to such better results. I am
>     not sure if this question makes sense, please correct me if I am
>     understanding this wrong.**\
>     We thank the reviewer for the thoughtful question. Our dual-network
>     architecture is motivated by the observation that system dynamics
>     and optimization are inherently coupled: optimal control decisions
>     depend on the system's evolution, and the system evolves as a
>     function of the control decisions. Our design seeks to explicitly
>     capture this interaction by training components that reflect both
>     dynamics and optimality jointly.
>
>     We note that SBTO, which we added as an additional baseline,
>     represents a decoupled approach: the surrogate model is trained
>     independently of the control objective and is used only at test time
>     to predict system trajectories. However, SBTO relies on an exact
>     optimizer that has full access to the surrogate and repeatedly
>     queries it during online optimization, whereas our method embeds the
>     control decision directly within a learned architecture. As such,
>     although SBTO provides a relevant baseline, the differences in
>     supervision and deployment make it difficult to compare between ours
>     and SBTO in a fair manner.
>
> -   **The results plots (Figures 3-5) are not the best quality and
>     distinguishing between the different lines is hard to
>     see.**\
>     We appreciate the reviewer's feedback regarding the visual clarity
>     of Figures 3--5. In the revised version, we have improved the figure
>     quality and adjusted the styling to ensure that the different lines
>     and methods are easily distinguishable.
>
> -   **Minor typo on row 287: \"operator\"**\
>     Thanks for spotting the typo, we fixed it.
>
> Thank you, again, for your time and feedback. We believe that the additional results and explanations included in our response have addressed the concerns presented, but we encourage you to follow-up on any points that remain unclear. Additionally, we encourage the reviewer to check the updated version of our submission, which has been extended to incorporate these additional results and discussions. We hope these revision will merit your strong support for this work!
>
> **References**
> \
> [1] Hwang et al., “Learning to control PDE systems with differentiable programming,” NeurIPS Workshop on Machine Learning and the Physical Sciences, 2022.
> \
> [2] de Jong et al., “Deep Operator Neural Network Model Predictive Control,” IEEE Open Journal of Control Systems, vol. 4, pp. 501–517, 2025.

---

### Author Response · Authors · 2025-11-21
**Authors' comment**

We thank the reviewers for their thoughtful questions and constructive feedback. We have carefully addressed each comment by providing detailed explanations and all the requested experiments to support our claims and strength our evaluation. These additions have been incorporated into the manuscript, with all revisions marked in blue.

We invite the reviewer to review our responses and the updated manuscript. We put significant effort into preparing the rebuttal and believe we have thoroughly addressed all concerns. We welcome further discussion and appreciate the opportunity to improve our work.

---

### Note · Program_Chairs · 2026-01-17
**Submission Desk Rejected by Program Chairs**

The following references in this submission do not refer to real documents and/or have major errors in bibliographic information:

 Maziar Raissi, Paris Perdikaris, and George Em Karniadakis. The deepgreen algorithm for learning generic pdes. Journal of Computational Physics, 420:109707, 2021. doi: 10.1016/j.jcp. 2020. 109707